



# Changes in stability and jumps in Dansgaard–Oeschger events: a data analysis aided by the Kramers–Moyal equation

Leonardo Rydin Gorjão[1,2,3,4,*], Keno Riechers[5,6*], Forough Hassanibesheli[7,5], Dirk Witthaut[1,2], Pedro G. Lind[4,8,9], and Niklas Boers[10,5,11]

[1]Forschungszentrum Jülich, Institute for Energy and Climate Research - Systems Analysis and Technology Evaluation (IEK-STE), 52428 Jülich, Germany
[2]Institute for Theoretical Physics, University of Cologne, 50937 Köln, Germany
[3]German Aerospace Center (DLR), Institute of Networked Energy Systems, Oldenburg, Germany
[4]Department of Computer Science, OsloMet – Oslo Metropolitan University, N-0130 Oslo, Norway
[5]Research Domain IV – Complexity Science, Potsdam Institute for Climate Impact Research, Telegrafenberg A31, 14473 Potsdam, Germany
[6]Department of Mathematics and Computer Science, Freie Universität Berlin, Berlin, Germany
[7]Department of Physics, Humboldt-Universität zu Berlin, Newtonstraße 15, 12489 Berlin, Germany
[8]NordSTAR – Nordic Center for Sustainable and Trustworthy AI Research, N-0166 Oslo, Norway
[9]Artificial Intelligence Lab, Oslo Metropolitan University, N-0166 Oslo, Norway
[10]Technical University of Munich, Germany; School of Engeineering & Design, Earth System Modelling
[11]Global Systems Institute and Department of Mathematics, University of Exeter, United Kingdom
[*]These authors contributed equally to this work.

**Correspondence:** Leonardo Rydin Gorjão (leonardo.rydin@gmail.com)

**Abstract.** Dansgaard–Oeschger (DO) events are sudden climatic shifts from cold to substantially milder conditions in the arctic region that occurred during previous glacial intervals. They can be most clearly identified in paleoclimate records of $\delta^{18}$O and dust concentrations from Greenland ice cores, which serve as proxies for temperature and atmospheric circulation patterns, respectively. The existence of stadial (cold) and interstadial (milder) phases is typically attributed to a bistability of the North

5    Atlantic climate system allowing for rapid transitions from the first to the latter and a more gentle yet still fairly abrupt reverse shift from the latter to the first. However, the underlying physical mechanisms causing these transitions remain debated. Here, we conduct a data-driven analysis of the Greenland temperature and atmospheric circulation proxies under the purview of stochastic processes. Based on the Kramers–Moyal equation we present a one-dimensional and two-dimensional derivation of the proxies' drift and diffusion terms, which unravels the features of the climate system's stability landscape. Our results

10    show that: (1) in contrast to common assumptions, the $\delta^{18}$O proxy results from a monostable process, and transitions occur in the record only due to the coupling to other variables; (2) conditioned on $\delta^{18}$O the dust concentrations exhibit both mono and bistable states, transitioning between them via a double-fold bifurcation; (3) the $\delta^{18}$O record is discontinuous in nature, and mathematically requires an interpretation beyond the classical Langevin equation. These findings can help understand candidate mechanisms underlying these archetypal examples of abrupt climate changes.





# 1 Introduction

In the presence of anthropogenically driven climate change, increasing amount of research focuses on the stability of the climate system's current state (Boers, 2021; Heinze et al., 2021; Rosier et al., 2021; Boers and Rypdal, 2021). Several climatic subsystems have been identified to potentially undergo abrupt transitions if global warming exceeds certain thresholds (Lenton and Schellnhuber, 2007; Lenton et al., 2008; Boers et al., 2021). Such abrupt transitions are often conceptually captured in terms of dynamical systems transitioning between alternative equilibrium states; for example when a stable equilibrium is lost and a system shifts to another attractor in response to the crossing of a bifurcation point, or when a stochastic perturbation pushes the system from one stable state to another. Furthermore, a system can experience rate-induced transitions during which the system fails to track the changing domain of attraction of a given equilibrium state and suddenly transitions to another state. Prominent examples for climate elements which are thought to be at risk of abrupt transitions under sustained anthropogenic greenhouse gas forcing are given by the Greenland Ice Sheet (Boers, 2021), the Amazon rainforest (Boulton et al., 2021), the Atlantic Meridional Overturning Circulation (AMOC) (Boulton et al., 2014; Boers and Rypdal, 2021), and the West Antarctic ice sheet (Rosier et al., 2021).

While the conceptual understanding of abrupt transitions in terms of dynamical system theory is well established, the only observational evidence of abrupt climate transitions stems from proxy records that encode the evolution of past climate variability (Brovkin et al., 2021). Understanding the physical causes of such past abrupt climate changes is crucial for improving the capability of comprehensive Earth System Models to faithfully simulate such transitions, and is therefore a prerequisite for assessing the risk of abrupt climate changes under future warming scenarios (Valdes, 2011; Boers et al., 2021). The most prominent example of past abrupt climate shifts are the Dansgaard–Oeschger events – a series of sudden warming events that dominated Greenland temperatures throughout the last glacial cycle, e.g., Refs. (Johnsen et al., 1992; Dansgaard et al., 1993; North Greenland Ice Core Projects members, 2004). These events have first been revealed in $\delta^{18}O$ record from Greenland ice cores (see Fig. 1), which serve as proxies for past air temperatures at the drilling site (Jouzel et al., 1997; Gkinis et al., 2014). Taking place on time scales of decades, the amplitudes of warming are estimated to exceed $10°C$ in most cases, and reach up to $16°C$ in the annual mean temperature over Greenland (Steffensen et al., 2008; Kindler et al., 2014; Gkinis et al., 2014; Capron et al., 2021). The sudden temperature increase is usually followed by a phase of moderate cooling, before the temperatures ultimately relax back to their pre-event levels in a second phase of more abrupt cooling. While the duration of the relatively warm phases termed Greenland Interstadials (GIs) stretches from centuries to millennia, the cold Greenland Stadials (GSs) typically persists over millennia before another sudden warming event starts a new DO cycle (Rasmussen et al., 2014). Moreover, in addition to $\delta^{18}O$, further Greenland ice core proxies bear the signature of the DO cycles, such as dust (Fuhrer et al., 1999; North Greenland Ice Core Projects members, 2004) and sodium concentrations (Erhardt et al., 2019), or the thickness of the annual deposition layers (Erhardt et al., 2019). The common interpretation of these concomitant abrupt shifts in the different proxy time series is that the sudden Greenland warming events were accompanied by sudden changes in the atmospheric circulation, retreat of the North Atlantic and Nordic Sea's sea ice cover, and increases in the amount of local precipitation, respectively, e.g., Refs. (Li and Born, 2019; Erhardt et al., 2019; Menviel et al., 2020). Importantly, despite substantial progress in recent





years, no general consensus has yet been achieved regarding the mechanism that causes the DO events. Different patterns of
interaction between the AMOC, sea ice cover, the Northern Hemisphere atmospheric circulation, and even the continental ice
sheets have been proposed and explored to explain the emergence of DO cycles (Broecker et al., 1985; Li et al., 2005; Petersen
et al., 2013; Cimatoribus et al., 2013; Dokken et al., 2013; Zhang et al., 2014; Kleppin et al., 2015; Lynch-Stieglitz, 2017; Vet-
toretti and Peltier, 2018; Boers, 2018; Boers et al., 2018; Li and Born, 2019; Gottwald, 2020; Menviel et al., 2020; Lohmann
et al., 2021).

Here, in order to reconstruct the dynamics that governed Greenland temperatures during the last glacial, we assume an
inverse modelling approach, in a fashion similar to the analysis conducted in, e.g., Refs. (Ditlevsen, 1999; Livina et al., 2010;
Kwasniok, 2013; Krumscheid et al., 2015; Boers et al., 2017; Hassanibesheli et al., 2020). The key concept is to regard
the paleo-climate record as the realisation of a Markovian and stationary stochastic process (Kondrashov et al., 2005, 2015)
which can be described in terms of a stochastic differential equation. In this setting there exist different ways to estimate the
deterministic drift and the stochastic diffusion component. Drawing on the Kramers–Moyal equation (Kramers, 1940; Moyal,
1949; Tabar, 2019), we recover the underlying drift term – or the potential landscape – that discloses the stability configuration
of the system, together with the diffusion term. In particular, we present the first Kramers–Moyal-based reconstruction of
the two-dimensional drift in the coupled system comprised of Greenland ice core $\delta^{18}$O values and dust concentrations. The
Kramers–Moyal equation generalises the Fokker–Planck description of stochastic processes, including explicitly the presence
of discontinuous elements. In this sense, it steps outside the classical description of a Langevin process, yet preserves a similar
interpretation of the drift and diffusion of the processes.

This article is structured as follows: In Sec. 2 we introduce the paleo-climatic proxies under examination and the detrending
method used to ensure that the data is approximately stationary. In Sec. 3 we introduce the Kramers–Moyal expansion as
the prime method to extract the potential landscape and examine the influence of stochastic noise and possible discontinuous
elements within the records. In Sec. 4 we present the results, beginning with a one-dimensional analysis of the two proxies, in
Sec. 4.1. We examine their separate potential landscapes and higher-order Kramers–Moyal coefficients, and therein manifest
the need to augment our examination to a two-dimensional plane. This is consequently discussed in Sec. 4.2, where we uncover
the conditioned potential landscapes of the joint proxy process. We examine the stability configurations of the $\delta^{18}$O and dust
time series in respective comparison, unveiling the arguments for mono-stability of the $\delta^{18}$O, and a mixed set of states for the
dust, which undergoes an double-fold bifurcation parametrised by the $\delta^{18}$O. In Sec. 6 we discuss the results and consequences
of our findings in a paleoclimate context.

## 2 Data and pre-processing

The analysis presented here is based primarily on the joint $\delta^{18}$O and dust concentration time series obtained by the North
Greenland Ice Core Project (NGRIP) (Ruth et al., 2003; North Greenland Ice Core Projects members, 2004; Gkinis et al.,
2014). The concentration of stable water isotopes expressed as $\delta^{18}$O values in units of permil is a proxy for the site temperature
at the time of precipitation (Jouzel et al., 1997; Gkinis et al., 2014). The concentration of dust, i.e. the number of particles





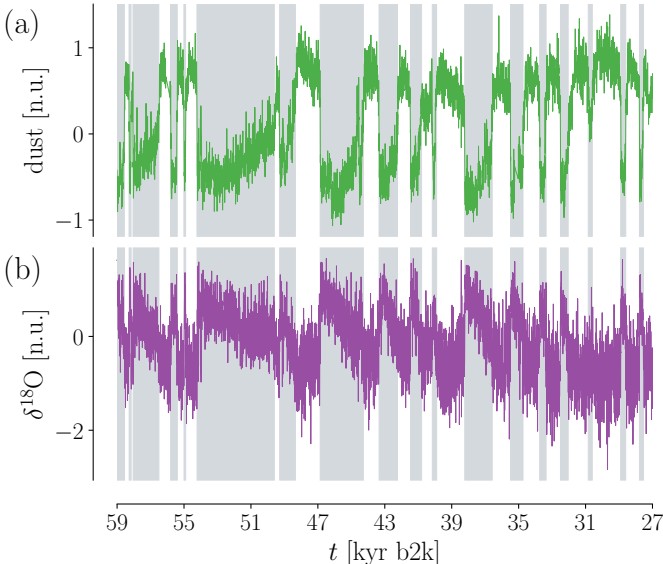

**Figure 1.** The dust (a) and the $\delta^{18}O$ (b) records from the NGRIP ice core in Greenland, from 59 kyr to 27 kyr before the year 2000 B.C.E. (b2k). The dust data is the natural logarithm of the actual dust concentrations, in order to facilitate visual comparison to the $\delta^{18}O$. All time series are normalised. The two proxies have been pre-processed, in order to ensure stationarity, by removing a linear trend drawn from the global average surface (see App. A). The grey shadings mark the Greenland Interstadial (GI) intervals. The time series can be found in (J. P. Steffensen, Centre for Ice and Climate, Niels Bohr Institute, 2014; Seierstad et al., 2014).

with diameter above one micron per millilitre is assumed to be controlled mostly by two factors: First, climatic conditions at the emission source, that is, the dust storm activity over East Asian deserts preconditioned on generally dry regional climate. Second, the transport efficiency, which is affected by the strength and position of the polar jet stream (Ruth et al., 2007; 85 Schüpbach et al., 2018; Erhardt et al., 2019). Correspondingly, the substantial changes in the dust concentrations across DO events are interpreted as large-scale reorganisations of the Northern Hemisphere's atmospheric circulation (Erhardt et al., 2019). Since the dust concentrations approximately follow an exponential distribution, we consider in the following re-scaled values by taking the natural logarithm and multiplying by $-1$ in order to emphasise the similarity to the $\delta^{18}O$ time series (cf. Fig. 1). For ease of notation we will use the term dust although technically we refer to the negative natural logarithm of the 90 dust concentration.

Data is available for both proxies at an equidistant resolution of 5 cm, from 1346.45 m to 2426.00 m of depth in the NGRIP ice core. This translates into non-equidistant temporal resolution ranging from sub-annual resolution at the beginning to $\sim 5$ years at the end of the period 9527.3 – 59944.5 yr b2k (before the year 2000). All ages are according to the Greenland Ice Core Chronology 2005 (GICC05), the common age-depth model for both proxies (Vinther et al., 2006; Rasmussen et al., 2006; 95 Andersen et al., 2006; Svensson et al., 2008).





The analysis conducted in this work assumes stationarity of the underlying data-generating process. However, Boers *et al.* (Boers, 2018) pointed out a low-frequency influence of the background climate, for example, expressed in terms of global ice volume, on the frequency of DO events, whith suppressed DO variability during the coldest parts of the glacial such as the Last Glacial Maximum. We therefore complement the NGRIP dust and $\delta^{18}$O data with a reconstruction of the global average surface temperature as presented in Ref. (Snyder, 2016). Moreover, Boers *et al.* (Boers et al., 2017) highlighted that $\delta^{18}$O and dust show high co-variability over the period 59–22 kyr b2k. However, this relationship weakens after 22 kyr b2k, thus constraining the potential data segment eligible for our study. Excluding also the Last Glacial Maximum from the data, we restrict our analysis to the period 59–27 kyr b2k, which is characterised by a fairly stable background climate and persistent co-variability between dust and $\delta^{18}$O. To compensate for the remaining influence of the background climate on the climate proxy records, we remove a linear trend with respect to the global average surface temperature from both time series (see App. A for the details). After the detrending we consider the data as the outcome of an approximately stationary process. As the final step of the pre-processing the data is binned to temporally equidistant increments of 5 years and normalised with respect to the average amplitude of the DO transitions.

Lastly, the analysis in this work is based on Markovian stochastic processes. To assess whether the data is Markovian, we analyse the auto-covariance function of the increments of the detrended data. The covariance is largely zero everywhere, except for a weak anti-correlation at the shortest lag, supporting the assumption that the data is approximately Markovian (see App. B).

## 3  Methods

In this section, we present the methods used to study the coupled $\delta^{18}$O and dust dynamics, drawing on the theory of stochastic processes and stochastic differential equations. Starting from the well-known Langevin process, we introduce the broader Kramers–Moyal (KM) framework. The Fokker–Planck equation associated with a Langevin process will be expanded to the more general Kramers–Moyal equation in one and two dimensions. While the KM setting is suited for the treatment of complex noise, the interpretation of a deterministic drift component is preserved. This allows us to investigate the stability configuration of the coupled $\delta^{18}$O–dust system. We begin by shortly introducing one-dimensional stationary Markovian stochastic processes.

### 3.1  Continuous and discontinuous stochastic processes

A one-dimensional stochastic process is a mapping from time $t \in \mathbb{R}$ into some adequate state space that describes the dynamics of a random variable $x(t)$, subject to random fluctuations. A prominent example for a stochastic process is given by the stationary Langevin equation, a stochastic differential equation of the form

$$\mathrm{d}x(t) = a(x)\mathrm{d}t + b(x)\mathrm{d}B(t), \tag{1}$$

with the drift term $a(x)$, the diffusion term $b(x)$, and an uncorrelated Brownian motion $B(t)$. If the properties of the dynamics do not change over time, i.e. $a(x)$ and $b(x)$ do not depend on time, these processes are called *stationary*. While the Langevin equation is continuous in time, stochastic processes can in principle have discontinuous features, such as sudden jumps. An





easy way to incorporate discontinuities is to include in Eq. (1) an elementary Lévy process $L(t)$, modulated with an amplitude $h(x)$ (Applebaum, 2011)

$$\mathrm{d}x(t) = a(x)\mathrm{d}t + b(x)\mathrm{d}B(t) + h(x)\mathrm{d}L(t). \tag{2}$$

130 The interpretation of $a(x)$ and $b(x)$ as drift and diffusion remains preserved under this generalisation. Note that Langevin processes are just a subclass of Lévy processes, and all these processes are Markovian.

## 3.2 The one-dimensional Kramers–Moyal equation

Stochastic processes can be either described in terms of random variables, following a stochastic differential equation as introduced above, or in terms of the evolution of their conditional probability density function $p(x,t|x',t')$, following a par-

135 tial differential equation. If a single particle's motion is governed by the Langevin equation, its probability density function $p(x,t|x',t')$ evolves according to the Fokker–Planck equation, given by

$$\begin{aligned}\frac{\partial}{\partial t}p(x,t|x',t') &= -\frac{\partial}{\partial x}D_1(x)p(x,t|x',t') \\ &+ \frac{\partial^2}{\partial x^2}D_2(x)p(x,t|x',t'),\end{aligned} \tag{3}$$

which we consider in a stationary case as above, i.e. without explicit time dependence of the coefficients $D_1(x)$ and $D_2(x)$. These coefficients directly relate to the drift and diffusion terms given in Eq. (1) by

140 $$D_1(x) = a(x), \tag{4a}$$

$$D_2(x) = \frac{1}{2}b^2(x). \tag{4b}$$

The Fokker–Planck equation can only describe continuous processes (Risken and Frank, 1996; Stemler et al., 2007; Gardiner, 2009; Tabar, 2019). Giving up the condition of continuity, the temporal evolution of the conditional probability density follows the Kramers–Moyal equation

145 $$\frac{\partial}{\partial t}p(x,t|x',t') = \sum_{m=1}^{\infty}\left(-\frac{\partial}{\partial x}\right)^m D_m(x)\,p(x,t|x',t'), \tag{5}$$

where $D_m(x)$ denotes the $m$th Kramers–Moyal (KM) coefficient, defined from the corresponding conditional moments $M_m(x,\tau)$ of the variable $x$ and a time-lag $\tau$, i.e.

$$\begin{aligned}D_m(x) &= \frac{1}{m!}\lim_{\tau\to 0}\frac{M_m(x,\tau)}{\tau} \\ &= \frac{1}{m!}\lim_{\tau\to 0}\frac{1}{\tau}\langle(x(t+\tau)-x(t))^m|_{x(t)=x}\rangle,\end{aligned} \tag{6}$$

where $\langle\cdots\rangle$ denotes the expected value. If a stochastic process is 'sufficiently' continuous, the third and all higher KM coef-

150 ficients vanish according to Pawula's theorem (Pawula, 1967a, b), and the Kramers–Moyal equation (Eq. (5)) reduces to the simpler Fokker–Planck (Fokker–Planck–Kolmogorov) equation (Eq. (3)). While the Langevin equation is the direct counterpart



of the Fokker–Planck equation, for the Kramers–Moyal equation a single particle's equation of motion can assume different functional forms that describe different (discontinuous) stochastic processes (one example is given by Eq. (2)). However, for numerous of these stochastic processes, the KM coefficients can be related to the properties of the stochastic process in the spirit of Eq. (4). Importantly, interpretation of the first and second KM coefficients as the drift and diffusion terms, respectively, is preserved in the presence of discontinuities.

In practice, as can be understood from Eq. (6), one of the pivotal elements of using a description such as the Kramers–Moyal equation is the possibility to estimate the coefficients $D_m(x)$ directly from data. To retrieve the KM coefficients $D_m(x)$ from a single realisation of a stochastic process, i.e. a single time series, we evaluate the transition probability densities in the limit of a vanishing time step $\tau \to 0$, which numerically corresponds to considering the shortest increment $\Delta t$ in the data ($\tau \to \Delta t$). In other words

$$D_m(x) \approx \frac{1}{m!} \frac{1}{\Delta t} \langle (x(t + \Delta t) - x(t))^m \,|_{x(t)=x} \rangle, \tag{7}$$

with which we estimate the various KM coefficients directly from the data. Details on the numerical implementation of this estimation procedure are given in App. C and App. D.

An important remark which will help in the following analysis of the paleoclimate time series is the physical interpretation of the drift. Thinking of the stochastic variable $x(t)$ as the position of a point particle allows to interpret the integral over the drift as a potential landscape

$$V(x) = - \int_{-\infty}^{x} D_1(x') \, \mathrm{d}x' + c. \tag{8}$$

This potential controls the motion of the particle in the sense that the deterministic part of the dynamics will drive the particle to the bottom of potential wells. Stochastic fluctuations, however, may counteract this relaxation and push the particle out of equilibrium and even into another well. We will use this to best illustrate the stability configuration of each proxy.

### 3.3 Distinguishing between continuous and discontinuous processes in time series

The striking difference between the Fokker–Planck equation (3) and the Kramers–Moyal equation (5) is the presence of higher-order Kramers–Moyal coefficients $D_m(x), m > 2$, which arise as the direct consequence of discontinuities in a Markovian stochastic process.

Consider the KM coefficients $D_m(x)$ estimated from the records of a stochastic process as outlined above. A first metric to discern whether this process is discontinuous is to evaluate the ratio of the fourth KM coefficient to the second one, $D_4(x)/D_2(x)$. This roughly compares the size of the the discontinuous paths with the size of the diffusive effects. Thus, values of the ratio $D_4(x)/D_2(x)$ close to zero imply continuous sample paths with no jumps in the data. Values larger than zero, i.e. $\sim 1$, indicate that the jump contribution is of the same order of magnitude as the diffusive contribution (Risken and Frank, 1996; Anvari et al., 2016; Tabar, 2019).

This assessment can be refined by regarding the Lehnertz–Tabar $Q$-ratio (Lehnertz et al., 2018), which takes advantage of the fact that continuous and discontinuous systems 'scale' in a different fashion. While a purely continuous stochastic process

diffuses proportionally to time $t$ (or possibly a power of time $t^\beta$ in anomalous diffusions (Einstein, 1905; von Smoluchowski, 1906; Havlin and Ben-Avraham, 1987)), discontinuous processes can cover large distances in short times, i.e. jump, which causes them to exhibit no scaling relations with time $t$. This can be evaluated via the comparative convergence of the conditional moments $M_m(x,\tau)$ of a stochastic process with the scaling $\tau$ (cf. Eq. (7)), given by

$$Q(x,\tau) = \frac{M_6(x,\tau)}{5M_4(x,\tau)} \sim \begin{cases} \tau, & \text{for diffusions,} \\ k, & \text{for jumpy processes,} \end{cases} \tag{9}$$

where the moments $M_m(x,\tau)$ are given by Eq. (7). If the process is purely diffusive $Q(x,\tau) \sim \tau$ (i.e. a linear function of $\tau$) and if the process yields discontinuous trajectories $Q(x,\tau) \sim k$ (i.e. a constant over $\tau$).

### 3.4 The two-dimensional case

Above, we have introduced the Kramers–Moyal equation for a one-dimensional stochastic process. However, with $\delta^{18}$O and dust representing two different yet coupled climate proxy variables, it is necessary to analyse their records in a combined manner. The two-dimensional Kramers–Moyal equation is given by (Risken and Frank, 1996; Lind et al., 2005; Tabar, 2019; Rydin Gorjão et al., 2019)

$$\begin{aligned} &\frac{\partial}{\partial t} p(x_1,x_2,t|x_1',x_2',t') = \\ &\sum_{i,j=1}^{\infty} (-1)^{i+j} \left( \frac{\partial^{i+j}}{\partial x_1^i \partial x_2^j} \right) D_{i,j}(x_1,x_2) p(x_1,x_2,t|x_1',x_2',t'). \end{aligned} \tag{10}$$

The coefficients $D_{i,j}(x_1,x_2)$ of the two-dimensional Kramers–Moyal equation can be estimated – analogously to the one-dimensional coefficients – from the record of a two-dimensional stochastic process $\mathbf{x}(t) = (x_1(t), x_2(t))$.

The drift terms $D_{1,0}(\boldsymbol{x})$ and $D_{0,1}(\boldsymbol{x})$ carry the same meaning as before. However, relating the higher order two-dimensional KM coefficients to the various parameters of a stochastic differential equation is not straightforward. In a two-dimensional setting, we can still use $D_{1,0}(\boldsymbol{x})$ and $D_{0,1}(\boldsymbol{x})$ to similarly investigate the deterministic part of the underlying dynamics in state space. An intuitive way to understand the motion of a two-dimensional process is to examine the vector field generated by the two drifts, given by

$$\boldsymbol{F}(x_1,x_2) = (D_{1,0}(x_1,x_2), D_{0,1}(x_1,x_2))^\top. \tag{11}$$

For various applications where the fluctuations are not comparable in size, i.e. where the diffusion elements are not of similar scale, one can draw a clearer picture of the motion of the two-dimensional system by referring to an *effective* vector field

$$\boldsymbol{F}_{\text{eff}}(x_1,x_2) = \left( \frac{D_{1,0}(x_1,x_2)}{D_{2,0}(x_1,x_2)}, \frac{D_{0,1}(x_1,x_2)}{D_{0,2}(x_1,x_2)} \right)^\top. \tag{12}$$

This indicates the probable motion of the two-dimensional dynamical variable in a re-scaled temporal axis, i.e. as if the noise contributions were of identical strength, and is an effective tool to disclose the regions of convergence of the coupled system in the two-dimensional state space.





Similarly to the one-dimensional case, one can obtain potential landscapes as integrals over the two drifts:

$$V_{1,0}(x_1|x_2) = - \int_{-\infty}^{x_1} D_{1,0}(x'_1, x_2) \, dx'_1 + C(x_2), \tag{13a}$$

$$V_{0,1}(x_2|x_1) = - \int_{-\infty}^{x_2} D_{0,1}(x_1, x'_2) \, dx'_2 + C(x_1). \tag{13b}$$

The potential landscapes naturally emerge only in a conditional form, since $D_{1,0}(x_1, x_2)$ represents the drift of the first dy-
namical variable $x_1$ conditioned on fixed values $x_2$ of the second dynamical variable. Idem for $D_{0,1}(x_1, x_2)$, representing the
drift of $x_2$ conditioned on $x_1$. Thus, Eqs. 13 (a) and (b) describe the deterministic motion of one variable assuming that the
other variable is kept constant.

The numerical analysis was performed with `python`'s `NumPy` (Harris et al., 2020), `SciPy` (Virtanen et al., 2020), and
`pandas` (Wes McKinney, 2010). Kramers–Moyal analysis was performed with `kramersmoyal` (Rydin Gorjão and Meirin-
hos, 2019) and `JumpDiff` (Rydin Gorjão et al., 2020). Figures were generated with `Matplotlib` (Hunter, 2007).

## 4   Results

This section first discusses the non-parametric estimate of the KM coefficients of the isolated dust and $\delta^{18}$O records in a
one-dimensional setting. Subsequently, the KM analysis of the two-dimensional coupled system is presented in detail.

### 4.1   Stability configuration and continuity of the dust and the $\delta^{18}$O in a one-dimensional setting

We begin by first examining the non-parametric estimates of the KM coefficients of the dust record in a one-dimensional
setting. Panels (a) and (c) of Fig. 2 show the first and second KM coefficients, obtained according to Eq. (7). The reconstructed
potential shown in Fig. 2 (b) exhibits two separate wells, i.e. two distinct minima. This suggests bi-stable dynamics, akin to
what one observes from the trajectories of the dust record (see Fig. 1 (a) for comparison). We find the second KM coefficient
to be fairly constant (Fig. 2 (c)) and the ratio between fourth and second KM coefficients to be negligible (Fig. 2 (d)), which
suggests that a Langevin process with additive noise is a viable description of the isolated dust dynamics. In such a setup, the
noise can stochastically induce transitions from one well to the other in qualitative agreement with the apparent sudden regime
shifts observed in the record. We will revisit the continuity of the dust record after the analysis of the $\delta^{18}$O in one dimension.
Note that the model equations employed here are by construction symmetric with respect to time, therefore, as it is, the model
cannot reproduce the temporal asymmetry that is visually suggested in the dust record.

We proceed with discussing analogous results for the $\delta^{18}$O record. Fig. 2 (e) and (f) display estimates of the drift and the
potential landscape for $\delta^{18}$O, respectively. Most prominently, the drift has only a single stable fixed point (zero-crossing of the
drift), or equivalently the potential function exhibits only a single well. The second KM coefficient is mostly constant, like in
the case of dust (Fig. 2 (g)). With respect to the normalised units, the first and second KM coefficients of $\delta^{18}$O exceed their
counterparts for dust by factors of approximately 3-4 and 10, respectively. This indicates that $\delta^{18}$O exhibits faster dynamics



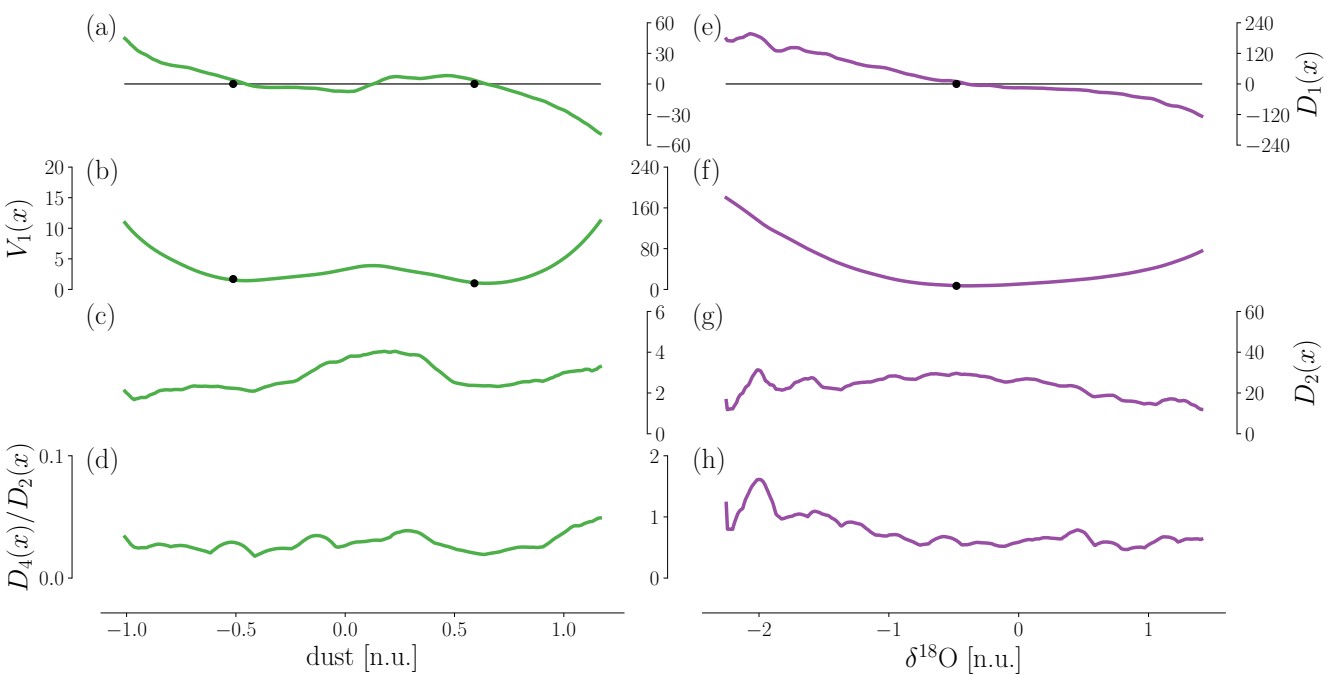

**Figure 2.** The non-parametric estimates of the first KM coefficient $D_1(x)$, the associated potential landscape $V(x)$, the second KM coefficient $D_2(x)$, and the ratio of the fourth to the second KM coefficient $D_4(x)/D_2(x)$. Left column for dust, right column for $\delta^{18}$O. Note that while the dust exhibits a bi-stable potential ((a)-(b)), the $\delta^{18}$O exhibits a mono-stable one ((e)-(f)). The second KM coefficient $D_2(x)$ is constant in both records ((c) and (g)). The ratio $D_4(x)/D_2(x)$ is small for the dust record, yet non-negligible for the $\delta^{18}$O, suggesting that this time series is a realisation of a discontinuous stochastic process. Details on the choice of kernel and bandwidth used for the KM coefficient estimation, as well as an analysis of the influence of the kernel bandwidth, can be found in App. C. A more detailed analysis of the second KM coefficients for both proxies can be found in App. D, which is supplemented by obtaining a corrective term for the Kramers–Moyal coefficients in Eq. (7) by extending the formal solution of the Kramers–Moyal/Fokker–Planck equation in Eqs. (5) and (3).

than dust. Moreover, we find that the fourth KM coefficient $D_4(x)$ for the $\delta^{18}$O is of the same magnitude as the second KM coefficient $D_2(x)$ (Fig. 2 (h)). This points to the potential presence of discontinuities in the record, which we will revisit shortly.

Given the high correlation between the dust and the $\delta^{18}$O records, the differences in the reconstructed potentials and the ratio between fourth and second KM coefficient are remarkable. At first sight, the monostability of the reconstructed $\delta^{18}$O potential contradicts the apparent two regime nature of the time series. There are two possible explanations for this discrepancy: First,

regime switching of monostable stochastic process can be achieved through complex noise structures (e.g., Lévy-like noise, generalised Fokker–Planck equations, or fractal motions) (Chechkin et al., 2003, 2004; Metzler and Klafter, 2004). Secondly, a similar effect can be obtained in a two-dimensional setting if the dynamics of one dynamical variable *explicitly depends* on the other, which would be impossible to judge from the one-dimensional analysis presented so far. Thus, within the limits of this analysis – that is assuming that the process is Markovian and stationary and that the system under study is fully represented by





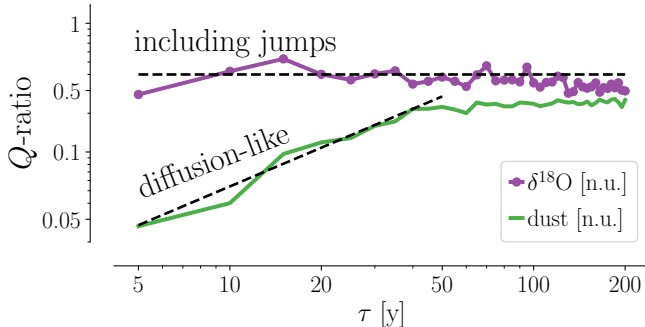

**Figure 3.** The Lehnertz–Tabar $Q$-ratio of the dust and the $\delta^{18}O$ concentration, following Eq. (9), in a double-logarithmic scale. For the $\delta^{18}O$ one observes a constant relation of $Q(x,\tau)$ with $\tau$, indicating that this time series is the realisation of a jumpy (discontinuous) processes. The dust concentration exhibits a linear relation with $\tau$, thus is a purely diffusive process. The state $x$ in $Q(x,\tau)$ is chosen at the maximum of the distribution of the time series.

dust and $\delta^{18}O$ (no coupling to further hidden variables) – the source of the regime switching must either be endowed by more complex noise processes or by the coupling between the dust and the $\delta^{18}O$ systems.

     While we have found the ratio between the fourth and second KM coefficient to be negligible in the case of the dust record, for $\delta^{18}O$ our analysis yields $D_4(x)/D_2(x) \sim 1$. We remind the reader that, for a *continuous* stochastic process $x(t)$, all KM coefficients $D_m(x) = 0, m > 2$, according to Pawula's theorem. When dealing with real world data this is never strictly the
case, of course, thus examining the 'ratio of jumps to diffusive motion', that is $D_4(x)/D_2(x)$, serves only as a first indication if the process is continuous or not. Still, our results suggest that the dust record can be regarded as a realisation of a *continuous* stochastic process on the time scale of 5 yr, while the $\delta^{18}O$ is likely to comprise discontinuities on this time scale. We use the stricter Lehnertz–Tabar $Q$-ratio to underpin our assessment further. In Fig. 3, we clearly see a constant relation of $Q(x,\tau)$ with respect to $\tau$ for the $\delta^{18}O$ record, suggesting that this stochastic process includes jumps. In contrast, we observe a linear
relation between $Q(x,\tau)$ and $\tau$ for the dust count, suggesting a purely diffusive process without jumps. We note here that the presence of correlated forms of noise is also sufficient to generate higher-order Kramers–Moyal coefficients (though not affect their scaling or the $Q$-ratio). However, we exclude this option as the auto-correlation of the increments of the data shows no correlations apart from the shortest increment, as seen in App. B.

## 4.2   $\delta^{18}O$ and dust proxies in a two-dimensional setting

The different stability features in the dust and $\delta^{18}O$ observed in our one-dimensional analysis propel us to study the two proxies in a two-dimensional, coupled setting. This allows us to investigate potential couplings and, as we will show, to reconcile the two-regime nature of the $\delta^{18}O$ time series with the single potential well reconstructed in the one-dimensional analysis. In the following $x_1$ and $x_2$ refer to dust and $\delta^{18}O$, respectively.





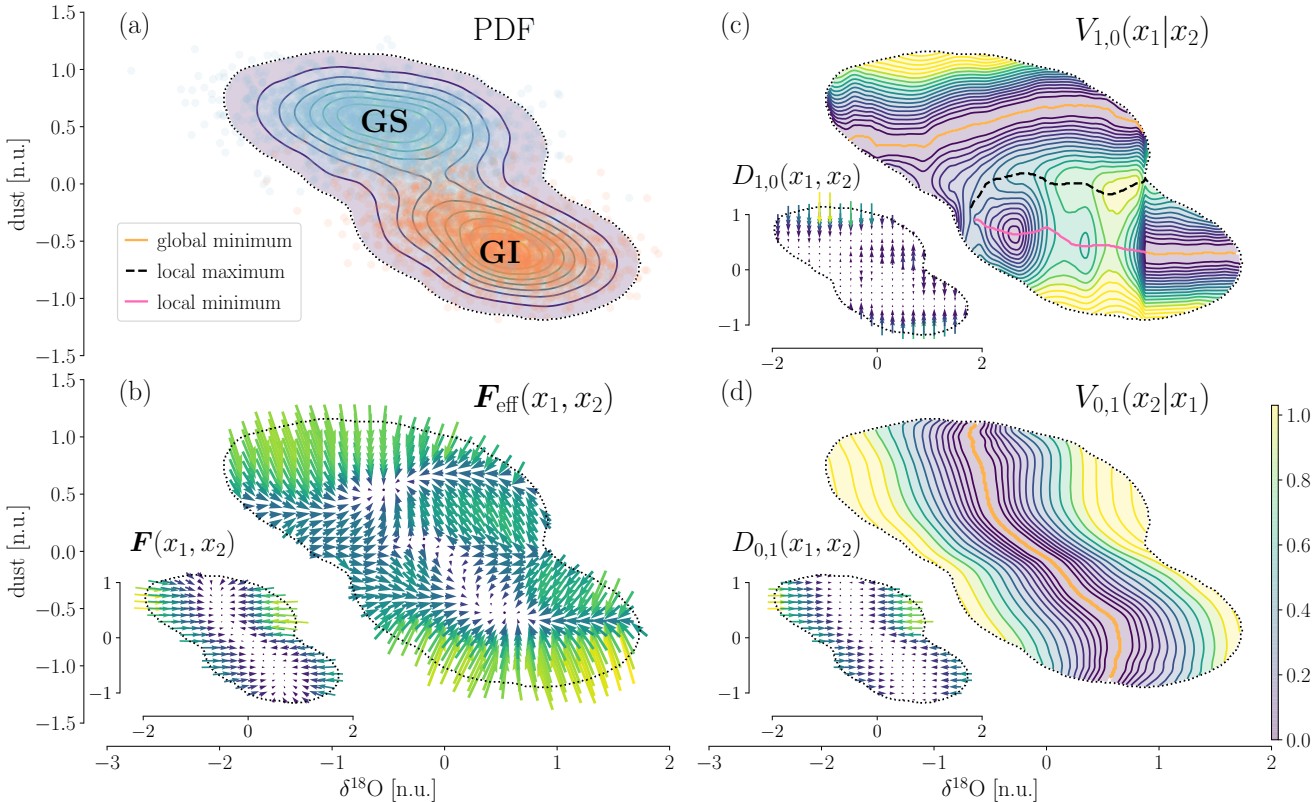

**Figure 4.** Two-dimensional PDF, potential landscapes, and vector fields. In (a) the PDF. The contour indicates a cutoff $> 0.015$ of the PDF, the state space we consider henceforth. The dotted elements are the records, separated into stadials (GS) and interstadials (GI). In (b) the effective vector field $\boldsymbol{F}_{\text{eff}}$ and in the inset $\boldsymbol{F}$. In (c) the potential landscape $V_{1,0}(x_1|x_2)$ of the dust, conditioned on the $\delta^{18}$O. The inset shows $D_{1,0}(x_1, x_2)$. In (d) the potential landscape $V_{0,1}(x_2|x_1)$ of the $\delta^{18}$O, conditioned on the dust. The inset shows $D_{0,1}(x_1, x_2)$. For $V_{0,1}(x_2|x_1)$, in (d), one finds that the location of the minimum of the landscape changes with the value of the dust conentration and undergoes no bifurcation itself. The system is always mono-stable, yet the minimum is not fixed in state space. In stark contrast, the dust landscape $V_{1,0}(x_1|x_2)$, in (c), can show up to three fixed points, depending on the value of $\delta^{18}$O. The system exhibits a *double-fold bifurcation*, transitioning from a single (stable) fixed point for negative values of $\delta^{18}$O, bifurcating to three fixed points (two stable), and again returning to a single (stable) fixed point for positive values of $\delta^{18}$O This offers a good explanation of the apparent 'regime switching' in the dust record, as the system has two stable fixed points co-existing in some regions of the state space. In (b), the effective vector field shows the direction a conceptual particle follows in this two-dimensional space, telling us how $\delta^{18}$O and dust interact and the expected trajectory the coupled system follows.

First we inspect the drift coefficients as before in the one dimensional setting, and reconstruct the conditional potentials according to Eq. (13). This yields two two-dimensional scalar fields whose physical explanatory power is limited to one direction each.





### 4.2.1 Double-fold bifurcation of the dust

The reconstructed conditional potential $V_{1,0}(x_1|x_2)$ of the dust is displayed in Fig. 4 (d). As a conditioned potential, it can be read by taking vertical 'slices' of the potential. Depending on the value of $\delta^{18}O$, the potential of the dust changes from a mono-
stable to a bi-stable regime. Where for approximately $\delta^{18}O < -1.0$ there is only one stable fixed point (a global minimum), for approximately $-1.0 < \delta^{18}O < 0.9$ there are three fixed points, two stable ones (a local minimum and a global minimum) and an unstable one (the local maximum) between them. For approximately $\delta^{18}O > 0.9$ there is again just one stable fixed point (a global minimum). With the position of these stable fixed points depending continuously on $\delta^{18}O$ we find here the characteristic form of a *double-fold bifurcation*. Fig. 4 (d) suggests that the second bifurcation ($\delta^{18}O = 0.9$) is in fact located at a slightly
higher value, since the merger of the upper stable branch and the unstable branch is not fully covered by the reconstruction.

In such a setting, abrupt transitions as those observed in the dust record can happen in two ways: either random fluctuations move the system across the unstable branch (if present, depending on the value of the control parameter) or the control parameter, in this case the $\delta^{18}O$, crosses a bifurcation point and the currently attracting stable fixed point is dissolved. In both cases, the system will transition fairly abruptly to the alternative stable branch. Rate-induced tipping seems unlikely in this case, since
the unstable branch is mostly a constant with respect to a change of the control parameter (i.e. $\delta^{18}O$). This structure prevents to cross the unstable branch by means of a rapid shift in $\delta^{18}O$.

### 4.2.2 Coupling of the $\delta^{18}O$ drift with the dust

We now turn our attention to the $\delta^{18}O$ variable. In Fig. 4 (c) we present the reconstructed potential $V_{0,1}(x_2|x_1)$ for $\delta^{18}O$, conditioned on the value $x_1$ for the dust. Taking a horizontal 'slice' we recover a parabolic shape with a single minimum.
Qualitatively, this feature is preserved across the entire range of potential conditioning dust values. However, the position of the minimum $\delta^{18}O^*$ appears to be determined by the dust in a continuous manner, with high rate of change for intermediate dust values whilst no change for more extreme dust values. Our finding of $\delta^{18}O$ following a mono-stable process is thus confirmed in the two-dimensional analysis, with the added feature that the potential minimum's position is subject to change in response to an 'external control' imposed by the dust.

In summary, in the two-dimensional analysis the monostability of the $\delta^{18}O$ and the bistability of the dust found in the one-dimensional analysis remain preserved. We find a continuous dependency of the position of the stable $\delta^{18}O$ fixed point on the dust. In contrast, a breakdown of the dust's bistability for extreme values of $\delta^{18}O$ can be observed. These findings are consistent with the observed regime switching of both records, which we struggled to reconcile with the results obtained from the one-dimensional analysis. If attracted by the upper stable branch, the dust assumes values on the order of $\sim 0.5$. This
implies a $\delta^{18}O$ stable fixed point position of $\sim -0.6$. A transition in the dust to the lower stable branch associated with dust values $\sim -0.5$ shifts the stable fixed point of $\delta^{18}O$ to $\sim 0.6$. The bistability embedded in the potential governing the dust is thus transferred to the $\delta^{18}O$ and provides an explanation for the observed regime switches of the record. The stable regime ($\delta^{18}O \sim -1.5$, dust $\sim 1$) can be identified with Greenland stadials, while the regime ($\delta^{18}O \sim 2$, dust $\sim -1$) corresponds to Greenland interstadials.





The study of the conditional potentials provided insight in the underlying static stability configuration of the coupled $\delta^{18}$O–dust system. However, to gain a more specific understanding of the dynamics, we investigate the (effective) vector field of motion $\boldsymbol{F}$ ($\boldsymbol{F}_{\mathrm{eff}}$) in the next step.

For each point in state space, the vector field $\mathbf{F}(x_1, x_2) = (D_{1,0}(x_1, x_2), D_{0,1}(x_1, x_2))^{\top}$ indicates the expected direction of movement of the system (see inset of Fig. 4 (b)). One can see that the restoring force in the $\delta^{18}$O direction substantially

exceeds the one in the dust direction. This is in line with the magnitude of both the first and second KM coefficients obtained in the one-dimensional analysis (see Fig. 2) and points to a time scale separation in the dynamics of the coupled system. In the $\delta^{18}$O direction strong fluctuations are quickly compensated by a strong restoring force. In the dust direction, however, the fluctuations and the restoring force are smaller than that of the $\delta^{18}$O, while the ratio between noise and drift is comparable in both directions. This can be interpreted as a time scale separation with fast dynamics happening along the $\delta^{18}$O dimension.

In order to make the dynamics along the dust direction visible, we rescale the two-dimensional potentials in relation to their diffusion (the second Kramers–Moyal coefficients $D_{2,0}$ and $D_{2,0}$). The effective vector field $\mathbf{F}_{\mathrm{eff}}$, obtained according to Eq. (12), is displayed in Fig. 4 (b). While in the un-scaled vector field $\mathbf{F}$ the influence of the dust drift on the coupled system is practically hidden by strength of the $\delta^{18}$O drift, we can now observe more complex structures in the effective vector field. Two main regions of convergence can easily be identified around ($\delta^{18}$O $\sim -0.6$, dust $\sim 0.5$) and ($\delta^{18}$O $\sim 0.6$, dust $\sim -0.5$)

which correspond to Greenland stadials and interstadials, respectively, as mentioned previously. These convergent regions consistently coincide with the two maxima in the two-dimensional density shown in Fig. 4 (a). The effective vector field of motion does not indicate a clear path that the system would take in order to transition between stadial and interstadial states. This leaves open the possibility that transitions between stadial and interstadial states are mainly induced by noise as argued by, e.g., Ref. (Ditlevsen et al., 2007) (i.e. noise-induced tipping), facilitated by a shallow potential barrier close to the minima

of the (effective) vector field.

## 5   Discussion

We have used the one and two-dimensional Kramers–Moyal equation to investigate the combined dust and $\delta^{18}$O record from the NGRIP ice core for the time interval 59–27 kyr b2k, which exhibits pronounced DO variability. The approach was chosen to disclose the dynamical features of this two-dimensional system. In the following, we discuss how our study relates to previously

published investigations of the same data and how it contributes to the broad discourse on DO variability.

Although we obtain slightly different and in parts opposing result, our study ties in naturally with previous data-driven analysis of Greenland ice core proxy data. After the work presented by Boers et al. (Boers et al., 2017) it is only the second study to follow a two-dimensional inverse modelling approach with respect to Greenland ice core data.

Adopting a Langevin-type approach (Ditlevsen, 1999) found the calcium record from the GRIP ice core (Fuhrer et al., 1993)

with annual resolution to be consistent with a bi-stable drift term and $\alpha$-stable noise. We cannot confirm the presence of $\alpha$-stable noise in the dust record – which is often regarded as an equivalent to calcium – from the NGRIP ice core. This might be due to the lower resolution of the data analysed here.



Livina *et al.* (Livina et al., 2010) reported on a changing number of stable states detected in one-dimensional GRIP and NGRIP $\delta^{18}$O and GRIP calcium data with 20 yr and annual resolution, respectively. Their analysis attests bistability of the

$\delta^{18}$O record for the period investigated here for both ice cores. However, throughout the last glacial maximum – which we intentionally excluded from our study – Livina *et al.* (Livina et al., 2010) find monostability consistently in all three time series. Assuming a stationary process for the period 60–20 kyr b2k, the bistability of $\delta^{18}$O potential was later confirmed by Kwasniok (Kwasniok, 2013) based on 50 yr resolution data from NGRIP and GRIP.

Our results suggest that the two regime nature is probably not an intrinsic feature of the $\delta^{18}$O, but rather an effect of the

coupling to other climate variables. Assuming that $\delta^{18}$O exclusively represents local temperatures, this result seems reasonable as one would not expect two distinct stable temperature regimes with all controlling factors kept fixed. In contrast, considering the position of the jet stream a dynamical bistability is far more plausible and could explain the bistability of the dust record found in this study. We note that the bistability of jet stream has been evidenced – although in a somewhat different setting and sense – in reanalysis data of modern climate (Woollings et al., 2010). Evidently, our analysis is limited to solely two climate

proxy variables and the stability of the one can only be assessed conditioned on the other, leaving aside potential coupling to further external factors. Yet our results question the prevailing perception of the $\delta^{18}$O record as the signature of an intrinsically bistable process. This is at least partly in line with the findings of Lohmann et al. (Lohmann and Ditlevsen, 2019), who have shown that a fast-slow limit cycle model outperforms a simple double well model if $\delta^{18}$O is identified with the model's fast component.

Boers *et al.* (Boers et al., 2017) were the first to study the dynamical features of the combined $\delta^{18}$O–dust record. They proposed a third-order polynomial two-dimensional drift in combination with a non-Markovian term and Gaussian white noise to model the coupled dynamics. While our approach is limited to a Markovian setting, it allows for more general forms of the drift and of the noise. In particular, we have shown that the $\delta^{18}$O record cannot be treated as a time-continuous process. The non-vanishing fourth KM coefficient in the $\delta^{18}$O, which indicates forcing beyond typical Gaussian white noise, could

point to an external trigger that directly acts on the Greenland temperatures. A sudden shift in the latter could then entail a regime switch in the atmospheric configuration. However, the interpretation of the fourth KM coefficient is not straightforward and depends on the exact choice of the stochastic process model. The role of discontinuities in the $\delta^{18}$O record merits further investigation. Moreover, it should be mentioned that non-Markovian processes, as proposed in Ref. (Boers et al., 2017), can also give rise to higher-order KM coefficients.

The results obtained in our analysis do not give a clear answer to the question for the exact mechanism that triggered DO events. In principle, the revealed double-fold bifurcation would allow for bifurcation-induced transitions and thus for a limit-cycle behaviour. However, the record show that system does not track the stable fixed point branches until the bifurcation points, but tends to transition earlier (not shown). Also, the structure of the $\delta^{18}$O drift is incompatible with a deterministic cyclic motion in the dust-$\delta^{18}$O plane. In fact, the specific structure of the double-fold bifurcation leaves room for a weak

barrier between stadial and interstadial states in the vicinity of the bifurcation point, thus creating a 'channel'-like passage, through which the system passes.



We conclude therefore that – based on our results – the DO transitions are to large degree induced by noise, acting on the background of a double-fold dynamics governing the dust, for which the $\delta^{18}$O acts as control parameter. These finding do not contradict previous studies that proposed limit-cycle models to explain the $\delta^{18}$O record, since the cyclic motion was not

expected to happen in a state space comprised of Greenland temperatures and atmospheric large scale circulation (Kwasniok, 2013; Lohmann and Ditlevsen, 2019).

## 6   Conclusion

In this article, we have analysed the records of $\delta^{18}$O and dust concentrations from the NGRIP ice core from a data-driven perspective (Ruth et al., 2003; North Greenland Ice Core Projects members, 2004; Gkinis et al., 2014). The central point of our

study was to examine the stability configuration of the coupled $\delta^{18}$O–dust process by reconstructing its potential landscape. For this aim we utilised the Kramers–Moyal equation which generalises the Fokker–Planck equation in the sense that higher-order Kramers–Moyal coefficients can be related to discontinuities in the stochastic processes.

In a first step, a standard one-dimensional Kramers–Moyal analysis revealed a monostable potential for the isolated $\delta^{18}$O record and a bistable one for the dust. This finding calls the prevailing understanding that the Greenland ice core $\delta^{18}$O record

stem from bistable dynamics into question. The qualitative difference between the reconstructed potentails is remarkable given the high co-variability of the two time series and their synchronous two-regime character, which dominates not only the dust but also the $\delta^{18}$O record. Moreover, we found non-vanishing higher-order Kramers–Moyal coefficients for $\delta^{18}$O, indicating the presence of discontinuities in the record, assuming the process is truly Markovian. This renders the Langevin equation unsuited to fully describe the underlying process and requires the addition of jumps. In contrast, according to our analysis, the isolated

dust record is a continuous process that is described well by the Langevin equation.

In a second step, we expanded our analysis to a two-dimensional setting that takes into account possible couplings between the $\delta^{18}$O and dust time series. Our two-dimensional examination of the conditioned potential landscapes confirms our initial finding of a mono-stable potential for the $\delta^{18}$O, wherein the minimum's position is controlled by the value of the dust. The dust variable, on the other hand, seems to undergo a *double-fold bifurcation* parametrised by the $\delta^{18}$O, where we can observe

the change from a single (stable) fixed point to three fixed points (two stable, one unstable), and again to a single (stable) fixed point, from small to large values of $\delta^{18}$O. Our analysis reveals two convergent regions in the $\delta^{18}$O–dust state space in agreement with the two-regime nature of the coupled record. Importantly, our findings question the prevailing interpretation of the isolated $\delta^{18}$O record as the direct signature of an intrinsically bistable process. Regarding $\delta^{18}$O as a direct measure of the local temperature, it seems plausible that not the temperature itself is bistable but rather that the bistability is enshrined in

another climate variable that drives Greenland temperatures. The apparent two-regime nature of the $\delta^{18}$O record would thus only be inherited from the actual bistability of other processes. This may be the atmospheric circulation as represented by the dust proxy, or another external driver whose signature might be encoded in the higher-order KM coefficients of the $\delta^{18}$O.

Many physical mechanisms have been proposed as candidates for explaining the DO events, with most of them building on the proposed bistability of the Atlantic Meridional Overturning Circulation (AMOC), e.g., Refs. (Ganopolski and Rahmstorf,





2001; Clark et al., 2002; Vettoretti and Peltier, 2018; Lohmann et al., 2021). While some studies argue that DO cycles are the signature of self-sustained oscillations within the coupled sea-ice ocean system (Boers et al., 2018; Vettoretti and Peltier, 2018; Menviel et al., 2020), others advocate for an active role of the atmosphere or even ice sheets in the initialisation of DO events, e.g., Refs. (Kleppin et al., 2015; Zhang et al., 2014; Gottwald, 2020). The proposed self-sustained oscillation mechanism is not contradicted by our investigation and neither is a stochastic trigger embedded in the sea ice. Our results do also not contradict

an atmospheric trigger for the DO events – we see that if the dust switches from one state to the other this will in turn shift the level of the $\delta^{18}$O. Yet, the atmospheric transitions would in this picture not be induced by rare extreme events as proposed by Ditlevsen (Ditlevsen, 1999), but rather by a regular additive Gaussian noise. However, as mentioned previously, the absence of discontinuities in the dust record may also be a question of the temporal data resolution.

Certainly, DO events and their global expression feature a complex interplay of the AMOC, the North Atlantic and Nordic

Sea's sea ice cover, the polar jet stream and probably more climatic subsystems such as ice sheets or the East Asian Monsoon system (Cheng et al., 2013). Our analysis considered only a two-dimensional projection of the very high-dimensional dynamics and can therefore not be expected to deliver all details of the triggering mechanism. Neither the ocean-focused self-sustained oscillation hypothesis, nor the idea that the atmosphere acts as a trigger, can be ruled out based on our findings. Nevertheless, our results challenge prevailing assumptions, e.g., regarding the bistability and the smoothness of the temperature proxy record

and adds valuable information that may help further constrain physical hypotheses to explain the DO events in the future. Analysis structurally similar to this one should be applied to other pairs of Greenland proxies to investigate the corresponding two-dimensional drift. Finally, our study underlines the need for higher resolution data, as the scarcity of data points is a limiting factor for the quality of non-parametric estimate of the KM coefficients.

*Code availability.* The code used for this study will be made available by the authors upon request.

*Data availability.* The original measurements of $\delta^{18}$O and dust concentrations go back to (North Greenland Ice Core Projects members, 2004) and (Ruth et al., 2003), respectively. The 5 cm resolution data together with corresponding GICC05 ages used for this study were first published as a Supplement to (Gkinis et al., 2014) and can be downloaded from www.iceandclimate.nbi.ku.dk/data/NGRIP_d18O_and_dust_5cm.xls (last access: 18. November 2021). The reconstruction of global average surface temperatures is available as a Supplement to (Snyder, 2016) under https://static-content.springer.com/esm/art%3A10.1038%2Fnature19798/MediaObjects/41586_2016_BFnature19798_MOESM258_ESM.xlsx

(last access: 18. November 2021).

## Appendix A: Data detrending

As mentioned in Sec. 2, this study focuses on the period 59–27 kyr b2k. Detrending of the data is needed to ensure that the time series can be considered stationary processes, which is an underlying assumption for the Kramer–Moyal analysis performed in our investigation. To compensate for the influence of the background climate on the climate proxy records of dust and $\delta^{18}$O, we



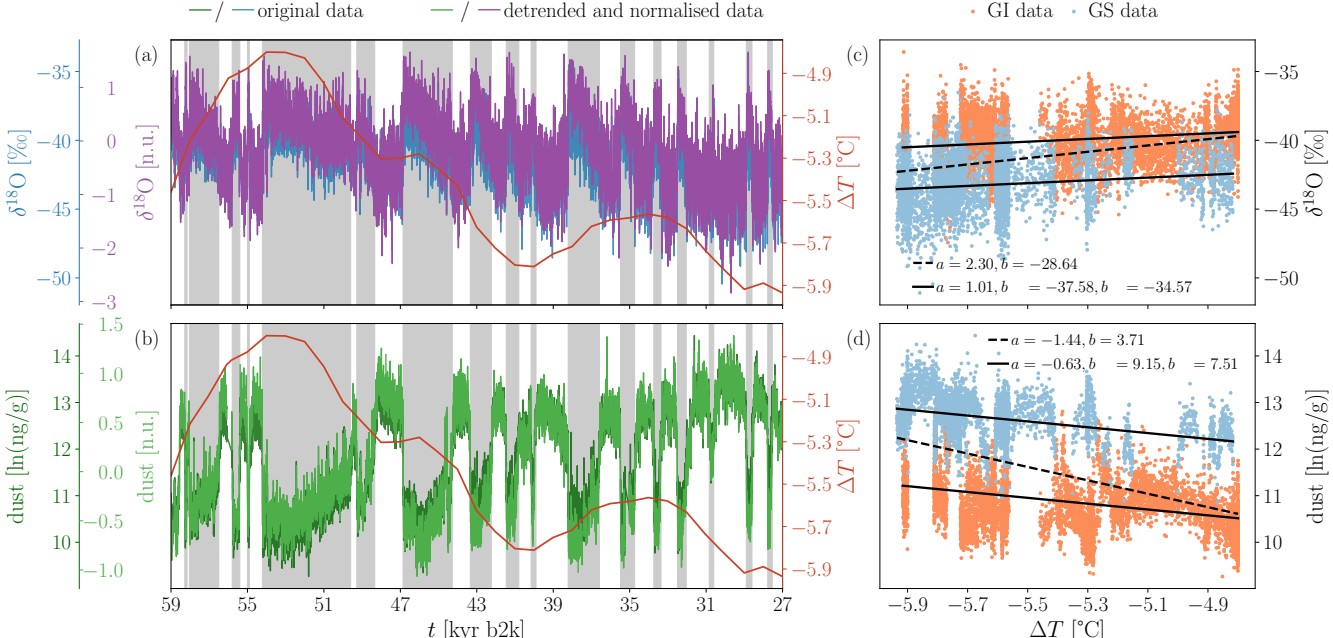

**Figure A1.** Removal of a linear trend in the NGRIP $\delta^{18}$O and dust time series (North Greenland Ice Core Projects members, 2004) with respect to a global average surface temperature reconstruction (Snyder, 2016). In panel (a) both original $\delta^{18}$O (blue) as well as detrended and normalised (purple) are shown. Idem for the dust record in panel (b) (dark green and light green, respectively). The background temperature given in anomalies to present day climate is shown in both aforementioned panels (red). Panels (c) and (d) show a scatter plot the original $\delta^{18}$O and dust data with respect to temporarily corresponding temperature anomalies, respectively. Data from Interstadials (Stadials) is shown in orange (light blue). The black dashed line results from a simple linear fit to the entire data, while the continuous black lines correspond to the fitting scheme that uses a single slope but two different offsets to separately fit the Stadial and Interstadial data.

remove a linear drift with respect to reconstructed global average surface temperatures (Snyder, 2016) from both time series.
Fig. A1 illustrates the detrending scheme for both time series. Due to the two regime nature of the time series, a simple linear regression overestimates the temperature dependencies (see Fig. A1(c) and (d), dashed line). Instead, we separate the data from Greenland Stadials and Greenland Interstadials and then minimise the expression

$$\left( \sum_{i=1}^{N} \left( \delta^{18}O(t_i) - a\Delta T(t_i) - \begin{cases} b_{\mathrm{GI}}, \text{ if } t_i \in \mathrm{GI} \\ b_{\mathrm{GS}}, \text{ if } t_i \in \mathrm{GS} \end{cases} \right)^2 \right)^{1/2}, \tag{A1}$$

with respect to the parameters $a$, $b_{\mathrm{GI}}$, and $b_{\mathrm{GS}}$ (correspondingly for dust). $t_i \in \mathrm{GS}$ (GI) indicates that a given time $t_i$ falls into a Stadial (Interstadial) period. The index $i$ runs over all data points and $N$ denotes the total number of points. The resulting $a$ is used to detrend the original data with respect to the temperature. The detrended data is then normalised by subtraction of its mean and division by the difference between the mean Stadial and mean Interstadial values.



## Appendix B: Markov property of the data

The Markov property of the data, central in this work, is a necessary property whilst designing Markovian stochastic models
to describe any paleo-climatic data, as the name suggests. These include the most commonly used stochastic models, e.g.,
Langevin processes, based on having independent increments of the data (Risken and Frank, 1996; Friedrich et al., 2011). This
is best understood by examining the Chapman–Kolmogorov equation of the joint probability densities $p_{i_1,\ldots,i_n}(x_1,\ldots,x_n)$,
with $\{x_i\}$ a collection of random variables ordered with $t_1 < \cdots < t_n$

$$p_{t_1,\ldots,t_n}(x_1,\ldots,x_n) =$$
$$p_{t_1}(x_1)p_{t_2;t_1}(x_2 \mid x_1)\cdots p_{t_n;t_{n-1}}(x_n \mid x_{n-1}), \tag{B1}$$


where the Markov property here allows us to separate each joint probability density as being solely a function of two adjacent
segments. One of the most straightforward ways of evaluating the Markov property for data is to examine the auto-correlation
function of the increments of the data at the shortest incremental distance. That is, take the data $x_t$, construct the differences
$\Delta x_t = x_{t+1} - x_t$, and obtain the auto-correlation function $\rho(\tau)$

$$\rho(\tau) = \frac{\mathrm{E}\left[(\Delta x_t - \mu_t)(\Delta x_{t+\tau} - \mu_{t+\tau})\right]}{\sigma_t \sigma_{t+\tau}}, \tag{B2}$$

here $\mu$ is the mean and $\sigma^2$ the variance of $\Delta x_t$. In Fig. B1 we display the auto-correlation functions $\rho(\tau)$ of the two time series,
which is only lightly anti-correlated at the shortest lag of $\tau = 5y$.

Here, we include a small note of caution for the interested reader. Pre-processing paleo-climatic data is usually implemented
to reduce the noise or remove short or long term trends. A common method to remove long-term trends in the records is to
apply a low-pass filter. This will invariantly lead to spurious correlations, thus it should be considered with care, dependent on
the data analysis techniques employed. In our case, this would be disastrous. Low-pass filtering the data would create spurious
correlations in the incremental time series and Markovianity would be lost.

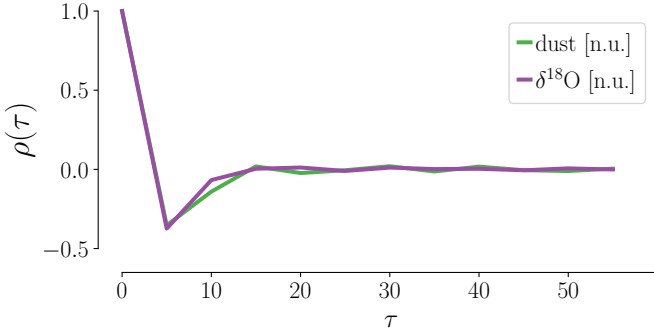

**Figure B1.** Autocorrelation $\rho(\tau)$ of the increments $\Delta x_t$ of $\delta^{18}$O and dust records. Both records show a weak anti-correlation at the shortest
lag $\tau = 5y$, and no correlation for $\tau > 5y$. We thus consider the data Markovian.





**Appendix C: Nadaraya–Watson estimator of the Kramers–Moyal coefficients and bandwidth selection**

In order to carry out the estimation in Eq. 7 we map each data point in the corresponding state space to a kernel density and
then take a weighted average over all data points

$$
\begin{aligned}
D_m(x) &\sim \frac{1}{m!}\frac{1}{\Delta t}\langle (x(t+\Delta t)-x(t))^m | x(t)=x\rangle \\
&\sim \frac{1}{m!}\frac{1}{\Delta t}\frac{1}{N}\sum_{i=1}^{N-1} K(x-x_i)(x_{i+1}-x_i)^m.
\end{aligned}
\tag{C1}
$$

Alike selecting the number of bins in a histogram, when employing kernel-density estimation with an Nadaraya–Watson
estimator for the Kramers–Moyal coefficients $D_m(x)$, one needs to select both a kernel and a bandwidth (Nadaraya, 1964;
Watson, 1964; Lamouroux and Lehnertz, 2009). Firstly, the choice of the kernel is the choice of a function $K(x)$ for the
estimator $\widehat{f}_h(x)$, where $h$ is the bandwidth at a point $x$

$$
\widehat{f}_h(x) = \frac{1}{nh}\sum_{i=1}^{n} K\left(\frac{x-x_i}{h}\right).
\tag{C2}
$$

for a collection $\{x_i\}$ of $n$ random variables. The kernel $K(x)$ is such that $K(x)=1/hK(x/h)$ and is normalisable $\int_{-\infty}^{\infty} K(x)\mathrm{d}x = 1$ (Tabar, 2019). The bandwidth is equivalent to the selection of the number of bins, except that binning in a histogram is always
"placing numbers into non-overlapping boxes". The optimal kernel is the commonly denoted Epanechnikov kernel (Epanech-
nikov, 1967), but Gaussian kernels can be used as well. These nevertheless require a compact support in $(-\infty,\infty)$, thus on a
computer they require some sort of truncation (even if Fourier space, as the Gaussian shape remains unchanged).

The selection of an appropriate bandwidth $h$ can be aided – unlike the selection of the number of bins – by the Silverman's
rule-of-thumb (Silverman, 1998), given by

$$
h_{\mathrm{S}} = \left(\frac{4\hat{\sigma}^5}{3n}\right)^{\frac{1}{5}},
\tag{C3}
$$

where again $\sigma^2$ is the variance of the time series. In Fig. C1 three different bandwidths are used to evaluated the various KM
coefficient, as given in Fig. 4 The bandwidths are the optimal bandwidth given by the Silverman's rule-of-thumb $h_{\mathrm{S}}$, three
times $h_{\mathrm{S}}$, and one-third $h_{\mathrm{S}}$.

Note that in neither of the examples with different bandwidths we notice a change of the potential shape of the various
records. The mono-stability of the potential $V(x)$ of $\delta^{18}$O is persistent, as is the bi-stability of the potential $V(x)$ of the dust
concentration.

**Appendix D: Second-order correction to the Fokker–Planck/Kramers–Moyal operator**

In order to correctly retrieve from data the Kramers–Moyal coefficients, we need to evaluate the operation in the Fokker–Planck
equation Eq. (3). In fact, we showed that this equation is not sufficient to describe the fast transitions in the $\delta^{18}$O record. Let





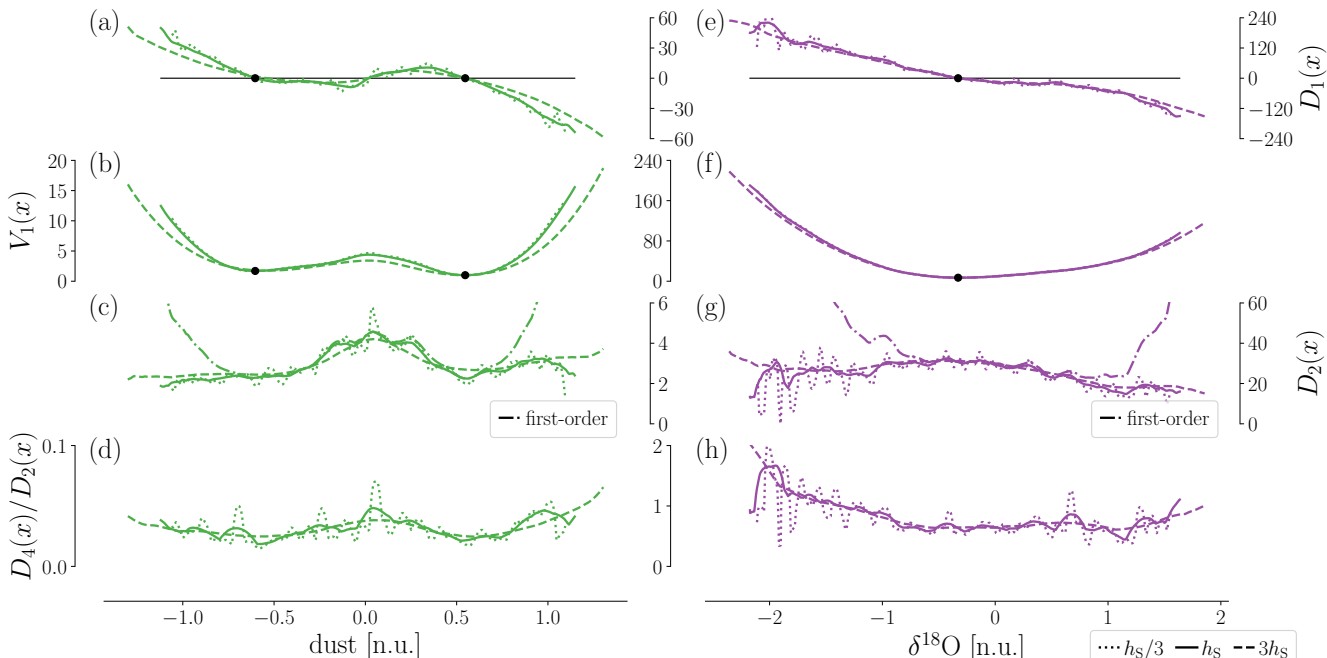

**Figure C1.** The effect of the bandwidth selection $h_S$ on the KM estimations, in identical fashion to Fig. 2. The non-parametric estimates of the first KM coefficient $D_1(x)$, the associated potential landscape $V(x)$, the second KM coefficient $D_2(x)$, and the ratio of the fourth to the second KM coefficient $D_4(x)/D_2(x)$. Left column for dust, right column for $\delta^{18}$O. Three bandwidths used for the Nadaraya–Watson kernel-density estimator: the optimal Silverman's rule-of-thumb $h_S$, three times $h_S$, and one-third $h_S$. The Nadaraya–Watson kernel-density estimator's bandwidths $h_S$ for $\delta^{18}$O is 0.131 and for dust 0.103. In all cases, the interpretation of the estimator remains the same: bi-stability in the dust, mono-stability in the $\delta^{18}$O. In (c) and (g) are included the first-order estimator for the second KM coefficient $D_2(x)$, i.e. without corrective terms, discussed in App. D.

us nevertheless focus on this equation for the moment, and rewrite it in a more formal manner as an operator

$$\frac{\partial}{\partial t}p(x,t+\tau|x',t) = \frac{\partial}{\partial x}D_1(x)p(x,t+\tau|x',t)$$
$$+ \frac{\partial^2}{\partial x^2}D_2(x)p(x,t+\tau|x',t)$$
$$= \mathcal{L}_{FP}\ p(x,t+\tau|x',t), \tag{D1}$$

with $\mathcal{L}_{FP}$ the formal Fokker–Planck operator and

$$D_m(x) = \frac{1}{m!}\lim_{\tau\to\infty}\frac{M_m(x,\tau)}{\tau}, \tag{D2}$$

where $M_m(x,\tau)$ is the $m$-order conditional moment, i.e.

$$M_m(x,\tau) = \int_{-\infty}^{\infty}(x'-x)^m p(x',t+\tau|x,t)\,\mathrm{d}x'. \tag{D3}$$





which we introduced in Eq. (7) in a similar notation. If we limit $m \leq 2$, we are truly talking about a Langevin process described by the Fokker–Planck equation, with $\mathcal{L}_{\mathrm{FP}}$ the formal Fokker–Planck operator. We also saw that we can generalise the problem and not truncate the terms at second order, thus including an infinite series of conditional moment $M_m(x, \tau)$ would give rise

to the Kramers–Moyal equation and the Kramers–Moyal operator $\mathcal{L}_{\mathrm{KM}}$. The subsequent second-order correction is showcased here for the Fokker–Planck equation, based on Ref. (Gottschall and Peinke, 2008; Rydin Gorjão et al., 2021). For sake of coherence, we utilise here the second-order corrections to show that the second Kramers–Moyal coefficient – the diffusion strength – can be seen as constant, i.e. not state depended.

In order to solve Eq. (3), one takes the formal step considering an initial conditions $\delta(x-x')$ as a starting point and employing
the exponential representation of the operator, which we can decompose it into a power series as

$$
\begin{aligned}
p(x, t+\tau | x', t) &= \exp\left(\tau \mathcal{L}_{\mathrm{FP}}\right) \delta(x - x') \\
&= \sum_{k=0}^{\infty} \frac{(\tau \mathcal{L}_{\mathrm{KM}})^k}{k!} \delta(x - x').
\end{aligned}
\tag{D4}
$$

From here we consider the first-order and second-order approximation, i.e. truncation of the operator, as

$$
\exp\left(\tau \mathcal{L}_{\mathrm{FP}}\right) \sim 1 + \tau \mathcal{L}_{\mathrm{FP}} + \frac{\tau^2}{2} \mathcal{L}_{\mathrm{FP}} \mathcal{L}_{\mathrm{FP}} + \mathcal{O}(\tau^3).
\tag{D5}
$$

Considering only the first-order, $\sim \tau$, we recover the well-known relation between the conditional moments and the Kramers–Moyal coefficients, given by

$$
D_m(x) = \lim_{\tau \to 0} \frac{M_m(x, \tau)}{(m!)\tau}.
\tag{D6}
$$

If we now include the second-order approximation, i.e. we consider terms up to $\sim \tau^2$, we obtain a corrective term for the second Kramers–Moyal coefficient

$$
\begin{aligned}
D_1(x) &= \lim_{\tau \to 0} \frac{1}{\tau} M_1(x, \tau), \\
D_2(x) &= \lim_{\tau \to 0} \frac{1}{2\tau} \left( M_2(x, \tau) - M_1(x, \tau)^2 \right).
\end{aligned}
$$


We employ this correction to our examination to show that the diffusion coefficient, i.e. the amplitude of the fluctuations, is constant in space. In Fig. C1(c) and (g) we display both the first-order and the corrected, second-order diffusion coefficient. In this we can see that utilising solely the first-order correction could lead us to erroneously consider the diffusion term as state dependent (having a parabolic shape), suggesting a multiplicative noise. By implementing the second-order corrective terms a
considerable improvement of the estimation is achieved, to what we judge to be simple additive (not state dependent) noise in the records.

*Author contributions.* All authors contributed equally to the design of the paper. L.R.G. and K.R. conducted the analysis.





*Competing interests.* No competing interests

*Acknowledgements.* LRG and DW gratefully acknowledge support from the Helmholtz Association via the grant *Uncertainty Quantification*
*– From Data to Reliable Knowledge (UQ)* with grant agreement no. ZT-I-0029. This work was performed by LRG as part of the Helmholtz
School for Data Science in Life, Earth and Energy (HDS-LEE). NB acknowledges funding by the Volkswagen foundation. This is TiPES
contribution #XX; the Tipping Points in the Earth System (TiPES) project has received funding from the European Union's Horizon 2020
research and innovation programme under grant agreement no. 820970.



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
