# Peer review of "Stable stadial and interstadial states of the last glacial's climate identified in a combined stable water isotope and dust record from Greenland"

_Earth System Dynamics, 2021_

## Referee Comment (RC2)

[referee-annotated manuscript omitted]

---

## Author Comment (AC1)

Answers: green Comments by reviewer: violet

Comment on esd-2021-95

Peter Ditlevsen (Referee)

Referee comment on "Changes in stability and jumps in Dansgaard–Oeschger events: a data analysis aided by the Kramers–Moyal equation" by Leonardo Rydin Gorjão et al., Earth Syst. Dynam. Discuss., https://doi.org/10.5194/esd-2021-95-RC1, 2022

First of all, we would like to thank the referee for his careful and constructive review and the overall positive feedback. We will in fact adopt most of his suggestions and are convinced that this will improve the manuscript substantially. The changes that we will make to our manuscript will become clear from our point-by-point answers to the referee's comments below.

In this paper two paleoclimatic ice core records are analyzed. These, the water isotope, d18O, and the dust concentration records are analyzed for the period 59-27 kyr BP, which is the glacial period dominated by regular occurrences of Dansgaard-Oeschger events. There are two major points in the paper: Firstly, the data are modelled as a stochastic process using the Kramers-Moyal equation to investigate the importance of (discontinuous) jumps in the noise. Secondly, the two records are modelled as a two-dimensional joined process.

It seems to me that the two points are only loosely related, and the authors could consider presenting them in two separate papers.

We thank the reviewer for the suggestion. We felt that the results presented – which as pointed out, could be separated into two manuscripts – still warranted a single manuscript, for the following reasons:

- 1. A 1D approach is a natural starting point for our investigation, from which our analysis unveils various inconsistencies motivating an investigation in a 2D setting.
- 2. The methodology is the same for both analyses. We easily could imagine a referee asking for the 2D investigation if we presented the 1D analysis exclusively.

In a strict sense, however, we do not provide full models. We had already considered including an explicit stochastic model for the discontinuous phenomena, but felt it was premature to include this. If the referee and editor still feel that our results would be better presented in two papers, we would of course be glad to consider this further, e.g. in terms of a Part I and Part II?

I enjoyed reading the paper and find it publishable. However, there are a few issues below calling for revisions before publication. I have two major concerns regarding the two parts, and some minor points: As to the first point, I have not seen the Kramers-Moyal (KM) equation applied to these data before, so this is a novel approach. The equation, for which the Taylor expansion of the conditional probability density function is taken to higher order than two, covers the case where the noise term in the governing Langevin equation is not gaussian, but contains jumps. It is stated that in the case of Levy processes the Fokker-Planck equation does not apply.

Actually, for the most relevant class of Levy processes, the alpha-stable Levy processes an extension of the Fokker-Planck equation based on the characteristic function of the

alpha-stable process exists (see: Samorodnitsky and Taqqu (1994) 'Stable non-gaussian processes, Chapman and Hall, NY. or Ditlevsen, PRE, 60, 172-179). The challenge in applying the KM equation is the estimate of the higher order coefficients (eq. 6) for the data series: Since the higher order terms are (increasingly) dominated by the extremes in the increments the finite time series very quickly "dilutes". A main result (eq. 9 and figure 3) includes the sixth moment of the increments. I thus miss an analysis of uncertainties and reliability in these estimates. I find that this is essential for publishing this (nice!) result.

We thank the reviewer for this comment, it is a clear oversight on our part to state that one *cannot* describe Lévy-driven processes (or Lévy-noise driven Langevin equations) with a Fokker–Planck equation. We will amend the related statements in the manuscripts and point to the correct references wherein this is discussed. Similarly, we will include an explicit formulation to flash out what exactly is meant by discontinuity in our context, as also requested by the other referee in his review.

In order to estimate the uncertainties in the Q-ratio estimates in Eq. 9 and Fig. 3, in the revised paper we will introduce a metric for estimating the uncertainty of estimating higher-order moments in the conditional moments needed to estimate Q. Since we do not propose any explicit stochastic model for the discontinuous contributions in  $\delta^{18}$ O, we will analyse in place the uncertainty of Langevin processes with identical drift and diffusion as those estimated for the dust and  $\delta^{18}$ O proxies, and will show that the Q-ratio of the discontinuous  $\delta^{18}$ O behaves substantially different from a conventional continuous Langevin processe.

Another point which could be given a little more attention is the fact (as also correctly stated) that the strong time-asymmetry in the data (the sawtooth shape) cannot be captured by the model. How does this influence the relevance in including higher order terms (higher than second order) in eq. 5?

We thank the reviewer for the remark. In some sense our choice of showing first the two separate one-dimensional analyses, alongside with precisely the aforementioned observation of the sawtooth shape, is to point out time-asymmetry cannot be captured in a *one-dimensional* setting under a Langevin-equation. However, the statement we make in the manuscript

*I.233* Note that the model equations employed here are by construction symmetric with respect to time, therefore, as it is, the model cannot reproduce the temporal asymmetry that is visually suggested in the dust record.

is not precise, in the sense that only after having estimated the drift and the constant diffusion and the absent 4th-order KM coefficient, the KM results for the dust are inconsistent with the apparent time asymmetry – this inconsistency is thus not by construction. This is also exactly the point the referee points to: a time asymmetric stochastic process is very likely to exhibit non-zero 4th-order KM coefficients. On the contrary, designing a time asymmetric stochastic process with drift and diffusion exclusively is far more difficult in a purely autonomous setting – yet not impossible. In short: the time asymmetry of the data is an indicator that one should include higher-order terms in the KM expansion.

In order to give an example: we can achieve time asymmetry very simply by considering a process like a Langevin process, just augmented with a discontinuous trajectory. A simple example would be

dx(t) = -a(x)dt + bdW(t) + cdJ(t)

where J(t) is a Poisson process with a jump rate  $\lambda$ >0. In this simplest of formulations, if there is at least 1 jump from the Poisson jump process, the process becomes time-asymmetric. (From an applied point of view, there are some considerations to be respected regarding the relation of the amplitudes of a(x), b, and c.)

In light of the construction of our paper – which deliberately avoids writing down specific stochastic processes as the one given above – we shun from including this, but included the higher-order terms from the Kramers–Moyal equation, which point at the existence of such discontinuities in the ice-core time series.

We note that the strong time-asymmetry can – and is – captured in a two-dimensional setting, just as we show in our two-dimensional analysis.

As to the second point, the major results are presented in figure 4. Obviously, when considering a one-dimensional record, the drift can always be seen as a result of a potential. This is not the case in two - and higher dimensions, where gradient drift is a non-generic case. I'm sure that the authors are aware of this, the drift is a two-dimensional flow field, as also shown in the small inserts in the subplots of figure 4. I find the construct of pseudo-one-dimensional potentials ( $V(x_1|x_2)$ ) both confusing and useless. I suggest that the authors consider abandoning this all together (as well as the notion of a potential landscape). The interpretation in figure 4(c) of a double fold bifurcation is obscure, and -I believe- wrong.

We thank the reviewer for the comment. We ourselves have struggled with the "pseudo one-dimensional" potentials. It is clear to us that if the reviewer finds them unhelpful, then our doubts about their usability are confirmed. We will remove them in the revised version and directly showcase instead the two-dimensional drifts as quiver plots, which we hope are clearer.

The referee's comment on Figure 4(c) (together with a similar comment by the other referee) clearly shows that we have not conveyed our thoughts properly and therefore we will improve the manuscript as follows:

We will revise the discussion of Figure 4 to make our conclusions more precise. After reviewers' feedback we still keep the interpretation of Fig. 4(c) that it shows a double-fold bifurcation if one treats dust as the dynamical variable and  $\delta^{18}$ O as control variable. Since we will not use the notion of conditioned potentials, we will explain this double-fold bifurcation in terms of the nullclines of the dust drift, conditioned on the  $\delta^{18}$ O. The definition of a bifurcation always depends on what is the variable and what is the parameter. For example, the standard form of a fold bifurcation is given by  $x^2 = a$ , where *x* bifurcates when the control parameter *a* crosses *0*. Obviously, no bifurcation occurs if we reverse the roles of *a* and *x*.

There remains of course the possibility that the double-fold structure that we observe is spurious and simply an artefact arising due to the scarcity of data (especially in the region of the state space where we observe the saddle-node bifurcations). We pursued two approaches to rule this out.

First: we have tried to adapt the data analysis as suggested by the reviewer (see our next reply). We applied PCA to obtain a new 'rotated' basis ( $p_1$ ,  $p_2$ ) and projected the data onto that new basis. Subsequently we applied the same method to the rotated data as we did to the original data. In this case we find one variable,  $p_1$ , which appears monostable and mostly independent of the other variable  $p_2$ . In contrast, the dynamics of the other variable  $p_2$

strongly depends on  $p_1$ . Hence, the two processes do not decouple as hypothesised by the reviewer. Now, if we take  $p_2$  as the dynamic variable and  $p_1$  as a control parameter, we do no longer see an archetypal double-fold bifurcation as correctly remarked by the reviewer. However, the prime dynamic fingerprints are still there! The nullcline has the shape of a lying S (see Figure below) and if you rotate it back you rediscover the double-fold bifurcation.

Second: we generated synthetic data in a truly decoupled setting, with a double well potential in one direction and a single well in the other. Then we rotated the synthetic data, introducing a coupling between them, and again applied the estimation of the KM coefficients. The results (see Figure below) truly differ from our Fig. 4(c).

Together, these findings make us confident that the interpretation of Fig. 4c as a double-fold bifurcation in the dust (conditioned on  $\delta^{18}$ O) is meaningful.

Taking this discussion further, the interpretation of the results depends very much on the interpretation of the variables. After all,  $\delta^{18}$ O and dust are only two observables and  $\delta^{18}$ O is only a proxy for the temperature (and dust an even more uncertain proxy for atmospheric circulation). Hence, it might be more appropriate to view them as indicator variables of more fundamental, unobserved atmospheric variables.

In conclusion, it is of course impossible to reach a definite answer on the entire physical mechanism leading to the observed bimodality from our approach. Still, we can derive some very interesting conclusions. In particular, the hypothesis that the bistability is rooted in the temperature, which then drives the dust variable as a dependent variable, is highly unlikely given our results. Furthermore, it should be further investigated if a bistability in circulation patterns exists, which can then drive a bimodality of the Greenland temperatures. In the revised version, we will sharpen the discussion section to make the interpretation clearer.

What I find interesting is the scatter plot in 4(a), which nicely explain the results in figure 2, namely that the stationary distribution for the dust is bimodal while it is unimodal for the d18O: This corresponds to the marginal distributions in 4(a) (projections onto the axis.

The authors could consider analyzing a rotation (linear combination of the two variables) of the data along an axis connecting the two maxima (GS and GI) and a perpendicular direction. In this way one would obtain a "clean" two state dynamics and a "clean" one state perpendicular dynamics. First thing would be to check for independence. Just a suggestion.

We thank the reviewer for the suggestion. We have considered a rotation to a new set of axes given by  $v_1 = [-0.80, -0.59]$  and  $v_2 = [0.80, -0.59]$  (obtained via PCA), in relation to our frame of reference, which are orthogonal. Note that also as suggested, we are using the negative of the logarithm of the dust.

From this we draw two new "projection" time series,  $p_1$  and  $p_2$ , that have a Pearson correlation  $\rho(p_1, p_2) = 0.01$ . We can similarly draw the drifts in a two--dimensional setting from these time series, as seen below:

---

## Author Comment (AC2)

Answers: green
Comments by reviewer: violet

Comment on esd-2021-95

Tamás Bódai (Referee)

Referee comment on "Changes in stability and jumps in Dansgaard–Oeschger events: a data analysis aided by the Kramers–Moyal equation" by Leonardo Rydin Gorjão et al., Earth Syst. Dynam. Discuss., https://doi.org/10.5194/esd-2021-95-RC2, 2022

First of all, we would like to thank the referee for his thorough and detailed review of our submitted manuscript. His scrutiny from an outside perspective revealed that the manuscript is not as clear as it should be about some central points regarding its conceptual design. We believe that we can in this answer clarify our approach and clear up some misunderstandings. This review gives us the chance to substantially improve the manuscript's clarity with respect to the conceptual design, its explanatory power and its limitations.

The paper "Changes in stability and jumps in Dansgaard–Oeschger events: a data analysis aided by the Kramers–Moyal equation" analyses d18O and dust data from a Greenland ice core in order to gain further understanding of the famous Dansgaard-Oeschger (DO) events. They preprocess the time series with the aim of establishing a stationary stochastic process. They estimate Kramers-Moyal (KM) coefficients, which could possibly reveal jumps in the process, outside the framework of the Fokker-Planck equation, corresponding to what can be naively seen as regime transitions. They explore the added value of joint fitting of the d18O and dust data over treating them separately.

The referee correctly summarised the approach of our manuscript. There are two minor points we would like to clarify.

First, we do not claim that the nonzero 4th-order KM coefficient – which we interpret as evidence for a jump-like stochastic forcing on the d18o – fully explains the regime switches. We only say that the jumps in the stochastic forcing might play a role in the regime switches and should not be discarded in the analysis.

Second, the term 'fitting' usually describes a method where a model output is compared to the data. Then the model parameters are tuned such that the model would optimally approximate the data. The approach pursued in our study is different. We estimate the Kramers-Moyal coefficients directly from the data and no model-output to data comparison is required for this method; in particular, we do not minimise a distance or cost function. We deliberately refrain from presenting explicit stochastic model equations.

I'm not very convinced that the applied methodology is suitable. As far as i see, the authors do not test their null-hypothesis (H0) of a stationary process.

We fully agree with the referee that one cannot assume per se that the investigated time series are stationary. Also, the referee is correct in the sense that we did not provide a statistical test that supports the stationarity of the investigated time series. In the revised version of the manuscript we will employ a slightly different detrending of the data and provide tests that do support stationarity.

Since the referee's general suspicion towards the applicability of the chosen method is the key criticism in his review, we will in the following give a detailed explanation why we consider our approach meaningful.

We believe that a comment the referee made in the pdf attached to his report, is helpful to understand the exact point of criticism he raised with respect to the stationarity assumption:

In line 401, we write:

> *'This may be the atmospheric circulation as represented by the dust proxy, or another external driver whose signature might be encoded in the higher-order KM coefficients of the δ18O.'*

which was commented by the referee with the words:

> This sounds like there is no problem with the methodology if that's the case.

This comment led us to the interpretation that the referee does not question that the climatic process giving rise to DO events can be considered stationary over the investigated time period. Instead, we understand the referee's point as follows:

The observed data is a projection of a high dimensional complex process onto the state space spanned by $\delta^{18}O$ and dust which are assumed to represent Greenland temperatures and atmospheric large-scale circulation. The applied methodology now assumes that all other degrees of freedom (or all other variables, termed 'bath') can be subsumed in an effective force and a stochastic force (i.e., noise), and be described in a SDE approach. This subsumption certainly relies on the type of interaction of the observed variables with the bath variables and requires a separation of time scales.

We believe that the referee doubts whether the relevant dynamics that gives rise to the observed DO variability in the data at hand are fully captured in the projection onto the observed, low-dimensional subspace. We interpret his objection to our stationarity assumption in the sense that the referee advocates for the presence of unobserved or hidden variables that cannot be subsumed in the bath treatment. Such a potential coupling to hidden variables can also be interpreted as non-stationarities of the observed dynamics.

For sake of clarity: In the following we will refer to unobserved variables that cannot be described as a bath by denoting them as *hidden variables*.

If the requirements for the eff. force + noise description of the unobserved variables are not fulfilled and one still tries to impose this framework – as we do – then the hidden variables which in fact have much more explicit impact on the observed variables' dynamics, still contribute to the KM coefficients and in particular to higher order KM coefficients. One would then in most cases observe that the model retrieved from the data does not fully explain the dynamics of the observed variables.

At this stage, we can make three important remarks:

1. The observations are limited to the d18o-dust space, so all we can do is try to investigate these. We cannot include further variables in our analysis, simply because there is no data.

2. Given (1) and in line with finding the simplest starting argument, it is natural to make the attempt to treat the unobserved variables as a bath and then scrutinise the consistency of the obtained KM coefficients with the observations. There will typically be some characteristics of the dynamics which are reasonably well explained by this approach, and others which are not. This is exactly the case in our study.

   For example: In our 1D analysis of the $\delta^{18}O$ we emphasise the inconsistencies of the obtained KM coefficients with the data. It is these inconsistencies that motivate us to

explore the next complicated approach of analysis which is the investigation of the coupled dust-d18o dynamics.

*'At first sight, the monostability of the reconstructed $\delta^{18}O$ potential contradicts the apparent two regime nature of the time series. There are two possible explanations for this discrepancy: First, regime switching of monostable stochastic process can be achieved through complex noise structures (e.g., Lévy-like noise, generalised Fokker–Planck equations, or fractal motions) (Chechkin et al., 2003, 2004; Metzler and Klafter, 2004). Secondly, a similar effect can be obtained in a two-dimensional setting if the dynamics of one dynamical variable explicitly depends on the other, which would be impossible to judge from the one-dimensional analysis presented so far. Thus, within the limits of this analysis – that is assuming that the process is Markovian and stationary and that the system under study is fully represented by dust and $\delta 18 O$ (no coupling to further hidden variables) – the source of the regime switching must either be endowed by more complex noise processes or by the coupling between the dust and the $\delta18O$ systems.' (l.243)*

3. If there would be a strong coupling to hidden variables, we would expect the data to show higher degrees of autocorrelation, since typically subsuming hidden variables mistakenly in a bath gives rise to a memory term. This essentially follows from the Mori-Zwanzig formalism.

In the 2D setting, we limit the KM analysis to the first and second-order coefficients, due to the scarcity of data. In our investigation we then focuses on the retrieved deterministic flow field and we do not claim to provide a comprehensive explanation for the dynamics of the coupled dust-$\delta^{18}O$ system. What we do find, however, is that this flow field does not, in itself, explain the fast transitions c2w and the slow w2c transitions – here, importantly, we point to the possibility that hidden variables may play a decisive role at various points in the manuscript.

*'The effective vector field of motion does not indicate a clear path that the system would take in order to transition between stadial and interstadial states. This leaves open the possibility that transitions between stadial and interstadial states are mainly induced by noise as argued by, e.g., Ref. (Ditlevsen et al., 2007) (i.e. noise-induced tipping), facilitated by a shallow potential barrier close to the minima of the (effective) vector field.' (l.321)*

(Which was commented by the referee as follows:

I'm really not convinced. I think the consideration of an important variable is missing.

*'Evidently, our analysis is limited to solely two climate proxy variables and the stability of the one can only be assessed conditioned on the other, leaving aside potential coupling to further external factors.' (l.349)*

*'The non-vanishing fourth KM coefficient in the $\delta18O$, which indicates forcing beyond typical Gaussian white noise, could point to an external trigger that directly acts on the Greenland temperatures.' (l.359)*

*'The results obtained in our analysis do not give a clear answer to the question for the exact mechanism that triggered DO events. In principle, the revealed double-fold bifurcation would allow for bifurcation-induced transitions and thus for a limit-cycle*

*behaviour. However, the records show that the system does not track the stable fixed point branches until the bifurcation points, but tends to transition earlier (not shown). Also, the structure of the δ 18 O drift is incompatible with a deterministic cyclic motion in the dust-δ 18 O plane. In fact, the specific structure of the double-fold bifurcation leaves room for a weak barrier between stadial and interstadial states in the vicinity of the bifurcation point, thus creating a 'channel'-like passage, through which the system passes.'*

In short:

- We agree that a coupling to hidden variables which cannot be subsumed in a SDE representation can be understood as a type of non-stationarity with respect to the dynamics of the observed variables.

- We believe the referee understands our claim that the data is stationary in the sense that we categorically exclude any coupling to hidden variables. This was, however, not our intention. We further agree that the relation between the KM coefficients and potential not bath-like hidden variables is not explained sufficiently in the manuscript as is.

- In a revised manuscript we would therefore point out the fact that we are potentially missing important parts of the systems state space and elaborate on how this relates to our assumptions of stationarity and how hidden variables can influence the estimation of the KM coefficients if one imposes the analytical framework which is build on stationarity. Also, we will argue that based on the very small autocorrelation, it is reasonable to assume a negligible coupling to hidden variables as a first-order approximation.

- We will also explain more precisely the steps we take in our analysis right at the beginning of the manuscript. That is, we start with the simplest models, and then elaborate on what these models do explain and what they fail to explain. Then we move to the next complicated models and do the same. So far, no study that was concerned with modelling Dansgaard-Oeschger variability has claimed to explain the dynamics of these events in full detail.

- We would also like to remind the referee of the fact that only limited data from ice cores is available for these long time periods and with sufficient temporal resolution. So all we can do is to investigate the observed variables and there is little we can do about the hidden ones – at least if we aim to stick to a methodology which is to a high degree data-driven.

I would not think that a stationary process described by the KM equations is consistent with a hypothetical nonstationary process that could not be rejected.

In principle, it is certainly true that the stationary KM equation, which forms the basis of our investigation, is not a consistent framework to describe a non-stationary process. We have made an effort to rule out obvious reasons for non-stationarity in the manuscript:

*'Excluding also the Last Glacial Maximum from the data, we restrict our analysis to the period 59–27 kyr b2k, which is characterised by a fairly stable background climate and persistent co-variability between dust and $\delta^{18}O$. To compensate for the remaining influence of the background climate on the climate proxy records, we remove a linear*

*trend with respect to the global average surface temperature from both time series (see App. A for the details).'*

For sake of clarity, in a revised of the manuscript, we will change the sentence:

*'Excluding also the Last Glacial Maximum from the data, we restrict our analysis to the period 59–27 kyr b2k, which is characterised by a fairly stable background climate and persistent co-variability between dust and δ¹⁸O.'*

to

*'Excluding also the Last Glacial Maximum from the data, we restrict our analysis to the period 59–27 kyr b2k, which is characterised by a fairly stable background climate, pronounced DO variability and persistent co-variability between dust and δ¹⁸O.'*

We include here a set of two unit root tests that indicate the data is stationary in the sense that there is no slow change in the process characteristics. These tests are the Augmented Dickey–Fuller test (ADF) and the Augmented Dickey–Fuller-GLS test (ADF-GLS). Both test for the possibility of a unit-root in the time series (null hypothesis). The alternative hypothesis is the time series does *not* have a unit root, i.e., it is stationary (in a broad sense).

The tests allow for different forms of trends behind the data. The ADF test allows for having solely a constant offset (no trend), solely a trend (no constant offset), a constant offset and a trend, and a constant offset, linear and quadratic trend. The ADF-GLS contains only a constant offset (no trend), or a constant and a trend.

| | ADF | | | | ADF | | | | ADF-GLS | | ADF-GLS | |
|---|---|---|---|---|---|---|---|---|---|---|---|---|
| | **Null Hypothesis: Non-stationarity** Reject if statistics is *smaller* than critical values | | | | **Null Hypothesis: Non-stationarity** Reject if statistics is *smaller* than critical values | | | | **Null Hypothesis: Non-stationarity** Reject if statistics is *smaller* than critical values | | **Null Hypothesis: Non-stationarity** Reject if statistics is *smaller* than critical values | |
| | **dust** | | | | **δ¹⁸O** | | | | **dust** | | **δ¹⁸O** | |
| | no trend | no constant | constant and linear trend | constant, linear, and quadratic trends | no trend | no constant | constant and linear trend | constant, linear, and quadratic trends | constant | constant and linear trend | constant | constant and linear trend |
| | statistics (*p*-value) [optimal lag] | statistics (*p*-value) [optimal lag] | statistics (*p*-value) [optimal lag] | statistics (*p*-value) [optimal lag] | statistics (*p*-value) [optimal lag] | statistics (*p*-value) [optimal lag] | statistics (*p*-value) [optimal lag] | statistics (*p*-value) [optimal lag] | statistics (*p*-value) [optimal lag] | statistics (*p*-value) [optimal lag] | statistics (*p*-value) [optimal lag] | statistics (*p*-value) [optimal lag] |
| critical value (p=0.05) | **-1.9410** | **-2.8620** | **-3.4112** | **-3.8333** | **-1.9410** | **-2.8620** | **-3.4112** | **-3.8333** | **-1.9470** | **2.8499** | **-1.9470** | **2.8499** |
| data | -4.9124 (1.566e-06) [6] | -5.0515 (1.753e-05) [6] | -5.3732 (4.082e-05) [5] | -5.481 (1.382e-04) [5] | -6.6017 (4.651e-10) [15] | -6.8034 (2.208e-09) [15] | -7.3196 (2.644e-09) [15] | -7.5415 (4.278e-09) [15] | -3.4422 (6.373e-04) [6] | -5.2217 (9.989e-06) [5] | -3.7747 (1.904e-04) [15] | -6.6558 (1.487e-08) [15] |

All tests point to an absence of a unit root in our time series (results are valid also as p=0.01).

Secondly, we take the same KM analysis we performed and apply it in the first half and second half of the time series, to showcase the overall structure of the potentials/drifts remains unaltered.

[Figure]

We see that the overall shape of the drifts/potentials remains the same. Naturally, the zero-crossings of the drift change since in a one-dimensional analysis the coupling between the proxies cannot be considered in one-dimension. A similar recipe is taken again by dividing the time series into 3 subsequent thirds.

[Figure]

We note that this last case considers performing the KM analysis over a time series of ~2200 data points. Even in this regime of a very low number of data points, we observe the same double-well structure in the dust and single-well structure in the $\delta^{18}O$.

After these considerations the only source of potential non-stationarity that is left is the variability of other climatic subsystems (e.g. the AMOC) which are potentially coupled to Greenland temperatures and atmospheric circulation investigated in this manuscript, that is the coupling to hidden variables.

Finally, it should be mentioned that several influential investigations of the same data have followed a similar reasoning, that is, they rely to some extent on the assumption that the data

generating process can be described by autonomous model equations that comprise only the observed variable and no hidden variables. A small selection is:

Ditlevsen, P. D. Observation of α-stable noise induced millennial climate changes from an ice-core record. Geophys. *Geophys. Res. Lett.* **26**, 1441–1444 (1999).

Boers, N. *et al.* Inverse stochastic-dynamic models for high-resolution Greenland ice core records. *Earth Syst. Dyn.* **8**, 1171–1190 (2017).

Kwasniok, F. Analysis and modelling of glacial climate transitions using simple dynamical systems. *Philos. Trans. R. Soc. A Math. Phys. Eng. Sci.* **371**, (2013).

Livina, V. N., Kwasniok, F. & Lenton, T. M. Potential analysis reveals changing number of climate states during the last 60 kyr. *Clim. Past Discuss.* **5**, 2223–2237 (2010).

Say, we have a nonstationary Orntsetin-Uhlenbeck process of dx/dt = -a*x + B(t) + c*xi(t), (OU) where xi is white noise, and B(t) = b*sin(sin(2*t)+t) is some regular nonstationarity. It mimics some regime behaviour with sudden and regular transitions. We can easily see that the pdf of x is bimodal. If we didn't know the underlying process generating eq., and perhaps we somehow overlooked the regularity of the transitions, we might think the underlying model is:

dx/dt = F*x + c*xi(t), (H0)

where F = -V(x), V(x) being a double-well potential function. If we are in the small noise limit, we know that the pdf takes the shape of V(x), so, we could estimate V that way. Furthermore, we can estimate the noise strength 'c' in some standard way too.

Now the question is if it matters at all that we have an H0 other than the true process OU. It would not matter if in any appreciable way the processes perform the same, i.e., when, loosely speaking, they are consistent or approximately equivalent. For example, we can derive the probability distribution of residence times from H0, and perform a statistical test if our residence time data is consistent with that, or we can reject H0.We want to perform a so-called crucial experiment (experimentum crucis).

We start our investigation by estimating the KM coefficients up to 4[th] order of the isolated 1D time series for both, $\delta^{18}O$ and dust. In the case of the dust, this does immediately imply a Langevin type model, since $D_4$ is negligible. In the case of $\delta^{18}O$, we do not propose a full model since there are several SDE models which are consistent with the retrieved KM coefficients.

As proposed here by the referee, we then test the consistency of the results (that is of structure of the drift and diffusion and the $D_4$ in case of the $\delta^{18}O$) with respect to the data. We explicitly emphasise that the data contradict the recovered KM coefficients in the 1D case. This motivates us to consider the coupling between $\delta^{18}O$ and dust in the next more complicated setup. In 2D, we do not test a specific model, since the correspondence between KM coefficients and model coefficients is not trivial.

> *'At first sight, the monostability of the reconstructed $\delta^{18}O$ potential contradicts the apparent two regime nature of the time series. There are two possible explanations for this discrepancy: First, regime switching of monostable stochastic process can be achieved through complex noise structures (e.g., Lévy-like noise, generalised Fokker–Planck equations, or fractal motions) (Chechkin et al., 2003, 2004; Metzler and Klafter, 2004). Secondly, a similar effect can be obtained in a two-dimensional setting if the dynamics of one dynamical variable explicitly depends on the other, which would be impossible to judge from the one-dimensional analysis presented so far. Thus, within the limits of this analysis – that is assuming that the process is Markovian and*

*stationary and that the system under study is fully represented by dust and δ¹⁸O (no coupling to further hidden variables) – the source of the regime switching must either be endowed by more complex noise processes or by the coupling between the dust and the δ¹⁸O systems.'*

Considering H0 of the authors, another likely feature based on which H0 can be rejected is the saw tooth asymmetry, in particular, that the cold to warm, c2w, transitions are much more rapid than the warm to cold, w2c, ones.

If we consider the setting of a standard and stationary Langevin process as

$dx(t) = - a(x)dt + bdW(t),$

Where $a(x)$ is mean-reverting, we indeed have a process that is one-dimensional and time symmetric. We can easily break the time symmetry by introducing a discontinuous element, as for example

$dx(t) = - a(x)dt + bdW(t) + cdJ(t)$

where $J(t)$ is a Poisson process with a jump rate $\lambda > 0$. In this simplest of formulations, if there is at least 1 jump from the Poisson jump process, the process becomes time-asymmetric. (From an applied point of view, there are some requirements regarding the relation of the amplitudes of $a(x)$, b, and c, and the smoothness and differentiability of $a(x)$.) Thus, breaking time-symmetry in a one-dimensional setting is not impossible, on the contrary, a single discontinuous trajectory does the job.

We do not explicitly mention this or any other explicit stochastic process, but instead show that there must exist discontinuous trajectories in our time series (Fig. 2, lower panels with the $D_4(x)/D_2(x)$ ratio and Fig. 3, depicting the $Q$-ratio). This offers one answer to why we can have time asymmetry even in a one-dimensional setting. As correctly stated by the reviewer, it nevertheless does not exclude a potential presence of non-stationarities in the data. We note that in two dimensions time asymmetry is easily created by a state-dependent drift term $a(x,y)$.

As a last point, we agree there are other methods to estimate the various parameters in the system, as mentioned in regards to estimating $F(x)$ or c. In our manuscript we have focussed solely on the shape of $F(x)$, and in a very agnostic manner, avoid discussing the exact functional values of $F(x)$ of amplitude c.

Looking at the dust time series of Fig. 1 with much naivity wrt. physics, but with some experience about dynamical systems, I would think that c2w is an attractor crisis, whereas w2c is a noise induced tipping, and there is some slowly drifting control parameter, i.e., a nonstationarity when we exclude that parameter from our state variables.

The authors also make a reference to attractor crisis, in terms of a saddle-node bifurcation, but in some other context. It is the context of slices of a 2D potential function. This does not sound correct.

Following from the feedback of the other reviewer, even in a setting where in one dimension one has a double well and in the other a single well one might find a spurious double fold bifurcation in a KM analysis under the right rotation of the basis vectors. As we now explain, while it is not possible to reach a definite answer on the entire physical mechanism leading

to the observed bimodality from our approach, we can derive some very interesting conclusions, as we now discuss in detail. Please see the answer to the other reviewer.

The paper is very well written in a way, but it doesn't make for a very pleasant reading journeying through flawed results, starting with the single variable approach, and then — at least as i suspect — even the 2 variable approach.

I attach the pdf of the manuscript with comments saved as annotations. Hopefully the authors find it useful in some way.

Note: I always review non-anonymously, and never make recommendation for or against publication. The recommendation that i make is only to circumvent the rigidity of the submission system, and therefore please consider it void.

Tamas Bodai

Please also note the supplement to this comment:

https://esd.copernicus.org/preprints/esd-2021-95/esd-2021-95-RC2-supplement.pdf

We thank the reviewer for his valuable comments. We agree that, particularly using proxy data in the context of paleoclimatology, much of the outcome is subjected to limitations and imperfections. Still, we believe our proposed revisions will take this into account thoroughly and provide important new conclusions e.g. about how the two variables, dust and $\delta^{18}O$, are coupled. As discussed in the answer to the other reviewer, the widespread assumption that the bistability is rooted in the temperature, which then drives the dust variable as a dependent variable, seems unlikely from our results.

The particular aspect of including both 1D and 2D approaches follows from the fact that the 1D approach is a natural starting point for our investigation also considering other recent studies on date-driven model inference from Greenland ice core time series, and provides the reason why the 2D analysis is necessary, therefore motivating the second step of analysis.

**Specific Comments**
* * *
l.12     (3) the $\delta^{18}O$ record is discontinuous in nature, and mathematically requires an interpretation beyond the classical Langevin equation.

A discrete time sampling cannot result in a continuous series. Also, is this new result?

We agree with the reviewer that a discretised time series cannot be continuous, but this is not the point here. What we express in the paper is not a relation of the continuity or discontinuity of the discretised continuous time series, we refer solely to stochastic processes (not their discretised versions or measured time series). These, mathematically, can be continuous or discontinuous. The result is, to the extent of our knowledge, new. It arcs back to previous results that suggest, e.g., other potential discontinuous stochastic models to describe the proxies (Lévy processes), yet here

we include direct estimations of textbook measures to show that the data is indeed discontinuous (mathematically).

l.32     The most prominent example of past abrupt climate shifts are the Dansgaard–Oeschger events

maybe one of the most

Thank you for this comment. We will replace the statement by:

*One of the most prominent examples of past abrupt climate shifts are the Dansgaard–Oeschger events*

l.34     a series of sudden warming events that dominated Greenland temperatures throughout the last glacial cycle,  (Johnsen et al., 1992; Dansgaard et al., 1993; North Greenland Ice Core Projects members, 2004).

e.g., Refs was stroke through by the referee. We will change to:

[...] *a series of sudden warming events that dominated Greenland temperatures throughout the last glacial cycle, (e.g. Johnsen et al., 1992; Dansgaard et al., 1993; North Greenland Ice Core Projects members, 2004).*

l.57     The key concept is to regard the paleo-climate record as the realisation of a Markovian and stationary stochastic process (Kondrashov et al., 2005, 2015) which can be described in terms of a stochastic differential equation.

What if the sudden warming is a bifurcation

Here, we refer to our detailed answer to the referee's general comment.

l.63     The Kramers–Moyal equation generalises the Fokker–Planck description of stochastic processes, including explicitly the presence of discontinuous elements.

Is this equivalent with a nonstationary framework? I suppose not.

There is no difference here, the Fokker–Planck equation is a limit case of the Kramers–Moyal equation for vanishing higher-order Kramers–Moyal coefficients. The formulation of a non-stationary Fokker–Planck process is identical to that of a Kramers–Moyal process, it simply involves defining the Kramers–Moyal coefficients with a temporal dependency. See Risken (1996) chapter 4 for a general derivation of the non-stationary Kramers–Moyal equation and how to constrain it to a Fokker–Planck equation aided by Pawula's theorem. There are no limitations or different impacts of non-stationarity in either case.

l.67     In Sec. 2 we introduce the paleo-climatic proxies under examination and the detrending method used to ensure that the data is approximately stationary.

Is this correct methodology? Why would the detrended process not be nonstationary? I.e., is stationarity really "ensured"?

The detrending removes the anyways small non-stationarity that stems from the slowly varying background climate. However, it does not remove potential non-stationarities of the sort we believe the referee has in mind, namely those which are due to couplings to hidden variables. A series of tests have been included in our earlier reply above.

l.72    This is consequently discussed in Sec. 4.2, where we uncover the conditioned potential landscapes of the joint proxy process.

conditioned on what?

Conditioned on the respective other variable. However, in view of the comment by the other referee, we will abandon the notion of conditional potential anyways and rephrase this sentence accordingly.

l.109    To assess whether the data is Markovian, we analyse the auto-covariance function of the increments of the detrended data.

Nonstationarity overlooked this way.

Please see our detailed answer to the referee's general comment.

l.121    A prominent example for a stochastic process is given by the stationary Langevin equation

I thought in a Langevin eq. you have the differential on the left. This looks more like Ito or Stratanovic, and you should actually specify which one.

The formulation is given in Itô.

l.124    If the properties of the dynamics do not change over time, i.e. $a(x)$ and $b(x)$ do not depend on time, these processes are called stationary.

This is what i would contest.

Please see our detailed answer to the referee's general comment.

l.125    While the Langevin equation is continuous in time, stochastic processes can in principle have discontinuous features, such as sudden jumps.

But for what value of dt? var[dB] goes to 0 as dt does.

Discontinuity in our case is not related to a sampling rate. Please see the explanation below where we now explain in detail what is considered continuous/discontinuous in our formulation.

l.126    An easy way to incorporate discontinuities is to include in Eq. (1) an elementary Lévy process $L(t)$, modulated with an amplitude $h(x)$ (Applebaum, 2011)

Isn't it a bit of an interpretational issue? var[dL] does not exist. But then what' in the limit of 0 for dt?

The inexistence of the variance has no relation with the continuity of a stochastic process. The variance of a Poisson process always exists (all moments exist) and this is truly a discontinuous process. Please see the explanation below on Lindeberg's continuity condition.

I.135    If a single particle's motion is governed by the Langevin equation, its probability
         density function p(x, t|x ' , t ' ) evolves according to the Fokker–Planck equation,

         Does the FP not govern the unconditional density rather?

         Both forms exist. See Risken (1996) eq. 4.16 for a conditional density formulation
         (with the general Kramers–Moyal operator), or eq. 4.52 and 4.53 for solely the
         conditional density formulation of the Fokker–Planck equation.

I.140    So, this should tell whether (1) is Ito or Stratanovic. Please spell it out. My reference
         is Risken's book for this.

         The formulation is given in Itô.

I.145    Giving up the condition of continuity, the temporal evolution of the conditional
         probability density follows the Kramers–Moyal equation

         Can you please indicate here how discontinuity is allowed?

         Consider the Lindeberg's continuity condition *C(t)* for a markovian process, which
         states that a trajectory *x* is continuous if

$$C(t) \triangleq C(x, t, \delta) = \lim_{\tau \to 0^+} \frac{Prob[\ |\Delta x(t)| > \delta \quad |_{x(t)=x}]}{\tau}$$

$$= \lim_{\tau \to 0^+} \frac{\int_{|\Delta x(t)|=|x'-x|>\delta} p(x', t + \tau|x, t)\, dx'}{\tau} = 0$$

         For all *δ>0*, all *x* and *t*, with *Δx(t)=x(t+τ)-x(t)* and *p(x',t+τ|x,t)* the conditional prob.
         density. Take a stationary distribution for a Langevin-like process, characterised by a
         drift $D^{(1)}$ and a diffusion $D^{(2)}$, and no higher-order KM terms ($D^{(4)}=D^{(3)}=0$, and Pawula's
         theorem), and insert it into the Lindeberg's continuity condition

$$C(x, t, \delta) = \lim_{\tau \to 0^+} \frac{1}{2\tau\sqrt{\pi D^{(2)}(x, t)\tau}} \left\{ \int_{-\infty}^{-\delta+x} \exp\left(-\frac{(x' - x - D^{(1)}(x, t)\tau)^2}{4D^{(2)}(x, t)\tau}\right) dx' \right.$$
$$\left. + \int_{x+\delta}^{\infty} \exp\left(-\frac{(x' - x - D^{(1)}(x, t)\tau)^2}{4D^{(2)}(x, t)\tau}\right) dx' \right\}$$

         Which we separate into two integrals:

$$\equiv \lim_{\tau \to 0^+} \frac{1}{2\tau\sqrt{\pi D^{(2)}(x, t)\tau}} (I + II)$$

         With I as

$$I = \int_{-\infty}^{-\delta+x} \exp\left(-\frac{(x'-x-D^{(1)}(x,t)\tau)^2}{4D^{(2)}(x,t)\tau}\right) dx'$$

$$= \sqrt{4D^{(2)}(x,t)\tau} \int_{-\infty}^{(-\delta-D^{(1)}(x,t)\tau)/\sqrt{4D^{(2)}(x,t)\tau}} \exp(-u^2)\, du$$

$$= \sqrt{4D^{(2)}(x,t)\tau}\, \frac{\sqrt{\pi}}{2}\, erfc\left[\frac{\delta+D^{(1)}(x,t)\tau}{4D^{(2)}(x,t)\tau}\right]$$

Take an approximation of the error function expanded in $\tau$

$$I = 2D^{(2)}(x,t)\tau \frac{\exp\left\{-(\delta+D^{(1)}(x,t)\tau)^2/4D^{(2)}(x,t)\tau\right\}}{\delta+D^{(1)}(x,t)\tau} \times$$

$$\left\{1 - \frac{2D^{(2)}(x,t)\tau}{(\delta+D^{(1)}(x,t)\tau)^2} + \frac{12(D^{(2)}(x,t)\tau)^2}{(\delta+D^{(1)}(x,t)\tau)^4} + \cdots\right\}$$

In the limit of $\tau \to 0^+$, I=0. *Mutatis mutandis* for II. This proves the continuity of a Langevin-like process. The central assumption here lies on the fact that $D^{(1)}$ and $D^{(2)}$ are sufficiently well behaved and smooth.

We now consider the case wherein there are higher-order moments and we consider directly the conditional moments and a limit $dt \to 0$, such that

$$M^{(m)}(x) = \lim_{dt\to 0} \frac{1}{dt}\langle |x(t+dt)-x(t)|^m |_{x(t)=x}\rangle$$

From here we show that

$$\lim_{dt\to 0} \frac{Prob[|\Delta x| > \delta\,|_{x(t)=x}]}{dt} \leq \frac{M^{(m)}(x)}{\delta^m}$$

Which, importantly, is *not* zero. Follow our definition of the Kramers–Moyal and conditional moments as given in the paper to arrive at

$$\langle |u-x|^m\rangle|_{x(t)=x} = \int_{-\infty}^{\infty} du\, |u-x|^m\, p(u,t+dt|x,t)$$

$$= \int_{-\infty}^{x-\delta} du\, |u-x|^m\, p(u,t+dt|x,t) + \int_{x-\delta}^{x+\delta} du\, |u-x|^m\, p(u,t+dt|x,t)$$

$$+ \int_{x+\delta}^{\infty} du\, |u-x|^m\, p(u,t+dt|x,t)$$

From which we construct

$$\langle |u-x|^m\rangle|_{x(t)=x} \geq \int_{|u-x|>\delta} du\, |u-x|^m\, p(u,t+dt|x,t)$$

by ignoring the second term in the 3 integrals above. Use now that

$$|u - x|^m > \delta^m$$

Take the limit as above to obtain

$$\lim_{dt \to 0} \frac{1}{dt} \langle |u - x|^m \rangle |_{x(t)=x} \geq \delta^m \lim_{dt \to 0} \frac{1}{dt} \int_{|u-x|>\delta} du \; p(u, t + dt | x, t)$$

Which is the Lindeberg condition above. Previously we saw that Langevin-like equations result in an equality with *zero*. The existence of any higher-order KM coefficients dilutes this to yield a non-zero value, proving the discontinuity. This inequality can be made into a strict equality in certain cases, like the aforementioned stochastic process with Poissonian jumps. In the case of Poissonian jumps, we know that the discontinuity is simply given by their jump rate λ.

[These derivations are taken from MRR Tabar (2019) *Analysis and Data-Based Reconstruction of Complex Nonlinear Dynamical Systems*]

l.153 However, for numerous of these stochastic processes, the KM coefficients can be related to the properties of the stochastic process in the spirit of Eq. (4).

Provide a reference please.

In a revised version of the manuscript we will cite Risken (1996), Tabar (2019), and Anvari *et al.* (2016).

l.168 To retrieve the KM coefficients D m (x) from a single realisation of a stochastic process, i.e. a single time series, we evaluate the transition probability densities in the limit of a vanishing time step τ → 0, which numerically corresponds to considering the shortest increment Δt in the data (τ → Δt).

Is it so? I suspect that it might be but might be not. It can be that you are trying to do your estimation of a process, which requires a higher resolution of data.

Essentially, you might be unable to perform the estimation (even if the modeling assumption was completely perfect). What you want to see perhaps is that as you reduce the time resolution, your estimate seems to converge. I.e., you might need to deliberately coarsen your resolution.

Below we include a reploting of Fig. 2 in the manuscript wherein we coarsen the resolution τ from the original 1 to 2 and 3 (i.e., we consider only half and a third of the total amount of data). The overall shapes remain unaffected. Bandwidth selection of the Nadaraya–Watson estimator still follows Silverman's optimal bandwidth selection, using an Epanechnikov kernel. The total amount of data-points considered are: *τ=1*, 6399 data-points; *τ=2*, 3199 data-points; *τ=3*, 2133 data-points.

We note that following our observation from the auto-correlation of the increments of the time series, seen in Appendix B, Fig. B1, we can safely claim we are above the Einstein–Markov length and thus these estimates are not affected by microscopic noise correlation and subsequently taking coarser resolutions yield identical results (just as shown), which are only affected by having fewer data-points from which to draw the estimates.

[Figure]

I think it is not a good procedure to perform the estimation and then generate ample synthetic data by the assumed model and check if estimates at a certain resolution are biased.

We assume the referee inserted the 'not' inadvertently and hence we interpret this comment as if the referee suggested to apply the above procedure. In this case our answer would be as follows: To the best of our understanding, this is not something we can perform in our evaluation: First, we can naturally estimate all statistical moments of the time series, but to generate synthetic data we have to make an assumption on the underlying process. For instance, the Tabar Q ratio presented in Figure 3 shows the presence of discontinuities or jumps in the process. We refrain from formulating a specific model for these jumps on the basis of the limited data we have. Our paper goes to great length to avoid precising a model – that we believe is one of the strengths of the paper. Thus, unless we constrain ourselves to a model, we cannot do this.

If the 'not' was not a typo, then we do not fully understand this comment as we do not simulate any synthetic data.

It is possible that at the biased estimates the biases are indicated low, but at the true "unbiased" values estimates are very biased at that time resolution. It might sound circular, but it isn't, i believe.

l.178   Thus, values of the ratio D 4 (x)/D 2 (x) close to zero imply continuous sample paths with no jumps in the data.

The estimate will be a finite nonzero number. How do i know if this is small, close to zero, or not? What should i compare it with?

The term is proportional to the sampling rate $\tau$ as $M^2 \sim \tau$ and $M^4 \sim \tau^2$ (M the conditional moments) if we are considering a Langevin process. Thus, the smaller the sampling rate, the smaller the ratio. For the case of jumps, the relation $M^4 \sim \tau^2$ is no longer valid.

l.183  This assessment can be refined by regarding the Lehnertz–Tabar Q-ratio (Lehnertz et al., 2018), which takes advantage of the fact that continuous and discontinuous systems 'scale' in a different fashion. While a purely continuous stochastic process diffuses proportionally to time t (or possibly a power of time $t^\beta$ in anomalous diffusions (Einstein, 1905; von Smoluchowski,1906; Havlin and Ben-Avraham, 1987)), discontinuous processes can cover large distances in short times, i.e. jump, which causes them to exhibit no scaling relations with time t.

I don't know what this means.

We will include an appendix in the revised manuscript detailing what we mean with discontinuity, how is the Kramers–Moyal formulation a candidate to describe discontinuous processes, and how to understand the scaling relations in order to understand the Q-ratio. We shall as well reformulate this passage and subsequently point to the appendices that best explain what is meant with scaling.

l.205  For various applications where the fluctuations are not comparable in size, i.e. where the diffusion elements are not of similar scale, one can draw a clearer picture of the motion of the two-dimensional system by referring to an effective vector field

Reference please.

In fact, we did not adopt this approach from other studies. Thus we will replace the above statement by:

*Given the different levels of diffusion along the two dimensions, we introduce here an effective vector field by rescaling the drift components by the value of the corresponding diffusion in each direction.*

l.211  Similarly to the one-dimensional case, one can obtain potential landscapes as integrals over the two drifts:

Reference please. How is the potential defined over a multidimensional phase space? I think there is only one potential function, not one per variable! See p. 133-134 of Risken, or https://iopscience.iop.org/article/10.1088/1361-6544/ab86cc e.g. eq. (3)

But mind that you can only say that you found a potential from obs data if your modelling assumptions are correct.

In a two dimensional state space one can consider the dynamics along a single dimension while hypothetically freezing the motion along the other dimension. This leads to a 1D setting, where the typical notion of a potential applies.

However, we decided to abandon the notion of the conditioned potentials and will therefore reformulate statements and the equations accordingly.

l.228  We find the second KM coefficient to be fairly constant (Fig. 2 (c)) and the ratio between fourth and second KM coefficients to be negligible (Fig. 2 (d)), which suggests that a Langevin process with additive noise is a viable description of the isolated dust dynamics.

I cannot judge whether this is large or small.

Please see our answer to the referee's comment to line 178.

l.233    Note that the model equations employed here are by construction symmetric with respect to time, therefore, as it is, the model cannot reproduce the temporal asymmetry that is visually suggested in the dust record.

This should raise concern about the modeling assumption.

In fact, the model equations only become symmetric wrt. time after estimating the diffusion to be constant over space and the D4 coefficient to be negligible. They are not symmetric by construction as explained in our detailed answer to the referee's general comment. We will therefore rephrase:

'Note that with the second KM coefficient being constant and the fourth KM coefficient being negligible, the obtained model equations are symmetric with respect to time. Therefore, as is, the model cannot reproduce the temporal asymmetry that is visually suggested in the dust record.'

l.236    Most prominently, the drift has only a single stable fixed point (zero-crossing of the drift), or equivalently, the potential function exhibits only a single well.

The drift implies a fixed point, maybe. Terminology.

Yes, indeed. We will replace the statement by:

'Most prominently, the drift has only a single zero-crossing, or equivalently the potential function exhibits only a single well.'

l.240    Moreover, we find that the fourth KM coefficient D 4 (x) for the δ 18 O is of the same magnitude as the second KM coefficient D 2 (x) (Fig. 2 (h)).

I see that D4 is compared to D2 actually. But what does comparability, a ratio of 1, mean? Still, is this much in some sense?

Please see our answer to the referee's comment on line 178.

l.239    Given the high correlation between the dust and the δ18O records, the differences in the reconstructed potentials and the ratio between fourth and second KM coefficient are remarkable.

Yes, but in the same time we can just plot the histograms of dust and d18O and we will see a bi- and unimodal distribution, no? IF the 1D models were correct, and the noise was really small, then the histograms give you the respective potential functions.

It is true that already the PDF of dust and δ18O suggest a qualitative difference between the two time series. We will integrate the PDFs of both records in the figure 4 of the revised manuscript. However, the correspondence between a records histogram and the underlying potential can easily be corrupted by multiplicative gaussian noise, let alone more complex noise. So the qualitative difference in the potentials can only be evidenced in this more comprehensive type of analysis.

Maybe thought the very correlation bw. dust and d18O prompts that the 1D models cannot be good assumptions.

Sorry, we are not sure we understand this comment.

One needs to somehow test the hypothesis of the model. Checking Markovianity in the appendix is probably very insufficient.

Please see our response to the referee's general comment above.

l.243    At first sight, the monostability of the reconstructed δ 18 O potential contradicts the apparent two regime nature of the time series.

Is the histogram bimodal? Bimodality would imply regime behaviour (of whatever origin), but regime behaviour does not imply bimodality of the hist'.

We fully agree with the referee. The histogram of the $\delta^{18}O$ is in fact monomodal. However, a two-regime nature of the record can indeed be evidenced by eye. In the manuscript we introduced the term two-regime nature maybe with a lack of explanation. We would therefore add a paragraph on this already in the data section.

l.258    In Fig. 3, we clearly see a constant relation of $Q(x, \tau)$ with respect to $\tau$ for the $\delta^{18}O$ record, suggesting that this stochastic process includes jumps.

Or that your assumption of stationarity is crucially wrong. We want certainty. There is little value in evidence alone that can imply two very different things. We want to know which one is true.

Please see our reply to the referee's general comment.

l.274    The reconstructed conditional potential $V_{1,0}(x_1|x_2)$ of the dust is displayed in Fig. 4 (d). As a conditioned potential, it can be read by taking vertical 'slices' of the potential.

You haven't defined that. And so we don't see that the 1D potentials are really slices of the one multivariate pot'.

As mentioned previously, we will abandon the notion of conditioned potentials, and hence reformulate the above statement accordingly.

l.290    However, the position of the minimum δ 18 O* appears to be determined by the dust in a continuous manner, with high rate of change for intermediate dust values whilst no change for more extreme dust values.

What does this refer to?

We meant to say that for intermediate values of the dust a small change in the dust – seen as a control parameter – causes a fairly strong change in the position of the minimum of the conditioned $\delta^{18}O$ potential (high rate of change).

As we abandon the notion of conditioned potentials, we will rephrase the above statement.

l.297    These findings are consistent with the observed regime switching of both records, which we struggled to reconcile with the results obtained from the one-dimensional analysis.

Perhaps there is an amount of hindsight in this but this is exactly how time series would look like when we have a 2D double well potential with two axes of symmetry.

Do not define the variables by the axes of symmetry, and you will see coordinated transitions. Furthermore, if you tip your coordinate system only slightly, closing small

angles with the axes of symmetry, then you have the chance that despite the transitions, the marginal distribution of one of the variables will be still unimodal.

However, such a system would not feature asymmetry in that you have a more sudden transition in one direction than the other.

We fully agree with the referee's comment. Since this point is hardly a criticism, we do not intend to change the manuscript in response to this. We do point out the possibility to obtain two-regime-like, yet unimodally distributed time series in one variable, if this variable is suitable coupled to another one.

'*At first sight, the monostability of the reconstructed δ 18 O potential contradicts the apparent two regime nature of the time series. There are two possible explanations for this discrepancy: First, regime switching of monostable stochastic process can be achieved through complex noise structures (e.g., Lévy-like noise, generalised Fokker–Planck equations, or fractal motions) (Chechkin et al., 2003, 2004; Metzler and Klafter, 2004). Secondly, a similar effect can be obtained in a two-dimensional setting if the dynamics of one dynamical variable explicitly depends on the other, which would be impossible to judge from the one-dimensional analysis presented so far.*' (l.243)

l.323   This leaves open the possibility that transitions between stadial and interstadial states are mainly induced by noise as argued by, e.g., Ref. (Ditlevsen et al., 2007) (i.e. noise-induced tipping), facilitated by a shallow potential barrier close to the minima of the (effective) vector field.

I'm really not convinced. I think the consideration of an important variable is missing. It's worth considering this recent paper:

https://link.springer.com/article/10.1007/s00382-020-05476-z

The sawtooth feature is there, for one thing. But my feeling is that the rapid transition is rather due to crisis, the disappearance of a regime, as a result of nonstationary dynamics (nonstationary in a reasonable modeling framework).

Please see our detailed reply to the referee's general comment.

l.358   In particular, we have shown that the δ 18 O record cannot be treated as a time-continuous process.

If the fast transition is attractor crisis, then it is a continuous process. You can only make this statement if the modelling assumption was water tight. But it isn't. So, strictly speaking, you have not shown what you say.

In the light of what has been said before, the statement should be made more precise:

'*In particular, we have shown that the isolated δ18O record cannot be treated as a time-continuous process in a one-dimensional SDE setting.*'

l.364   In principle, the revealed double-fold bifurcation would allow for bifurcation-induced transitions and thus for a limit-cycle behaviour.

What you found is not what you say. I think the methdology of (13) is problematic.

We are not sure if we fully understand this comment. We will not use (13) in the revised manuscript and instead explain the presence of the double-fold bifurcation in terms of the nullclines of dust in the δ¹⁸O, dust state space.

l.372   We conclude therefore that – based on our results – the DO transitions are to large degree induced by noise, acting on the background of a double-fold dynamics governing the dust, for which the δ18O acts as control parameter.

I'm not convinced at all, sorry. How could the faster dynamics control the slower like this?!

We agree that this statement should be attenuated by saying:

*'In the two dimensional subspace spanned by δ¹⁸O and dust, assuming that no couplings to hidden variables substantially influence the dynamics, our results suggest that the DO transitions are to large degree induced by noise, acting on the background of a double-fold dynamics governing the dust, for which the δ¹⁸O acts as control parameter.'*

l.373   These findings do not contradict previous studies that proposed limit-cycle models to explain the δ18O record, since the cyclic motion was not expected to happen in a state space comprised of Greenland temperatures and atmospheric large scale circulation (Kwasniok, 2013; Lohmann and Ditlevsen, 2019).

I think your finding is the result of your methodology. It is your methodology that should/could be contradicted. Consider the possibility that the cited papers do contradict your methodology.

I'm not sure how, but some statistical test might be able to reject your null-model. Perhaps the saw-tooth asymmetry feature can be a basis of such a formal, precise test.

Or maybe another test is whether the residence time data is consistent with the predictions of the fitted model. Of course, i mean a formal hypothesis test again.

There is really no value in further considering models that can be rejected by data outright.

A question is whether negative results should be published.

Many studies experiment with simple models in order to investigate Dansgaard–Oeschger variability. Some of those models – as a double well potential plus noise (Lohmann, 2018; Kwasniok, 2013; Livina, 2010) – can *a priori* be seen to not capture essential features of the NGRIP δ¹⁸O time series. Nevertheless assessing their performances, and finally deciding they would not explain the phenomena under study is a valuable contribution.

The difference in our case is that due to the ambiguity of the 4ᵗʰ-order KM coefficient we do not propose a full model in order to simulate the process. We acknowledge that the referee does not believe that DO variability can be explained in the dust-δ¹⁸O state space and we will emphasise the possibility of missing a crucial dimension of the dynamics in our analysis in the revised version of the manuscript.

I think it depends on how nontrivial they are. So, it should be considered by the authors that looking at Fig. 1 a stationary model is trivially wrong.

We have already explained why even a stationary 1D model is not trivially wrong.

l.416    Our analysis considered only a two-dimensional projection of the very high-dimensional dynamics and can therefore not be expected to deliver all details of the triggering mechanism.

Regarding the merit of a study, we can consider the question: can attractor crisis be modelled by a stochastic process that features some large jumps?

We do not fully understand the question. An attractor crisis as a potential trigger of c2w transitions would in first place need a higher dimensional state space. However, as already mentioned, we aimed to carry out a data-driven study, but no additional data is available for the investigated time period at the required resolution.

l.417    Neither the ocean-focused self-sustained oscillation hypothesis, nor the idea that the atmosphere acts as a trigger, can be ruled out based on our findings.

Isn't this a problem?! Wouldn't it be a problem that your paper did not reduce uncertainty about which one it is?!

Of course, we had hoped that our investigation would more clearly point to either the one or the other direction. However, the results are as they are and still contain valuable information.

l.422    Finally, our study underlines the need for higher resolution data, as the scarcity of data points is a limiting factor for the quality of non-parametric estimate of the KM coefficients.

It would have been good to see in an appendix the dependence of estimates on time resolution.

We assume the referee refers to his comment made in the general comments section:

Essentially, you might be unable to perform the estimation (even if the modeling assumption was completely perfect). What you want to see perhaps is that as you reduce the time resolution, your estimate seems to converge. I.e., you might need to deliberately coarsen your resolution.

Please see the 'General Comments' section for our answer.

---

## Author Response (AR1)

Answers: green
Comments by reviewer: violet

Comment on esd-2021-95

Tamás Bódai (Referee)

Referee comment on "Changes in stability and jumps in Dansgaard–Oeschger events: a data analysis aided by the Kramers–Moyal equation" by Leonardo Rydin Gorjão et al., Earth Syst. Dynam. Discuss., https://doi.org/10.5194/esd-2021-95-RC2, 2022

First of all, we would like to thank the referee for his thorough and detailed review of our submitted manuscript. His scrutiny from an outside perspective revealed that the original manuscript was not as clear as it should have been about some central points regarding its conceptual design. We have taken the opportunity to substantially improve the manuscript's clarity with respect to the conceptual study design, its explanatory power and its limitations in response to this review.

We have put substantial additional effort into improving the manuscript and have aimed to split it into two parts as suggested by the other referee, with one part focussing on the one-dimensional analysis of the individual d18o and dust proxy time series and the other part assessing the two-dimensional drift and therewith the coupling between the two records. With regard to the first, we have encountered some technical difficulties that need more time to be addressed thoroughly. For now we therefore e resubmit a strongly streamlined manuscript which solely focuses on the reconstruction of the two-dimensional drift underlying the coupled (d18o, dust) time series. We will pursue the investigation of the higher order KM coefficients in one dimension and aim at submitting a second manuscript in this regard at a later stage.

The paper "Changes in stability and jumps in Dansgaard–Oeschger events: a data analysis aided by the Kramers–Moyal equation" analyses d18O and dust data from a Greenland ice core in order to gain further understanding of the famous Dansgaard-Oeschger (DO) events. They preprocess the time series with the aim of establishing a stationary stochastic process. They estimate Kramers-Moyal (KM) coefficients, which could possibly reveal jumps in the process, outside the framework of the Fokker-Planck equation, corresponding to what can be naively seen as regime transitions. They explore the added value of joint fitting of the d18O and dust data over treating them separately.

The referee correctly summarised the approach of our manuscript. There are three minor points we would like to clarify.

First, the usage of the term *'stationary'* was in fact somewhat imprecise in the original version of the manuscript. On several occasions, it was replaced by the term *'time-homogeneous'* in the revised manuscript. The reason for this is, that the estimation of the KM coefficients relies on the assumption that the underlying dynamics do not change over the investigated period covered by the data. That means that the dynamics must be 'time-homogeneous'.

Additionally, the estimation of the Kramers-Moyal coefficients relies on repeated return of the system to the corresponding region in state space. This is closely related to a certain degree of stationarity of the data, which we guarantee by applying a detrending in the data preprocessing.

Compare the new manuscript l. 114 and the subsequent paragraphs up to line 155.

*The analysis conducted in this work relies on the following assumptions and technical conditions:*

*(i) the data-generating process is sufficiently time-homogeneous over the considered time period;*

*(ii) the process is Markovian at the sampled temporal resolution;*

*(iii) the data is equidistant in time;*

*(iv) the relevant region of the state space is sampled sufficiently densely by the available data.*

Second, we did not claim in the original manuscript that the nonzero 4th-order KM coefficient – which we interpret as evidence for a jump-like stochastic forcing on the d18o – fully explains the regime switches. We only say that the jumps in the stochastic forcing might play a role in the regime switches and should not be discarded in the analysis.

However, as mentioned previously, we removed the one-dimensional analysis which included the assessment of higher order KM coefficients from the manuscript.

Third, the term 'fitting' usually describes a method where a model output is compared to the data. Then the model parameters are tuned such that the model would optimally approximate the data. The approach pursued in our study is different. We estimate the Kramers-Moyal coefficients directly from the data and no model-output to data comparison is required for this method; in particular, we do not minimise a distance or cost function. We deliberately refrain from presenting explicit stochastic model equations.

I'm not very convinced that the applied methodology is suitable. As far as i see, the authors do not test their null-hypothesis (H0) of a stationary process.

We fully agree with the referee that one cannot assume per se that the investigated time series are stationary and that they have been generated in a time-homogeneous process. Also, the referee is correct in the sense that we did not provide a statistical test that supports the stationarity of the investigated time series. In the revised manuscript we include the result of a statistical test that attests stationarity to the data (see Table 1).

Since the referee's general suspicion towards the applicability of the chosen method is the main criticism in his review, we will in the following give a detailed explanation why we consider our approach meaningful.

We believe that a comment the referee made in the pdf attached to his report, is helpful to understand the exact point of criticism he raised with respect to the stationarity assumption:

In line 401 of the original manuscript, we wrote:

> *'This may be the atmospheric circulation as represented by the dust proxy, or another external driver whose signature might be encoded in the higher-order KM coefficients of the δ18O.'*

which was commented by the referee with the words:

> This sounds like there is no problem with the methodology if that's the case.

This comment led us to the interpretation that the referee does not question that the climatic process giving rise to DO events can be considered stationary over the investigated time period. Instead, we understand the referee's point as follows:

The observed data is a projection of a high dimensional complex process onto the state space spanned by $\delta^{18}O$ and dust which are assumed to represent Greenland temperatures and atmospheric large-scale circulation. The applied methodology now assumes that all other degrees of freedom (or all other variables, termed 'bath') can be subsumed in an effective force and a stochastic force (i.e., noise), and be described in a SDE approach. This

subsumption certainly relies on the type of interaction of the observed variables with the bath variables (or hidden variables) and requires a corresponding separation of time scales.

We believe that the referee doubts whether the relevant dynamics that gives rise to the DO variability observed in the data at hand are fully captured in the projection onto the observed, low-dimensional subspace. We interpret his objection to our time-homogeneity and stationarity assumptions in the sense that the referee advocates for the presence of unobserved or hidden variables that cannot be subsumed in the form of a deterministic and stochastic force. Such a potential coupling to hidden variables can also be interpreted as explicit time dependencies or non-stationarities of the observed dynamics. In short, the referee doubts whether a time-homogeneous stochastic differential equation can describe the data.

For sake of clarity: In the following we will refer to unobserved variables that cannot be described as a bath by denoting them as *hidden variables*.

If the requirements for the eff. force + noise description of the unobserved variables are not fulfilled and one still tries to impose this framework – as we do – then the hidden variables which in fact have much more explicit impact on the observed variables' dynamics, still contribute to the KM coefficients and in particular to higher order KM coefficients. One would then in most cases observe that the model retrieved from the data does not fully explain the dynamics of the observed variables.

At this stage, we can make three important remarks:

1. The observations are limited to the d18o-dust space, so all we can do is try to investigate these. We cannot include further variables in our analysis, simply because there is no data.

2. Given (1) and in line with finding the simplest starting argument, it is natural to make the attempt to treat the unobserved variables as a bath and then scrutinise the consistency of the obtained KM coefficients with the observations. There will typically be some characteristics of the dynamics which are reasonably well explained by this approach, and others which are not. This is exactly the case in our study:

    a. The obtained drift explains the two regimes.

    b. The obtained drift does not explain the trends, which govern Greenland interstadials.

3. If there would be a strong coupling to hidden variables, we would expect the data to show higher degrees of autocorrelation, since typically subsuming hidden variables mistakenly in a bath gives rise to a memory term. This essentially follows from the Mori-Zwanzig formalism.

In the revised manuscript we limit the KM analysis to the leading-order coefficients, due to the scarcity of data. In our investigation we then focuses on the retrieved deterministic flow field and we do not claim to provide a comprehensive explanation for the dynamics of the coupled dust-$\delta^{18}$O system. What we do find, however, is that this flow field does not, in itself, explain the fast transitions c2w and the slow w2c transitions – here, importantly, we point to the possibility that hidden variables may play a decisive role at various points in the manuscript.

> *'However, the analysis of the vector field (Fig. 3 (b))does not indicate any clear paths the system takes in order to transition between stadial and interstadial states. The shape of the nullclines can, in principle, allow for a situation where a perturbation along the δ18O direction pushes the dust across its bifurcation point, triggering a transition of*

*the dust, which in turn stabilises the δ18O perturbation. The combined drift F (x1 , x2), however, exhibits strong restoring forces along the δ18O direction which render this mechanism rather implausible.*

*Viewed from either stable fixed point, perturbations along the dust direction could in contrast push the system across the basin boundary relatively easily. Certainly, a combination of noise along both directions may also be able to drive the system across*

*the region of weak divergence that separates the two attractors. We note that the mild relaxation that is typical for Greenland interstadials cannot be explained from results of this analysis alone. (l.260)*

The original formulation of this paragraph, which was slightly edited was commented be the referee as follows:

I'm really not convinced. I think the consideration of an important variable is missing.

The possibility that our investigation of the combined dust and d18o record misses a link to other important dimensions of the process is now explicitly formulated in a dedicated paragraph of the revised manuscript:

*'Clearly, the state space spanned by δ18O and dust is a very particular one. On the one hand the interpretation of the two proxies as indicators of Greenland temperatures and the hemispheric circulation state of the atmosphere bears qualitative uncertainties and should certainly not be considered a one-to-one mapping. On the other hand, other climate subsystems not directly represented in the data analysed here, like the AMOC for example, are likely to have played an important role in the physics of DO variability as well. Even if δ18O ratios and dust concentrations were to exclusively represent Greenland temperatures and the atmospheric circulation state, the recorded climate variables were certainly highly entangled with other climate variables such as the AMOC strength, the Nordic Sea's and North Atlantic's sea ice cover, or potentially North American ice sheet height (e.g. Menviel et al., 2020; Li and Born, 2019; Boers et al., 2018; Zhang et al., 2014; Dokken et al., 2013). In our analysis, such couplings are subsumed in the δ-correlated noise term ξ – an approach which may rightfully be criticised to be overly simplistic. However, given the lack of climate proxy records that jointly represent more DO-relevant components of the climate system on the same chronology, the chosen method reasonably complements existing data-driven investigations of DO variability. For example Boers et al. (2017) similarly examined the dynamical features of the combined δ18O–dust record. They proposed a third-order polynomial two-dimensional drift in combination with a non-Markovian term and Gaussian white noise to model the coupled dynamics. While our approach is limited to a Markovian setting, it allows for more general forms of drift (and noise). Being non-parametric, it does not rely on prior model assumptions in this regard. It is not per se clear how the couplings to 'hidden' climate variables (i.e., those not represented by the analysed proxy record) influence the presented drift reconstruction and there is certainly a risk of missing a relevant part of the dynamics.' (l.294)*

In short:

- We agree that a coupling to hidden variables which cannot be subsumed in a SDE representation can be understood as a type of time-inhomogeneity (and under certain circumstances as a non-stationarity) with respect to the dynamics of the observed variables.

- We believe the referee understands our claim that the data is time-homogeneous (in the original manuscript we mistakenly used the term 'stationary') in the sense that we categorically exclude any coupling to hidden variables. This was, however, not

our intention. We believe that this is conveyed sufficiently clearly in the revised manuscript (see citation above).

- Furthermore, in the revised manuscript we are more explicit about the assumptions underlying our analysis and about their justification. See the revised manuscript lines 114-156.

- We would also like to remind the referee of the fact that only limited data from ice cores is available for these long time periods and with sufficient temporal resolution. So all we can do is to investigate the observed variables and there is little we can do about the hidden ones – at least if we aim to stick to a methodology which is to a high degree data-driven.

I would not think that a stationary process described by the KM equations is consistent with a hypothetical nonstationary process that could not be rejected.

In this answer, we interpret the term 'stationary' used in the original manuscript and the review as to mean 'time-homogeneous'.

In principle, it is certainly true that the time homogeneous KM equation, which forms the basis of our investigation, is not a consistent framework to describe a time-inhomogeneous process. We have made an effort to rule out obvious reasons for time-inhomogeneity and possibly related non-stationarity in the manuscript:

The corresponding statement in the original manuscript

*'Excluding also the Last Glacial Maximum from the data, we restrict our analysis to the period 59–27 kyr b2k, which is characterised by a fairly stable background climate and persistent co-variability between dust and δ¹⁸O.' (l.102, original manuscript)*

Was replaced by the more precise formulation:

*'With regard to (i) a low-frequency influence of the background climate on the proxy values and on the frequency of DO events is evident (see Fig. 1), with suppressed DO variability during the coldest parts of the glacial and longer interstadials for its warmer parts (e.g. Rial and Saha, 2011; Roberts and Saha, 2017; Mitsui and Crucifix, 2017; Lohmann and Ditlevsen, 2018b; Boers et al., 2017, 2018). We therefore restrict our analysis to the period 59–27 kyr b2k, which is characterised by a fairly stable background climate and persistent co-variability between dust and δ18O (Boers et al., 2017). To remove the remaining influence of the background climate on the climate proxy records we remove a trend that is nonlinear in time from both time series. This trend is obtained by linearly regressing the proxy data against reconstructed global average surface temperatures (Snyder, 2016)' (l.119)*

After these considerations the only source of potential non-stationarity that is left is the variability of other climatic subsystems (e.g. the AMOC) which are potentially coupled to Greenland temperatures and atmospheric circulation investigated in this manuscript, that is the coupling to hidden variables.

Finally, it should be mentioned that several influential investigations of the same data have followed a similar reasoning, that is, they rely to some extent on the assumption that the data generating process can be described by autonomous model equations that comprise only the observed variable and no hidden variables. A small selection is:

Ditlevsen, P. D. Observation of α-stable noise induced millennial climate changes from an ice-core record. Geophys. *Geophys. Res. Lett.* **26**, 1441–1444 (1999).

Boers, N. *et al.* Inverse stochastic-dynamic models for high-resolution Greenland ice core records. *Earth Syst. Dyn.* **8**, 1171–1190 (2017).

Kwasniok, F. Analysis and modelling of glacial climate transitions using simple dynamical systems. *Philos. Trans. R. Soc. A Math. Phys. Eng. Sci.* **371**, (2013).

Livina, V. N., Kwasniok, F. & Lenton, T. M. Potential analysis reveals changing number of climate states during the last 60 kyr. *Clim. Past Discuss.* **5**, 2223–2237 (2010).

As mentioned previously, we have supplemented the manuscript with tests for stationarity in (see Table 1).

Say, we have a nonstationary (we intepret this as time-inhomogeneous or non-autonomous) Orntsetin-Uhlenbeck process of $dx/dt = -a*x + B(t) + c*xi(t)$, (OU) where xi is white noise, and $B(t) = b*sin(sin(2*t)+t)$ is some regular nonstationarity (time inhomogeneity). It mimics some regime behaviour with sudden and regular transitions. We can easily see that the pdf of x is bimodal. If we didn't know the underlying process generating eq., and perhaps we somehow overlooked the regularity of the transitions, we might think the underlying model is:

$dx/dt = F*x + c*xi(t)$, (H0)

where $F = -V(x)$, $V(x)$ being a double-well potential function. If we are in the small noise limit, we know that the pdf takes the shape of $V(x)$, so, we could estimate V that way. Furthermore, we can estimate the noise strength 'c' in some standard way too.

Now the question is if it matters at all that we have an H0 other than the true process OU. It would not matter if in any appreciable way the processes perform the same, i.e., when, loosely speaking, they are consistent or approximately equivalent. For example, we can derive the probability distribution of residence times from H0, and perform a statistical test if our residence time data is consistent with that, or we can reject H0.We want to perform a so-called crucial experiment (experimentum crucis).

In order to explain the observations, it is a natural approach to start with the simplest model, which in this case is a time homogeneous (stochastic) process model. Previous studies (see citations above) have investigated the performance of a 1D double-well potential in the presence of (Gaussian or alpha stable) noise or 2D relaxation oscillator models of vdP or FHN type and concluded that these models do explain several characteristics of the record, while they fail to explain others. Our investigation ties in with this branch of research: by using a non-parametric estimation of the KM coefficients, we loosen the assumptions previous studies made on the structural form of drift and diffusion. We hence adopt the next complicated modelling approach. Other studies decided to stick to simple model equation but introduce explicit time dependencies as for example:

Roberts, A. & Saha, R. Relaxation oscillations in an idealized ocean circulation model. *Clim. Dyn.* **48**, 2123–2134 (2016).

Mitsui, T. & Crucifix, M. Influence of external forcings on abrupt millennial-scale climate changes: a statistical modelling study. *Clim. Dyn.* **48**, 2729–2749 (2017).

Doing both, allowing for explicit time dependence while at the same time giving on the assumptions on the structural form of drift and diffusion would of course be desirable but is not possible due to the scarcity of data.

Considering H0 of the authors, another likely feature based on which H0 can be rejected is the saw tooth asymmetry, in particular, that the cold to warm, c2w, transitions are much more rapid than the warm to cold, w2c, ones.

The revised manuscript focuses solely on the two-dimensional analysis. In two dimensions the drift alone can break time-asymmetry and generate a saw-tooth shaped trajectory (in

either of the dimensions). We admit, that the reconstructed drift does not seem to break time asymmetry, at least not in an obvious manner - but this discrepancy is also explicitly stated in the revised manuscript:

> 'We note that the mild relaxation that is typical for Greenland interstadials cannot be explained from results of this analysis alone.' (l.267)

It shall be mentioned that even in one dimensions, the asymmetry of the data would not per se rule out a time-homogeneous SDE model:

One can easily break the time symmetry by introducing a discontinuous element, as for example

dx(t) = - a(x)dt + bdW(t) + cdJ(t)

where J(t) is a Poisson process with a jump rate λ>0. In this simplest of formulations, if there is at least 1 jump from the Poisson jump process, the process becomes time-asymmetric.

Looking at the dust time series of Fig. 1 with much naivity wrt. physics, but with some experience about dynamical systems, I would think that c2w is an attractor crisis, whereas w2c is a noise induced tipping, and there is some slowly drifting control parameter, i.e., a nonstationarity when we exclude that parameter from our state variables.

The authors also make a reference to attractor crisis, in terms of a saddle-node bifurcation, but in some other context. It is the context of slices of a 2D potential function. This does not sound correct.

In the discussion of the coupled two-dimensional dynamics, we refrained from using the notion of conditioned potentials in the revised version of the manuscript. Though, we believe it is still useful to discuss the two components of the drift (along the d18o and along the dust direction) separately as functions of the two-dimensional space spanned by d18o and dust. We justify this in the revised manuscript as follows (L. 308 XXX):

> 'We will first discuss the two drift components D 1,0 and D 0,1 (see Eq. (5)) separately as functions of the two-dimensional space spanned by δ18O ratios and dust concentrations. In the component-wise analysis, the analysed component takes the role of a dynamical variable, while the respective other assumes the role of a controlling parameter. In this setting, corresponding nullclines can be computed, which reveal the bifurcation and stability structure of the two individual drift components. Intersections of the two components' nullclines yield fixed points of the coupled system, which are stable if both nullclines are stable at the intersection.' (l.205)

Regarding the dust-nullcline we stick to our assessment, that it shows the typical structure of a double-fold bifurcation parametrized by $δ^{18}O$. We have made several changes to Sect. 4.1 of the revised manuscript in order to convey this interpretation more convincingly.

After all, studying the drift component's nullclines we find the hypothesis that the bistability is rooted in the Greenland temperatures (represented by d18o), which then drives the dust variable as a dependent variable, is highly unlikely. In contrast, from the dust-nullcline's shape we can conclude that the dust — and therefore the climate variable(s) it represents — remains a potential candidate for accommodating the bistability of the Arctic climate system during the last glacial.

The paper is very well written in a way, but it doesn't make for a very pleasant reading journeying through flawed results, starting with the single variable approach, and then — at least as i suspect — even the 2 variable approach.

I attach the pdf of the manuscript with comments saved as annotations. Hopefully the authors find it useful in some way.

Note: I always review non-anonymously, and never make recommendation for or against publication. The recommendation that i make is only to circumvent the rigidity of the submission system, and therefore please consider it void.

Tamas Bodai

Please also note the supplement to this comment:

https://esd.copernicus.org/preprints/esd-2021-95/esd-2021-95-RC2-supplement.pdf

We thank the reviewer for his valuable comments. We agree that, particularly using proxy data in the context of paleoclimatology, much of the outcome is subjected to limitations and imperfections. Still, we believe our revisions will take this into account thoroughly and provide important new conclusions e.g. about how the two variables, dust and δ¹⁸O, are coupled. As discussed in the answer to the other reviewer, the widespread assumption that the bistability is rooted in the temperature, which then drives the dust variable as a dependent variable, seems unlikely from our results.

The particular aspect of including both 1D and 2D approaches follows from the fact that the 1D approach is a natural starting point for our investigation also considering other recent studies on date-driven model inference from Greenland ice core time series, and provides the reason why the 2D analysis is necessary, therefore motivating the second step of analysis.

**Specific Comments**
* * *
l.12     (3) the δ¹⁸O record is discontinuous in nature, and mathematically requires an interpretation beyond the classical Langevin equation.

A discrete time sampling cannot result in a continuous series. Also, is this new result?

The revised paper is restricted to the analysis of the coupled two-dimensional drift and hence the above statement was removed.

l.32     The most prominent example of past abrupt climate shifts are the Dansgaard–Oeschger events

maybe one of the most

We agree, however, the sentence was removed in the revised version of the manuscript.

l.34     a series of sudden warming events that dominated Greenland temperatures throughout the last glacial cycle,  (Johnsen et al., 1992; Dansgaard et al., 1993; North Greenland Ice Core Projects members, 2004).

(e.g., Refs was stroke through by the referee.)

We have changed this to:

*[...] a series of abrupt warming events over Green land first evidenced in stable water isotope records from Greenland ice cores (Dansgaard et al., 1982, 1984, 1993; Johnsen et al., 1992; North Greenland Ice Core Projects members, 2004). (l.38)*

l.57 The key concept is to regard the paleo-climate record as the realisation of a Markovian and stationary stochastic process (Kondrashov et al., 2005, 2015) which can be described in terms of a stochastic differential equation.

What if the sudden warming is a bifurcation

The question is, what would be the control parameter. This could either be a variable internal to the climate system, which is coupled to the recorded d18o and dust, or it could be a control acting from outside of the climate system.

Regarding the latter, slow changes in the orbital parameters are supposed to influence DO variability (Mitsui and Cruzifix 2017, Roberts and Saha 2016) but there are no signs for these changes to trigger DO events, which leaves us with the first option. If another climate-internal variable was acting as a bifurcation parameter on the d18o and the dust, then this unobserved variable was in fact coupled to the observed variables. For the lack of data (on the same chronologie) we cannot investigate these coupling within this study. Thus, we investigate here the coupling between dust and d18o in the stochastic process setup, and do in fact find indication for a double fold-bifurcation in the dust controlled by the d18o (see section 4.1).

l.63 The Kramers–Moyal equation generalises the Fokker–Planck description of stochastic processes, including explicitly the presence of discontinuous elements.

Is this equivalent with a nonstationary framework? I suppose not.

The revised paper is restricted to the analysis of the coupled two-dimensional drift and hence the above statement was removed.

l.67 In Sec. 2 we introduce the paleo-climatic proxies under examination and the detrending method used to ensure that the data is approximately stationary.

Is this correct methodology? Why would the detrended process not be nonstationary? I.e., is stationarity really "ensured"?

We thank the referee for this comments. In the revised manuscript the statement does no longer appear as such but now reads:

*'We first present the paleoclimate proxies analysed in this study and explain how we pre-processed the data to make it suitable for estimating Kramers-Moyal coefficients (Sect. 2)' (l.75)*

Content-wise, the question raised within this comment is best answered in our explanations with respect to the data detrending:

*'After the detrending, all stadial (resp. interstadial) periods exhibit almost the same level of values, which allows considering the data as the outcome of a time-homogeneous and stationary process (compare Fig. 1 panels (f) and (g)). Levelling out the differences between the recurring climate periods guarantees a sufficiently dense sampling of the relevant region of the state space (iv) and prevents a blurring of the drift reconstruction (i).' (l.137)*

Finally, see Table 1 for the question if stationarity is fulfilled.

l.72 This is consequently discussed in Sec. 4.2, where we uncover the conditioned potential landscapes of the joint proxy process.

conditioned on what?

Conditioned on the respective other variable. However, in view of the comment by the other referee, we have abandoned the notion of conditional potential anyways and removed this sentence accordingly.

I.109  To assess whether the data is Markovian, we analyse the auto-covariance function of the increments of the detrended data.

Nonstationarity overlooked this way.

Please see our detailed answer to the referee's general comment.

I.121  A prominent example for a stochastic process is given by the stationary Langevin equation

I thought in a Langevin eq. you have the differential on the left. This looks more like Ito or Stratanovic, and you should actually specify which one.

There is no need to state which formulation is given. The evaluation under Itô or Stratonovich affects solely the evaluation of the drift by adding a spurious term affecting Eq. (4). Since we explicitly give the relation between the first and second KM coefficient in relation to a(x) and b(x) in Eq. (4), we are assuming Itô.

Note, however, that the term 'stationary' was replaced by time-homogeneous in the revised manuscript.

I.124  If the properties of the dynamics do not change over time, i.e. a(x) and b(x) do not depend on time, these processes are called stationary.

This is what i would contest.

In fact, the statement in the original manuscript was wrong and therefore deleted. The process is time-homogeneous, but not necessarily stationary.

I.125  While the Langevin equation is continuous in time, stochastic processes can in principle have discontinuous features, such as sudden jumps.

But for what value of dt? var[dB] goes to 0 as dt does.

We have added an Appendix E in order to clarify what discontinuity is, independent from the necessity of square integrability or vanishing second moment.

I.126  An easy way to incorporate discontinuities is to include in Eq. (1) an elementary Lévy process L(t), modulated with an amplitude h(x) (Applebaum, 2011)

Isn't it a bit of an interpretational issue? var[dL] does not exist. But then what' in the limit of 0 for dt?

Please see the answer above.

I.135  If a single particle's motion is governed by the Langevin equation, its probability density function p(x, t|x ′ , t ′ ) evolves according to the Fokker–Planck equation,

Does the FP not govern the unconditional density rather?

Both forms exist. See Risken (1996) eq. 4.16 for a conditional density formulation (with the general Kramers–Moyal operator), or eq. 4.52 and 4.53 for solely the conditional density formulation of the Fokker–Planck equation.

l.140 So, this should tell whether (1) is Ito or Stratanovic. Please spell it out. My reference is Risken's book for this.

The formulation is given in Itô. Please see our aforementioned answer to the comment on l.121.

l.145 Giving up the condition of continuity, the temporal evolution of the conditional probability density follows the Kramers–Moyal equation

Can you please indicate here how discontinuity is allowed?

The revised paper is restricted to the analysis of the coupled two-dimensional drift and hence the above statement was removed.

l.153 However, for numerous of these stochastic processes, the KM coefficients can be related to the properties of the stochastic process in the spirit of Eq. (4).

Provide a reference please.

The revised paper is restricted to the analysis of the coupled two-dimensional drift and hence the above statement was removed.

l.168 To retrieve the KM coefficients $D_m(x)$ from a single realisation of a stochastic process, i.e. a single time series, we evaluate the transition probability densities in the limit of a vanishing time step $\tau \to 0$, which numerically corresponds to considering the shortest increment $\Delta t$ in the data ($\tau \to \Delta t$).

Is it so? I suspect that it might be but might be not. It can be that you are trying to do your estimation of a process, which requires a higher resolution of data.

Essentially, you might be unable to perform the estimation (even if the modeling assumption was completely perfect). What you want to see perhaps is that as you reduce the time resolution, your estimate seems to converge. I.e., you might need to deliberately coarsen your resolution.

Below we include a reploting of Fig. 2 in the manuscript wherein we coarsen the resolution $\tau$ from the original 1 to 2 and 3 (i.e., we consider only half and a third of the total amount of data). The overall shapes remain unaffected. Bandwidth selection of the Nadaraya–Watson estimator still follows Silverman's optimal bandwidth selection, using an Epanechnikov kernel. The total amount of data-points considered are: $\tau=1$, 6399 data-points; $\tau=2$, 3199 data-points; $\tau=3$, 2133 data-points.

We note that following our observation from the auto-correlation of the increments of the time series, seen in Appendix B, Fig. B1, we can safely claim we are above the Einstein–Markov length and thus these estimates are not affected by microscopic noise correlation and subsequently taking coarser resolutions yield identical results (just as shown), which are only affected by having fewer data-points from which to draw the estimates.

[Figure]

Note that this figure is not included in the revised manuscript.

I think it is not a good procedure to perform the estimation and then generate ample synthetic data by the assumed model and check if estimates at a certain resolution are biased.

We assume the referee inserted the 'not' inadvertently and hence we interpret this comment as if the referee suggested applying the above procedure. In this case our answer would be as follows: To the best of our understanding, this is not something we can perform in our evaluation: First, we can naturally estimate all statistical moments of the time series, but to generate synthetic data we have to make an assumption on the underlying process. For instance, the Tabar Q ratio presented in Figure 3 (removed from the revised manuscript) shows the presence of discontinuities or jumps in the process. We refrain from formulating a specific model for these jumps on the basis of the limited data we have. Our paper goes to great length to avoid precising a model – that we believe is one of the strengths of the paper. Thus, unless we constrain ourselves to a model, we cannot do this.

If the 'not' was not a typo, then we do not fully understand this comment as we do not simulate any synthetic data.

It is possible that at the biased estimates the biases are indicated low, but at the true "unbiased" values estimates are very biased at that time resolution. It might sound circular, but it isn't, i believe.

l.178    Thus, values of the ratio D 4 (x)/D 2 (x) close to zero imply continuous sample paths with no jumps in the data.

The estimate will be a finite nonzero number. How do i know if this is small, close to zero, or not? What should i compare it with?

The revised paper is restricted to the analysis of the coupled two-dimensional drift and hence the above statement was removed.

l.183   This assessment can be refined by regarding the Lehnertz–Tabar Q-ratio (Lehnertz et al., 2018), which takes advantage of the fact that continuous and discontinuous systems 'scale' in a different fashion. While a purely continuous stochastic process diffuses proportionally to time t (or possibly a power of time t^β in anomalous diffusions (Einstein, 1905; von Smoluchowski,1906; Havlin and Ben-Avraham, 1987)), discontinuous processes can cover large distances in short times, i.e. jump, which causes them to exhibit no scaling relations with time t.

I don't know what this means.

The revised paper is restricted to the analysis of the coupled two-dimensional drift and hence the above statement was removed.

l.205   For various applications where the fluctuations are not comparable in size, i.e. where the diffusion elements are not of similar scale, one can draw a clearer picture of the motion of the two-dimensional system by referring to an effective vector field

Reference please.

In fact, we do not use the 'effective vector field' anymore in the revised version of the manuscript.

l.211   Similarly to the one-dimensional case, one can obtain potential landscapes as integrals over the two drifts:

Reference please. How is the potential defined over a multidimensional phase space? I think there is only one potential function, not one per variable! See p. 133-134 of Risken, or https://iopscience.iop.org/article/10.1088/1361-6544/ab86cc e.g. eq. (3)

But mind that you can only say that you found a potential from obs data if your modelling assumptions are correct.

In a two dimensional state space one can consider the dynamics along a single dimension while hypothetically freezing the motion along the other dimension. This leads to a 1D setting, where the typical notion of a potential applies.

However, we decided to abandon the notion of the conditioned potentials and have therefore reformulated statements and the equations accordingly. Since, there are numerous changes with respect to this issue, we do not list them here, but instead refer to the tracked changes file.

l.228   We find the second KM coefficient to be fairly constant (Fig. 2 (c)) and the ratio between fourth and second KM coefficients to be negligible (Fig. 2 (d)), which suggests that a Langevin process with additive noise is a viable description of the isolated dust dynamics.

I cannot judge whether this is large or small.

The revised paper is restricted to the analysis of the coupled two-dimensional drift and hence the above statement was removed.

l.233   Note that the model equations employed here are by construction symmetric with respect to time, therefore, as it is, the model cannot reproduce the temporal asymmetry that is visually suggested in the dust record.

This should raise concern about the modeling assumption.

The revised paper is restricted to the analysis of the coupled two-dimensional drift and hence the above statement was removed.

l.236    Most prominently, the drift has only a single stable fixed point (zero-crossing of the drift), or equivalently, the potential function exhibits only a single well.

The drift implies a fixed point, maybe. Terminology.

The revised paper is restricted to the analysis of the coupled two-dimensional drift and hence the above statement was removed.

l.240    Moreover, we find that the fourth KM coefficient $D_4(x)$ for the $\delta^{18}O$ is of the same magnitude as the second KM coefficient $D_2(x)$ (Fig. 2 (h)).

I see that D4 is compared to D2 actually. But what does comparability, a ratio of 1, mean? Still, is this much in some sense?

The revised paper is restricted to the analysis of the coupled two-dimensional drift and hence the above statement was removed.

l.239    Given the high correlation between the dust and the $\delta18O$ records, the differences in the reconstructed potentials and the ratio between fourth and second KM coefficient are remarkable.

Yes, but in the same time we can just plot the histograms of dust and d18O and we will see a bi- and unimodal distribution, no? IF the 1D models were correct, and the noise was really small, then the histograms give you the respective potential functions.

Maybe thought the very correlation bw. dust and d18O prompts that the 1D models cannot be good assumptions.

One needs to somehow test the hypothesis of the model. Checking Markovianity in the appendix is probably very insufficient.

The revised paper is restricted to the analysis of the coupled two-dimensional drift and hence the above statement was removed.

l.243    At first sight, the monostability of the reconstructed $\delta^{18}O$ potential contradicts the apparent two regime nature of the time series.

Is the histogram bimodal? Bimodality would imply regime behaviour (of whatever origin), but regime behaviour does not imply bimodality of the hist'.

We fully agree with the referee. The histogram of the $\delta^{18}O$ is in fact monomodal. However, a two-regime nature of the record can indeed be evidenced by eye. In the revised manuscript we now introduce the term two-regime nature with a bit more of explanation (l.111):

*'The two-regime character of the time series translates into a bimodal histogram of the dust data, as seen in Fig. 1 (h). In the case of the $\delta^{18}O$ data, the stronger trend during interstadials and the higher relative noise amplitude masks a potential bi-modality and the histogram appears unimodal.'*

l.258 In Fig. 3, we clearly see a constant relation of $Q(x, \tau)$ with respect to $\tau$ for the $\delta^{18}O$ record, suggesting that this stochastic process includes jumps.

Or that your assumption of stationarity is crucially wrong. We want certainty. There is little value in evidence alone that can imply two very different things. We want to know which one is true.

The revised paper is restricted to the analysis of the coupled two-dimensional drift and hence the above statement was removed.

l.274 The reconstructed conditional potential $V_{1,0}(x_1|x_2)$ of the dust is displayed in Fig. 4 (d). As a conditioned potential, it can be read by taking vertical 'slices' of the potential.

You haven't defined that. And so we don't see that the 1D potentials are really slices of the one multivariate pot'.

As mentioned previously, we have abandoned the notion of conditioned potentials, and hence reformulated the above statement accordingly and it now reads (l.212):

*'The estimated dust-drift D 0,1 (x 1 = δ 18 O, x 2 = dust) is displayed in Fig. 3 (c). This coefficient dictates the deterministic motion of the system along the dust direction; therein the δ 18 O ratio takes the role of the controlling parameter.'*

l.290 However, the position of the minimum δ 18 O* appears to be determined by the dust in a continuous manner, with high rate of change for intermediate dust values whilst no change for more extreme dust values.

What does this refer to?

We meant to say that for intermediate values of the dust a small change in the dust – seen as a control parameter – causes a fairly strong change in the position of the minimum of the conditioned $\delta^{18}O$ potential (high rate of change).

As we abandoned the notion of conditioned potentials, we have rephrased the above statement as follows (l.229):

*'The position of the fixed point changes with the value for dust in a continuous manner, with a high rate of change for intermediate dust values and small change for more extreme dust values.'*

l.297 These findings are consistent with the observed regime switching of both records, which we struggled to reconcile with the results obtained from the one-dimensional analysis.

Perhaps there is an amount of hindsight in this but this is exactly how time series would look like when we have a 2D double well potential with two axes of symmetry.

Do not define the variables by the axes of symmetry, and you will see coordinated transitions. Furthermore, if you tip your coordinate system only slightly, closing small angles with the axes of symmetry, then you have the chance that despite the transitions, the marginal distribution of one of the variables will be still unimodal.

However, such a system would not feature asymmetry in that you have a more sudden transition in one direction than the other.

We fully agree with the referee's comment which is hardly a criticism but rather a suggestion: We have included an analysis of the two-dimensional drift in a rotated frame to provide evidence that the data cannot be described by a two-dimensional double well potential with two axes of symmetry. Instead, the coupling between the two dimensions persists, even after a suitable rotation. See Section 4.4 of the revised manuscript.

l.323    This leaves open the possibility that transitions between stadial and interstadial states are mainly induced by noise as argued by, e.g., Ref. (Ditlevsen et al., 2007) (i.e. noise-induced tipping), facilitated by a shallow potential barrier close to the minima of the (effective) vector field.

I'm really not convinced. I think the consideration of an important variable is missing. It's worth considering this recent paper:

https://link.springer.com/article/10.1007/s00382-020-05476-z

The sawtooth feature is there, for one thing. But my feeling is that the rapid transition is rather due to crisis, the disappearance of a regime, as a result of nonstationary dynamics (nonstationary in a reasonable modeling framework).

Please see our detailed reply to the referee's general comment. However, we have changed the above statement as follows (l.261):

*'Viewed from either stable fixed point, perturbations along the dust direction could in contrast push the system across the basin boundary relatively easily. Certainly, a combination of noise along both directions may also be able to drive the system across the region of weak divergence that separates the two attractors. We note that the mild relaxation that is typical for Greenland interstadials cannot be explained from results of this analysis alone.'*

l.358    In particular, we have shown that the $\delta$ 18 O record cannot be treated as a time-continuous process.

If the fast transition is attractor crisis, then it is a continuous process. You can only make this statement if the modelling assumption was water tight. But it isn't. So, strictly speaking, you have not shown what you say.

The revised paper is restricted to the analysis of the coupled two-dimensional drift and hence the above statement was removed.

l.364    In principle, the revealed double-fold bifurcation would allow for bifurcation-induced transitions and thus for a limit-cycle behaviour.

What you found is not what you say. I think the methdology of (13) is problematic.

We are not sure if we fully understand this comment. We have not used (13) in the revised manuscript and instead now explain the presence of the double-fold bifurcation in terms of the nullcline the of dust in the $\delta^{18}$O, dust state space (Sect. 4.1).

*'This coefficient dictates the deterministic motion of the system along the dust direction; therein the $\delta$ 18 O ratio takes the role of the controlling parameter. We can trace the nullcline's branches which take a general s-shape as we vary $\delta$ 18 O.'* (l.212)

The possible transition mechanisms are discussed later. In the revised manuscript we do not mention a potential limit-cycle behaviour anymore.

'*The shape of the nullclines can, in principle, allow for a situation where a perturbation along the δ 18 O direction pushes the dust across its bifurcation point, triggering a transition of the dust, which in turn stabilises the δ 18 O perturbation. The combined drift F (x 1 , x 2 ), however, exhibits strong restoring forces along the δ 18 O direction which render this mechanism rather implausible. Viewed from either stable fixed point, perturbations along the dust direction could in contrast push the system across the basin boundary reltively easily. Certainly, a combination of noise along both directions may also be able to drive the system across the region of weak divergence that separates the two attractors. We note that the mild relaxation that is typical for Greenland interstadials cannot be explained from results of this analysis alone.' (l.262)*

l.372    We conclude therefore that – based on our results – the DO transitions are to large degree induced by noise, acting on the background of a double-fold dynamics governing the dust, for which the δ18O acts as control parameter.

I'm not convinced at all, sorry. How could the faster dynamics control the slower like this?!

We agree that this statement was too strong and is not included as such in the revised manuscript.

l.373    These findings do not contradict previous studies that proposed limit-cycle models to explain the δ18O record, since the cyclic motion was not expected to happen in a state space comprised of Greenland temperatures and atmospheric large scale circulation (Kwasniok, 2013; Lohmann and Ditlevsen, 2019).

I think your finding is the result of your methodology. It is your methodology that should/could be contradicted. Consider the possibility that the cited papers do contradict your methodology.

I'm not sure how, but some statistical test might be able to reject your null-model. Perhaps the saw-tooth asymmetry feature can be a basis of such a formal, precise test.

Or maybe another test is whether the residence time data is consistent with the predictions of the fitted model. Of course, i mean a formal hypothesis test again.

There is really no value in further considering models that can be rejected by data outright.

A question is whether negative results should be published.

Many studies experiment with simple models in order to investigate Dansgaard–Oeschger variability. Some of those models – as a double well potential plus noise (Lohmann, 2018; Kwasniok, 2013; Livina, 2010) – can *a priori* be seen to not capture essential features of the NGRIP δ¹⁸O time series. Nevertheless assessing their performances, and finally deciding they would not explain the phenomena under study is a valuable contribution.

We acknowledge that the referee does not believe that DO variability can be explained in the dust-δ¹⁸O state space and we will emphasise the possibility of missing a crucial dimension of the dynamics in our analysis in the revised version of the manuscript.

'*It is not per se clear how the couplings to 'hidden' climate variables (i.e., those not represented by the analysed proxy record) influence the presented drift*

*reconstruction and there is certainly a risk of missing a relevant part of the dynamics.'*
*(l.308)*

I think it depends on how nontrivial they are. So, it should be considered by the authors that looking at Fig. 1 a stationary model is trivially wrong.

We have already explained why even an autonomous 1D model is not trivially wrong. Now that the focus of the manuscript is on the 2D drift, the model equation (4) is likewise not trivially wrong, but instead a reasonable starting point for a data-driven investigation.

l.416    Our analysis considered only a two-dimensional projection of the very high-dimensional dynamics and can therefore not be expected to deliver all details of the triggering mechanism.

Regarding the merit of a study, we can consider the question: can attractor crisis be modelled by a stochastic process that features some large jumps?

We do not fully understand the question. An attractor crisis as a potential trigger of c2w transitions would in first place need a higher dimensional state space. However, as already mentioned, we aimed to carry out a data-driven study, but no additional data is available for the investigated time period at the required resolution.

l.417    Neither the ocean-focused self-sustained oscillation hypothesis, nor the idea that the atmosphere acts as a trigger, can be ruled out based on our findings.

Isn't this a problem?! Wouldn't it be a problem that your paper did not reduce uncertainty about which one it is?!

Of course, we had hoped that our investigation would more clearly point to either the one or the other direction. However, the results are as they are and still contain valuable information. The above statement does not appear as such in the revised manuscript.

l.422    Finally, our study underlines the need for higher resolution data, as the scarcity of data points is a limiting factor for the quality of non-parametric estimate of the KM coefficients.

It would have been good to see in an appendix the dependence of estimates on time resolution.

We assume the referee refers to his comment made in the general comments section:

Essentially, you might be unable to perform the estimation (even if the modeling assumption was completely perfect). What you want to see perhaps is that as you reduce the time resolution, your estimate seems to converge. I.e., you might need to deliberately coarsen your resolution.

Please see the 'General Comments' section for our answer.

Answers: green
Comments by reviewer: violet

Comment on esd-2021-95

Peter Ditlevsen (Referee)

Referee comment on "Changes in stability and jumps in Dansgaard–Oeschger events: a data analysis aided by the Kramers–Moyal equation" by Leonardo Rydin Gorjão et al., Earth Syst. Dynam. Discuss., https://doi.org/10.5194/esd-2021-95-RC1, 2022

First of all, we would like to thank the referee for his careful and constructive review and the overall positive feedback. We have in fact adopted most of his suggestions and are convinced that this has substantially improved the manuscript. The changes that we have made to our manuscript will become clear from our point-by-point answers to the referee's comments below.

In this paper two paleoclimatic ice core records are analyzed. These, the water isotope, d18O, and the dust concentration records are analyzed for the period 59-27 kyr BP, which is the glacial period dominated by regular occurrences of Dansgaard-Oeschger events. There are two major points in the paper: Firstly, the data are modelled as a stochastic process using the Kramers-Moyal equation to investigate the importance of (discontinuous) jumps in the noise. Secondly, the two records are modelled as a two-dimensional joined process.

It seems to me that the two points are only loosely related, and the authors could consider presenting them in two separate papers.

I enjoyed reading the paper and find it publishable. However, there are a few issues below calling for revisions before publication. I have two major concerns regarding the two parts, and some minor points: As to the first point, I have not seen the Kramers-Moyal (KM) equation applied to these data before, so this is a novel approach. The equation, for which the Taylor expansion of the conditional probability density function is taken to higher order than two, covers the case where the noise term in the governing Langevin equation is not gaussian, but contains jumps. It is stated that in the case of Levy processes the Fokker-Planck equation does not apply.

Actually, for the most relevant class of Levy processes, the alpha-stable Levy processes an extension of the Fokker-Planck equation based on the characteristic function of the alpha-stable process exists (see: Samorodnitsky and Taqqu (1994) 'Stable non-gaussian processes, Chapman and Hall, NY. or Ditlevsen, PRE, 60, 172-179).

We thank the reviewer for this comment. In the revised version of the manuscript we have entirely removed the discussion surrounding Lévy processes. We are currently working on this topic and we plan to present a more detailed analysis in a separate article.

The challenge in applying the KM equation is the estimate of the higher order coefficients (eq. 6) for the data series: Since the higher order terms are (increasingly) dominated by the extremes in the increments the finite time series very quickly "dilutes". A main result (eq. 9 and figure 3) includes the sixth moment of the increments. I thus miss an analysis of uncertainties and reliability in these estimates. I find that this is essential for publishing this (nice!) result.

We agree that the inclusion of higher-order KM coefficients would necessitate a deeper analysis of the uncertainties. We have decided to relegate this analysis to a future work and focus on the deterministic part of the dynamics for the time being.

Another point which could be given a little more attention is the fact (as also correctly stated) that the strong time-asymmetry in the data (the sawtooth shape) cannot be captured by the model. How does this influence the relevance in including higher order terms (higher than second order) in eq. 5?

This is an excellent question. Notably, the analysis of the noise and the higher-order KM coefficient is mostly disconnected to the result on the deterministic aspects covered by the leading KM coefficients such that we decided to focus on the latter aspect for the time being. The asymmetry is especially hard to capture, such that comprehensive analysis of the noise will require substantial additional work.

As to the second point, the major results are presented in figure 4. Obviously, when considering a one-dimensional record, the drift can always be seen as a result of a potential. This is not the case in two - and higher dimensions, where gradient drift is a non-generic case. I'm sure that the authors are aware of this, the drift is a two-dimensional flow field, as also shown in the small inserts in the subplots of figure 4. I find the construct of pseudo-one-dimensional potentials ($V(x\_1|x\_2)$) both confusing and useless. I suggest that the authors consider abandoning this all together (as well as the notion of a potential landscape). The interpretation in figure 4(c) of a double fold bifurcation is obscure, and -I believe- wrong.

We thank the reviewer for the comment. We have removed the use of "pseudo one-dimensional" potentials. We have adopted the more accurate term "nullcline" when referring to the stable and unstable branches in the 2D drifts.

Regarding the dust-nullcline we stick to our assessment, that it shows the typical structure of a double-fold bifurcation parametrized by $\delta^{18}O$. Given the referee's comment on Figure 4c (now Fig. 3c) of the original manuscript (together with a similar comment by the other referee) we have made several changes to Sec. 4.1 in order to convey this interpretation more convincingly. Therein we discuss the finding to a greater extent

There remains of course the possibility that the double-fold structure that we observe is spurious and simply an artefact arising due to the scarcity of data (especially in the region of the state space where we observe the saddle-node bifurcations). In Sec. 4.4 we follow the Reviewer's suggestions an introduced a rotation to the state space which shows that it is not possible to linearly decouple the two proxy variables

Together – that is using the proxy data directly and in a rotated space – these findings make us confident that the interpretation of Fig. 4c (now Fig. 3c) as a double-fold bifurcation in the dust (conditioned on $\delta^{18}O$) is meaningful. After all, studying the drift component's nullclines we find the hypothesis that the bistability is rooted in the Greenland temperatures (represented by d18o), which then drives the dust variable as a dependent variable, is highly unlikely. In contrast, from the dust-nullcline's shape we can conclude that the dust — and therefore the climate variable(s) it represents — remains a potential candidate for accommodating the bistability of the Arctic climate system during the last glacial. This result is consistently found in the one and two-dimensional analysis.

What I find interesting is the scatter plot in 4(a), which nicely explain the results in figure 2, namely that the stationary distribution for the dust is bimodal while it is unimodal for the d18O: This corresponds to the marginal distributions in 4(a) (projections onto the axis.

The authors could consider analyzing a rotation (linear combination of the two variables) of the data along an axis connecting the two maxima (GS and GI) and a perpendicular direction. In this way one would obtain a "clean" two state dynamics and a "clean" one state perpendicular dynamics. First thing would be to check for independence. Just a suggestion.

We thank the reviewer for the suggestion. As mentioned above, we have included a new sections, Sec. 4.4., wherein we include an identical KM analysis to a rotated two-dimensional system. We note that the coupling of the two rotated variables is *not negligible* and supports the existence of the double-fold structure we have previously presented – and continue to argue for.

**Minor points:**
* * *
Introduction: These data have been analyzed over many years and a lot is known. For a better overview and setting the present work in context, a representative presentation of work done over the years would be useful. There is a strong bias towards very recent publications.

We agree with the reviewer that there is indeed a bias towards the works in the more recent past in the original manuscript. In response to the this, we have added Ganopolski's *Abrupt Glacial Climate Changes due to Stochastic Resonance,* 2002, behind 'Different patterns of interaction between the AMOC, sea ice cover, the Northern Hemisphere atmospheric circulation, and even the continental ice sheets have been proposed and explored to explain the emergence of DO cycles'. For the context of our study, work that approaches Greenland ice-core proxy time-series from a stochastic process perspective is most relevant. Here we identified Ditlevsen, *Observation of $\alpha$-stable noise induced millennial climate changes from an ice-core record,* 1999, Livina *et al., Potential analysis reveals changing number of climate states during the last 60 kyr,* 2010, Rial *et al., Modeling Abrupt Climate Change as the Interaction Between Sea Ice Extent and Mean Ocean Temperature Under Orbital Insolation Forcing,* 2011, Kwasniok, *Analysis and modelling of glacial climate transitions using simple dynamical systems,* 2013, Mitsui *et a., Influence of external forcings on abrupt millennial-scale climate changes: a statistical modelling study,* 2017, Boers *et al., Inverse stochastic--dynamic models for high-resolution Greenland ice core records,* 2017, Lohmann, *A consistent statistical model selection for abrupt glacial climate changes,* 2018, and Hassanibesheli *et al., Reconstructing complex system dynamics from time series: a method comparison,* 2020.

Again: I'm not fan of the "potential landscape" metaphor.

This has been removed entirely in our revised manuscript.

Figure 1: I suggest to plot the dust record upside-down. This will visualize the strong dependence between the two records, and also make the saw tooth shape in the dust record much more apparent. Make the figure full text width, Ylabel: ln(dust) (no units), d18O (permil). Or normalized w.r.t. std. dev.

We thank the reviewer for the suggestion. In fact, we meant to do this already in the submitted manuscript and even say in the text that we did it (L. 102):

> *'Since the dust concentrations approximately follow an exponential distribution, we consider the negative natural logarithm of the dust concentration in order to emphasise the similarity to the $\delta^{18}O$ time series.'*

In the revised manuscript we have multiplied the log(dust) by minus one.

L90: A discussion of concentration vs flux of dust could be added

We agree with the referee that this point merits being mentioned. We added (L. 99)

> *'Typically, atmospheric changes affecting the dust flux onto the Greenland ice sheet are accompanied by changes in the snow accumulation of opposite sign Fischer et al. (2007).'*

L91: Data is -> Data are

Thank you, corrected.

L96: This was pointed out by others previously: Rial and Saha (2011), Abrupt Climate Change: Mechanisms, Patterns, and Impacts, Geophysical Monograph Series 193, Mitsui and Crucifix, arXiv:1510.06290, Lohmann and Ditlevsen, 2018, Clim Past 14. (and probably others).

Thank you, we have included these citations and have rephrased the sentence, in order to avoid attributing the finding to Boers et al. (2018).

It now reads:

> *'With regard to (i) a low-frequency influence of the background climate on the proxy values and on the frequency of DO events is evident (see Fig. 1), with suppressed DO variability during the coldest parts of the glacial and longer interstadials for its warmer parts (e.g. Rial and Saha, 2011; Mitsui and Crucifix, 2017; Lohmann and Ditlevsen, 2018; Boers, 2017, 2018).'*

L110: I do not understand why

This was an oversight, we included the autocorrelation as Fig. 2 in the revised manuscript. We replaced *'auto-covariance'* by *'autocorrelation'* in the main text.

L125: Stationarity require certain properties of a(x) (such that the process does not drift to infinity. It is better to denote it "homogenous" or "autonomous". (same comment on L138).

Thank you for making us aware of our slightly unprecise usage of the term *'stationary'*. In this revised version we no longer address the modelling of the data via SDEs but solely via the Kramers–Moyal equation.

L126: Langevin equation is continuous -> Langevin equation generate realizations which are continuous

Idem.

L155: There is no Eq 4. (4a and 4b are hardly equations)

Agreed. Yet in this revised manuscript we no longer discuss these.

L185: The purely continuous process (gaussian process) diffuse proportionally to t^(1/2) not t. That is: (sigma(t)=sqrt(4Dt)). The most natural jump processes in this context are the

alpha-stable processes, they do exhibit similar scaling relations with time (sigma(t)~t^(1/alpha)).

Idem.

L205: Also referring back to Figure 1: Wouldn't it be natural to rescale the data before doing the analysis.

We have revised the manuscript and now rescaled all data by their respective standard deviations.

L239: I do not understand the statement (which I believe is not correct): This indicates that d18O exhibits faster dynamics than dust. Please explain.

This is no longer discussed in the revised manuscript.

L245: The exotic explanation for monostability through a complex noise structure seems a little out of context here: The reason why the d18O record has a single maximum in the PDF (with a shoulder) is the sawtooth shape of the DO events masking the obvious two state nature of the record.

This is a fair point. Nonetheless, we believe our two-dimensional analysis provides an adequate explanation for the phenomenon. Following Fig.3d, we can see that the $\delta^{18}$O only has one stable fixed point, yet its location in state space depends on the dust. The monostability of the $\delta^{18}$O thus relates directly to: 1) the monostability of the proxy 2) the extent of time spent in the stadial state.

Section 4.2.1 is obscure to me.

We refer to our reply above relating to the double-fold bifurcation.

L273 and L288: 4 (d) <-> 4(c) .

Thank you, this was corrected.

L456: What happened to the index _t in mu and sigma^2?

Thank you, this has been removed.

I hope my comments are useful.

Very much so! We thank the reviewer for the very helpful commentary and for pointing out vital information we had failed to include in the manuscript.

**– THIS IS THE OLD STUFF –**

Answers: green
Comments by reviewer: violet

Comment on esd-2021-95

Peter Ditlevsen (Referee)

Referee comment on "Changes in stability and jumps in Dansgaard–Oeschger events: a data analysis aided by the Kramers–Moyal equation" by Leonardo Rydin Gorjão et al., Earth Syst. Dynam. Discuss., https://doi.org/10.5194/esd-2021-95-RC1, 2022

First of all, we would like to thank the referee for his careful and constructive review and the overall positive feedback. We have in fact adopted most of his suggestions and are convinced that this has substantially improved the manuscript. The changes that we have made to our manuscript will become clear from our point-by-point answers to the referee's comments below.

In this paper two paleoclimatic ice core records are analyzed. These, the water isotope, d18O, and the dust concentration records are analyzed for the period 59-27 kyr BP, which is the glacial period dominated by regular occurrences of Dansgaard-Oeschger events. There are two major points in the paper: Firstly, the data are modelled as a stochastic process using the Kramers-Moyal equation to investigate the importance of (discontinuous) jumps in the noise. Secondly, the two records are modelled as a two-dimensional joined process.

It seems to me that the two points are only loosely related, and the authors could consider presenting them in two separate papers.

I enjoyed reading the paper and find it publishable. However, there are a few issues below calling for revisions before publication. I have two major concerns regarding the two parts, and some minor points: As to the first point, I have not seen the Kramers-Moyal (KM) equation applied to these data before, so this is a novel approach. The equation, for which the Taylor expansion of the conditional probability density function is taken to higher order than two, covers the case where the noise term in the governing Langevin equation is not gaussian, but contains jumps. It is stated that in the case of Levy processes the Fokker-Planck equation does not apply.

Actually, for the most relevant class of Levy processes, the alpha-stable Levy processes an extension of the Fokker-Planck equation based on the characteristic function of the alpha-stable process exists (see: Samorodnitsky and Taqqu (1994) 'Stable non-gaussian processes, Chapman and Hall, NY. or Ditlevsen, PRE, 60, 172-179).

We thank the reviewer for this comment, it is a clear oversight on our part to state that one cannot describe Lévy-driven processes (or Lévy-noise driven Langevin equations) with a Fokker–Planck equation.

We have corrected our incorrect statement regarding the limitation of using the Fokker–Planck equation solely to describe Wiener processes. The following lines were added:

> *'It should be noted that one can extend the Fokker–Planck equation (or equivalently the Langevin approach) to other types of Lévy noise, e.g. α-stable distributions, and still correctly model the statistical properties of such systems with a Fokker–Planck equation (Ditlevsen, 1999b). In a more general description, generalised Fokker–Planck equations (Denisov et al., 2009) offer another avenue capable of describing non-Markovian properties as well as various Lévy-driven processes.'*

The challenge in applying the KM equation is the estimate of the higher order coefficients (eq. 6) for the data series: Since the higher order terms are (increasingly) dominated by the extremes in the increments the finite time series very quickly "dilutes". A main result (eq. 9 and figure 3) includes the sixth moment of the increments. I thus miss an analysis of uncertainties and reliability in these estimates. I find that this is essential for publishing this (nice!) result.

We have included in Fig. 3, in shaded areas, the standard deviation associated with the evaluation of Eq. 9 for different starting data-points of under-sampling. I.e., in order to calculate the Q-ratio we consider sequentially undersampled trajectories of the time series (this is given by τ in the x-axis). We consider different starting points of the undersampling, e.g. when τ=10 years we take every second data-point x_{2t} as well as x_{2t+1}.

Moreover, we will include an explicit formulation to flash out what exactly is meant by discontinuity in our context, as also requested by the other referee in his review. This can be found in the new appendix, App. G.

Another point which could be given a little more attention is the fact (as also correctly stated) that the strong time-asymmetry in the data (the sawtooth shape) cannot be captured by the model. How does this influence the relevance in including higher order terms (higher than second order) in eq. 5?

Whether there is an influence, we are not in a position to say so. There is *a priori* no necessary relation between the observed time asymmetry in the data and the presence of discontinuities in the data. Under the (somewhat strong) assumption that we can model δ18O as a time-homogeneous process, time asymmetry is likely due to an external coupling to other variables – which nevertheless leaves the question, why do we observe discontinuous trajectories in the data?

We added the paragraph:

> *'We should note that, a priori, the presence of discontinuous trajectories in the δ18O can potentially have no relation with the observed time asymmetry. Whether these two phenomena are connected, we cannot and no do know, nor can we assert, at this stage. We can nevertheless note that, from a one-dimensional modelling point-of-view, having discontinuous trajectories is sufficient to break time symmetry. It is naturally much easier to break time asymmetry by assuming a time asymmetric stochastic model.'*

[PREVIOUS TEXT BY KENO]

We thank the reviewer for the remark. A time asymmetric one dimensional stochastic process is very likely to exhibit non-zero 4th-order KM coefficients. On the contrary, designing a time asymmetric stochastic process with drift and diffusion exclusively is far more difficult in a purely autonomous setting – yet not impossible. Thus, not taking into account higher order KM coefficients in the one-dimensional analysis might lead to missing important aspects of the dynamics. To emphasise this aspect in the manuscript we have added the following statement:

L.137 - revised manuscript:

The recognition by Ditlevsen (1999), that the noise in the processes which lead to DO variability is more complex than Gaussian white noise was picked up only recently in the work by Gottwald (2020) and was one reason for us to investigate the NGRIP records of δ 18 O and dust in the Kramers–Moyal framework free of a noise model specification. Additionally, the fact that the strong time-asymmetry present in the data is difficult to reproduce with one dimensional Langevin type models, provides additional motivation for both: including higher order KM coefficients and investigating potential couplings between different paleorecords. Our research extends beyond the work presented by Hassanibesheli et al. (2020) in terms of the two-dimensional analysis. Furthermore, using higher resolved data we obtain qualitatively different results already in the one-dimensional setting.

We use the discrepancy between the pure Langevin dynamics, as suggested by the dust's one-dimensional KM estimation and the aforementioned observation of the sawtooth shape to motivate the two-dimensional analysis.

However, the statement we made in the manuscript

l.233     *Note that the model equations employed here are by construction symmetric with respect to time, therefore, as it is, the model cannot reproduce the temporal asymmetry that is visually suggested in the dust record.*

is not precise, in the sense that only after having estimated the drift and the constant diffusion and the absent 4th-order KM coefficient, the KM results for the dust are inconsistent with the apparent time asymmetry – this inconsistency is thus not by construction. Accordingly, we have rephrased this statement as follows:

L.263 - revised manuscript

'Note that the reconstructed double-well potential in combination with the evidenced additive noise cannot break time symmetry. Thus, the estimated KM coefficients explain important characteristics of the dust record, but not all of them.'

[END OF PREVIOUS TEXT BY KENO]

As to the second point, the major results are presented in figure 4. Obviously, when considering a one-dimensional record, the drift can always be seen as a result of a potential. This is not the case in two - and higher dimensions, where gradient drift is a non-generic case. I'm sure that the authors are aware of this, the drift is a two-dimensional flow field, as also shown in the small inserts in the subplots of figure 4. I find the construct of pseudo-one-dimensional potentials ($V(x\_1|x\_2)$) both confusing and useless. I suggest that the authors consider abandoning this all together (as well as the notion of a potential landscape). The interpretation in figure 4(c) of a double fold bifurcation is obscure, and -I believe- wrong.

We thank the reviewer for the comment. We have removed the use of "pseudo one-dimensional" potentials. The individual changes, wherever we referred to the 1D conditional potentials in the 2D analysis can be seen in the supplemented tracked-changes file and refrain from discussing every individual change made with respect to this point, here.

In the discussion of the coupled two-dimensional dynamics, we believe it is still useful to discuss the two components of the drift (along the d18o and along the dust direction) separately as functions of the two-dimensional space spanned by d18o and dust. We justify this in the revised manuscript as follows (L. 308 XXX):

*'We will first discuss the two drift components separately as functions of the two-dimensional space spanned by dust and $\delta^{18}O$. In the component-wise analysis, the analysed component takes the role of a dynamical variable, while the respective other assumes the role of a controlling parameter. The structure of the respective drift-component's nullclines has important implications for the combined two-dimensional dynamics, which we address subsequently.'*

Regarding the dust-nullcline we stick to our assessment, that it shows the typical structure of a double-fold bifurcation parametrized by $\delta^{18}O$. Given the referee's comment on Figure 4(c) of the original manuscript (together with a similar comment by the other referee) we have made several changes to Sect. 4.2.1 in order to convey this interpretation more convincingly.

There remains of course the possibility that the double-fold structure that we observe is spurious and simply an artefact arising due to the scarcity of data (especially in the region of the state space where we observe the saddle-node bifurcations) - we point this out in the revised manuscript

L. XXX

*'In particular, the apparent double-fold structure of the dust-nullcline could in the worst case be a spurious result due to the data scarcity in the regions around the two bifurcation points. One could imagine a scenario, where the nullclines would in fact be three straight parallel lines that would not fold and merge. Such parallel nullclines (the outer two being stable and the inner being unstable) correspond to a globally bistable drift independent of the δ 18 O value. We have, however, tested our method against synthetic data from that particular scenario and found the parallel nullclines to be well reconstructed, even with limited amount of data.'*

We pursued two approaches to rule this out.

Together, these findings make us confident that the interpretation of Fig. 4c as a double-fold bifurcation in the dust (conditioned on $\delta^{18}O$) is meaningful. After all, studying the drift component's nullclines we find the hypothesis that the bistability is rooted in the Greenland temperatures (represented by d18o), which then drives the dust variable as a dependent variable, is highly unlikely. In contrast, from the dust-nullcline's shape we can conclude that the dust — and therefore the climate variable(s) it represents — remains a potential candidate for accommodating the bistability of the Arctic climate system during the last glacial. This result is consistently found in the one and two-dimensional analysis.

What I find interesting is the scatter plot in 4(a), which nicely explain the results in figure 2, namely that the stationary distribution for the dust is bimodal while it is unimodal for the d18O: This corresponds to the marginal distributions in 4(a) (projections onto the axis.

The authors could consider analyzing a rotation (linear combination of the two variables) of the data along an axis connecting the two maxima (GS and GI) and a perpendicular direction. In this way one would obtain a "clean" two state dynamics and a "clean" one state perpendicular dynamics. First thing would be to check for independence. Just a suggestion.

We thank the reviewer for the suggestion. We have included a new appendix, App. F, wherein we include an identical KM analysis to a rotated two-dimensional system. We note that the coupling of the two rotated variables is *not negligible* and supports the existence of the double-fold structure we have previously presented – and continue to argue for.

**Minor points:**
* * *
Introduction: These data have been analyzed over many years and a lot is known. For a better overview and setting the present work in context, a representative presentation of work done over the years would be useful. There is a strong bias towards very recent publications.

We agree with the reviewer that there is indeed a bias towards the works in the more recent past in the original manuscript. In response to the this, we have added

- Ganopolski 2002
- Gildor2003
- VelezBelchi2005
- Ditlevsen2005

behind 'Different patterns of interaction between the AMOC, sea ice cover, the Northern Hemisphere atmospheric circulation, and even the continental ice sheets have been proposed and explored to explain the emergence of DO cycles'. For the context of our study, work that approaches Greenland ice-core proxy time-series from a stochastic process perspective is most relevant. Here we identified

- Ditlevsen 1999
- Ditlevsen 2005
- Livina 2010
- Rial and Saha (added in the revised manuscript)
- Kwasniok 2013
- Mitsui 2017 (added in the revised manuscript)
- Boers 2017
- Lohmann and Ditlevsen 2018
- Hassanibesheli 2020

We ask for your understanding, that the enormous amount of studies on DO variability makes it impossible for young researchers to be aware of all of them - and that young researchers have a natural tendency to the newer literature, which in turn builds upon older work and develops the ideas presented therein.

Again: I'm not fan of the "potential landscape" metaphor.

This has been removed entirely in our revised manuscript.

Figure 1: I suggest to plot the dust record upside-down. This will visualize the strong dependence between the two records, and also make the saw tooth shape in the dust record much more apparent. Make the figure full text width, Ylabel: ln(dust) (no units), d18O (permil). Or normalized w.r.t. std. dev.

We thank the reviewer for the suggestion. In fact, we meant to do this already in the submitted manuscript and even say in the text that we did it (L. 87 XXX):

> 'Since the dust concentrations approximately follow an exponential distribution, we consider in the following re-scaled values by taking the natural logarithm and multiplying by −1 in order to emphasise the similarity to the $\delta^{18}O$ time series (cf. Fig. 1).'

In the revised manuscript we have multiplied the log(dust) by minus one.

L90: A discussion of concentration vs flux of dust could be added

We agree with the referee that this point merits being mentioned. We added

L. XXXX

> 'Note that typically atmospheric changes affecting the dust flux onto the Greenland ice sheet are accompanied by changes in the snow accumulation of opposite sign. This enhances the corresponding change of the recorded dust particle concentration. However, for high accumulation Greenland ice cores the dust concentration changes still serve as a reliable indicator of atmospheric changes according to Fischer et al. (2007).'

L91: Data is -> Data are

Thank you, corrected.

L96: This was pointed out by others previously: Rial and Saha (2011), Abrupt Climate Change: Mechanisms, Patterns, and Impacts, Geophysical Monograph Series 193, Mitsui and Crucifix, arXiv:1510.06290, Lohmann and Ditlevsen, 2018, Clim Past 14. (and probably others).

Thank you, we have included these citations and have rephrased the sentence, in order to avoid attributing the finding to Boers et al. (2018).

In the original version, it said:

*'However, Boers et al. (Boers, 2018) pointed out a low-frequency influence of the background climate, for example, expressed in terms of global ice volume, on the frequency of DO events, with suppressed DO variability during the coldest parts of the glacial such as the Last Glacial Maximum.'*

This was changed to:

*'However, a low-frequency influence of the background climate, for example, expressed in terms of global ice volume, on the frequency of DO events, with suppressed DO variability during the coldest parts of the glacial such as the Last Glacial Maximum is evident (e.g. Rial and Saha, 2011; Mitsui and Crucifix, 2017; Lohmann and Ditlevsen, 2018; Boers, 2018).'*

L110: I do not understand why

This was an oversight, we included the auto-correlation in the appendix, not the auto-covariance.

We replaced *'auto-covariance'* by *'auto-correlation'* in the main text.

L125: Stationarity require certain properties of a(x) (such that the process does not drift to infinity. It is better to denote it "homogenous" or "autonomous". (same comment on L138).

Thank you, for making us aware of our slightly unprecise usage of the term *'stationary'*. In fact, here and at several occasions in the manuscript, we have replaced *'stationary'* by *'time-homogeneus',* since this is the right term in these contexts. Please see the tracked-changes files, for all occasions.

L126: Langevin equation is continuous -> Langevin equation generate realizations which are continuous

Thank you, we have replaced the sentence

*'While the Langevin equation is continuous in time, stochastic processes can in principle have discontinuous features, such as sudden jumps.'*

By

*'While the Langevin equation yields time-continuous trajectories, stochastic processes can in principle have discontinuous features,such as sudden jumps.'*

L155: There is no Eq 4. (4a and 4b are hardly equations)

Since we refer to the relation expressed by Eq.4 later in the text, we have merged Equations 4a and 4b into one numbered equation (4).

L185: The purely continuous process (gaussian process) diffuse proportionally to t^(1/2) not t. That is: (sigma(t)=sqrt(4Dt)). The most natural jump processes in this context are the alpha-stable processes, they do exhibit similar scaling relations with time (sigma(t)~t^(1/alpha)).

We have corrected this in the manuscript and have now included a new appendix, App. G, wherein we discuss what we mean by discontinuity. We have included as well the following statement in the text (L. XXX):

> *'We can picture this is a simple manner: A process without jumps takes a certain amount of time to diffuse over its state space; A jumpy process can, with a jump, cover most or all of its state space in an instant.'*

L205: Also referring back to Figure 1: Wouldn't it be natural to rescale the data before doing the analysis.

We have revised the manuscript and now rescaled all data by their respective standard deviations.

L239: I do not understand the statement (which I believe is not correct): This indicates that d18O exhibits faster dynamics than dust. Please explain.

From our estimates of the KM coefficients we see that both the drift and the diffusion along the d18o dimension exceed the ones along the dust dimension in the normalized units by around one order of magnitude. In the revised version we refrain from interpreting this as a fast-slow system, and replaced the sentence

*'This indicates that $\delta 18 O$ exhibits faster dynamics than dust.'*

By the more descriptive sentence

*'This indicates that $\delta 18 O$ is exposed to stronger noise, but at the same time, perturbations decay faster as a consequence of the stronger restoring force.'*

L245: The exotic explanation for monostability through a complex noise structure seems a little out of context here: The reason why the d18O record has a single maximum in the PDF (with a shoulder) is the sawtooth shape of the DO events masking the obvious two state nature of the record.

This is a fair point. We aimed at highlighting a discrepancy between the monostable potential and the two-regime character of the time series. The pdf can be explained from the data, as the referee correctly points out. The two regimes, however, contradict a monostable drift - at least at first sight. A two-regime time series in the presence of a mono-stable potential can be explained from either complex noise structures, such as jumps, or via a coupling to exogenous, hidden variables (see our answer to the other referee). Both manifest changes that can potentially be encoded in the 4th-order KM coefficient. In the revised version of the manuscript, we have left the original sentence unchanged, but supplemented it with more detailed explanation, such that the reference to the 'complex noise structure' would be less out of context:

*'In fact, if the two regime character of the time series was rooted in a complex noise, this would give rise to higher order KM coefficients evidenced in our analysis. Testing, if the non-vanishing fourth order KM coefficient truly was the source of the regime-switching, would require defining explicit process models, which goes beyond the scope of this study. Secondly, a similar effect can arise from coupling to hidden variables. In that case, the description of the data as a one-dimensional Markov process is actually not justified. Since the observed δ 18 O record is the projection of a very high dimensional trajectory onto a very specific subspace, it seems in fact plausible that important information about the dynamics is not captured by this projection. With the dust record being the only available data that covers the same period with the same temporal resolution, it is obvious to at least try to gain further insights by investigating the two records in a coupled two-dimensional framework. But before we move on to the two-dimensional analysis, we shall come back to the question of time-continuity of the investigated processes.'*

With this additional explanation, we aimed to address also the other referees subjection, that the d18o-dust system might not be time-homogenous due to coupling to hidden variables.

Section 4.2.1 is obscure to me.

We refer to our reply above relating to the double-fold bifurcation.

L273 and L288: 4 (d) <-> 4(c) .

Thank you, this was corrected.

L456: What happened to the index _t in mu and sigma^2?

Thank you, this was corrected.

I hope my comments are useful.

Very much so! We thank the reviewer for the very helpful commentary and for pointing out vital information we had failed to include in the manuscript.

---

## Author Response (AR2)

**Referee #1 (Reza Tabar)**

1) By checking the number of minima (k1) of PDF of two variables and number of attractive fixed points (k2) of drift terms, rule out that dynamics may have noise-induced phase transition. For k1 \neq k2 (k1 > k2) dynamics will have noise-induced transition.

We believe that this comment relies on a misunderstanding with respect to the term 'noise induced transition'. In our manuscript, the term refers to a transition between two stable states in response to a strong noise pulse. However, the same term is used to describe the transition from a unimodal to bimodal pdf in response to an increase of the noise amplitude in a monostable system. We believe that the referee had this second meaning in mind.

To prevent this misunderstanding we explicitly added the term 'noise-induced transition' to the description thereof (l.24):

*Second, random perturbations may push the system across a basin boundary (noise-induced transition).*

In fact, this concept might be relevant in view of the monostable $\delta^{18}O$ drift and the two-regime character of the record. Also, the reference (Majda, 2006) provided by referee #3 relates to this point to some extent. The combination of two apparent regimes and a single stable fixed point in the $\delta^{18}O$ time series and the role of multiplicative noise in this context will be discussed in upcoming research already underway. Both, this comment and the one by referee #3 are very helpful in this regard.

2) In lines 150, checking the short range correlations of increments will not show the Markovianity of time series. For linear Ornstein-Uhlenbeck process, despite of the process is Markov, the increments have negative correlations, see Ex. 21.1 (Tabar2019).

Remove this statement. Use a \chi^2 test (similar to sec. 16.4) or state that this is your assumption.

We agree with the referee, that the short range correlation of the time series' increments does not prove Markovianity. However, it does rule out important long-term memory effects. In the revised manuscripts, we emphasize that we do not provide a sufficient criterion for Markovianity.

Original:

*The autocorrelation functions of the increments of both proxies shown in Fig. 2 exhibit weak anti-correlation at a shift of one time step and exhibit negligible correlations beyond this. Such small level of correlation certainly speaks against memory effects to have played a major role in the emergence of the given time series and hence in favour of considering the data Markovian.*

Revised:

*The autocorrelation functions of the increments of both proxies shown in Fig. 2 exhibit weak anti-correlation at a shift of one time step, while correlations beyond this are negligible. Such a small level of correlation certainly rules out long-term memory effects to have played a major role in the emergence of the given time series. Bear in mind that this is a necessary yet not sufficient criterion to consider the data Markovian. For practical reasons we refrained from further Markovianity tests.*

3) Before Eq. 3, state combined \delta^{18}O and dust, … by two dimensional Langevin equation (using the It\^o description)…

We thank the referee for the comment. In the revised manuscript, we specify that the equation (3)

$$dx = F(x)dt + d\xi$$

must be understood in the Ito sense. We considered referring to the equation as a Langevin equation, but decided not to do so for the reason given also in the revised manuscript (l. 177):

*Notice that we could formulate our method equally well in terms of the simpler Fokker–Planck equation. However, operating with the Fokker–Planck equation implicitly assumes that the stochastic process under investigation follows a Langevin equation in a strict sense, i.e. the noise term in Eq. (3) would be restricted to the case of Brownian motion. However, in ongoing research we find indications that the description of the driving noise $\xi(t)$ as Brownian motion might be overly simplistic (Rydin Gorjão et al., 2023). The use of the KM instead of the Fokker–Planck equation in this work aims at emphasizing that $\xi(t)$ might be more complex than Brownian motion and contain for example discontinuous elements.*

4) In 195 the Silverman rule of thumb is valid for estimations of PDFs, there is not such rule for KM coefficients. Authors can use a gaussian kernel with bandwidth h=0.3.

We agree with the reviewer. Indeed Silverman's rule is only valid for the the estimation of PDFs, yet currently there are no known ideal bandwidth rules for KM coefficients. We believe it is best to use 1) the same "ideal" bandwidth and "ideal" kernel that we employ for the estimation of the PDF for the estimation of the KM coefficients; 2) we believe a kernel with bounded support is better since it necessarily only captures local effects in the estimation and does not require truncation in computational estimation.

We added a notice to the text to warn the reader about this particular issue. It reads (line 212):

*We note that the above formula for the ideal bandwidth has been developed for the estimation of the probability density function. As there is currently no consensus on the optimal kernel and bandwidth for the estimation of the KM coefficients, we will employ an Epanechnikov kernel with bandwidth h s throughout our work.*

5) Appendix A has not any content. The title of the appendix is an open problem.

We thank the referee for highlighting this oversight from our side. This was an artifact from a previous iteration of the manuscript that we overlooked. It has now been removed.

We thank the referee for the various suggestions and corrections and for highlighting some issues within our analysis. We hope that our revised manuscript and replies comprise an improved version of our work.

**Referee #3**

This paper employs a non-parametric kernel-density estimation of the drift coefficient of a two-dimensional stochastic process involving the \deltaO18 and dust NGRIP records to shed some light on DO events.

The author's findings are consistent with the view that atmospheric dynamics (represented here by the dust) controls DO events, in terms of stabilising the respective stadial and interstadial states and in triggering transitions a la Kleppin et al. Their findings, as the authors nicely elaborate, corroborate such a perspective.

I believe that the results are useful for the community and warrant publication. I have only a few minor issues the authors may want to consider.

I think the emphasis on Kramers-Moyal is a little too strong. The drift coefficients (6) and (7) follow directly from their equation (3) for the underlying dynamics. A discussion on how this is part of a Kramers-Moyal equation should be included, but maybe with a little less emphasis in the overall presentation.

We thank the referee for this constructive comment. The analysis could indeed also have been conducted in terms of the simpler and more common Fokker-Planck equation. The starting point of our investigation was the estimation of KM coefficients of the individual $\delta^{18}O$ and dust time series, which revealed contributions from non-Gaussian noise in the $\delta^{18}O$ record. However, there remain some discrepancies to be reconciled in that analysis. To maintain consistency across publications and to highlight that the noise is not necessarily Gaussian, we stick to the KM formulation also in this paper. Certainly, this merits explanation within the manuscript itself which we included as follows (l. 177).

*Notice that we could formulate our method equally well in terms of the simpler Fokker–Planck equation. However, operating with the Fokker–Planck equation implicitly assumes that the stochastic process under investigation follows a Langevin equation in a strict sense, i.e. the noise term in Eq. (3) would be restricted to the case of Brownian motion. This conflicts with findings from ongoing research which indicate that the description of the driving noise ξ(t) as Brownian motion might not be applicable (Rydin Gorjão et al., 2022). The use of the KM instead of the Fokker–Planck equation in this work aims at emphasizing that ξ(t) might be more complex than Brownian motion and contain for example discontinuous elements.*

We get back to this point in the discussion section of the revised manuscript (l.284):

*As mentioned previously, a univariat estimation of the individual $\delta^{18}O$ and dust KM coefficients indicates that at least the $\delta^{18}O$ noise comprises non-Gaussian and potentially discontinuous components which could play a central role with respect to the transition between the two identified stable states of the drift (Rydin Gorjão et al., 2022). However, there remain discrepancies to be reconciled in the analysis of the higher-order KM coefficients of the individual $\delta^{18}O$ and dust time series and until then, arguments about the role of non-Gaussian noise in the state transitions remain speculative. Ideally, higher-order KM coefficients should be computed for the two-dimensional record, however, this is prevented by the low data resolution.*

When I read the title, I was slightly taken aback that only the drift coefficient is considered.

Yes, we fully agree that the original title of the manuscript was misleading. It was inherited from an even earlier version of the manuscript. We replaced the title by

*Stable stadial and interstadial states of the last glacial's climate identified in a combined stable water isotope and dust record from Greenland*

I understand the difficulties in the estimation and interpretability of the higher-order coefficients, and the authors have clearly outlined this, and this is not a criticism of the work done and the results obtained, but simply a matter of presentation/refocus.

The authors are careful not to overstate their results and point to several limitations of their approach, which is much appreciated and puts their interesting findings in a broader context.

I would like to add one more point of caution: the authors' focus on the bimodality of the probability density function may be too simplistic to study regimes. Multimodal probability density functions are not necessary for the existence of regimes. For example, in cyclostationary systems (see Wirth (2001)) and chaotic systems with intermittent dynamics (see Majda et al (2006)) the probability density function may be unimodal whereas, if restricted to time windows focusing on the regimes, "hidden" regimes may be identified. This is an issue in atmospheric data (see also the discussions in Franzke et al. 2008, 2009). If the authors agree, it may be worthwhile to add this discussion (again, I don't think that the data allow additional analysis along these lines and I am not asking the authors to do this; simply, if they see fit, adding a discussion in the text).

We thank the referee for this remark. Indeed, the concepts and methods presented in the references could be relevant as well for the study of the NGRIP record. In particular, when studied in isolation the $\delta^{18}O$ record appears unimodal, while clearing exhibiting the signature of two-regimes. We will shortly present additional analysis wherein we assess the KM coefficients of the univariate dust and $\delta^{18}O$ time series. We think that this comment might be particularly helpful for this upcoming research. With respect to this manuscript, we believe that we already made an effort to carefully disentangle bistability, bimodality and the existence of two regimes. Still, we find the reference to Majda, 2006 important and helpful and therefore added the following comment (l.114):

*Notice that the somewhat counterintuitive combination of meta-stable distinct dynamical regimes and unimodal distributions of the associated variables has been discussed also in the context of atmospheric dynamics (Majda et al., 2006).*

Wirth, V., 2001: Detection of hidden regimes in stochastic cyclostationary time series. Phys. Rev. E, 64, 016136,

A. Majda C. Franzke, A. Fischer, and D. T. Crommelin, 2006: Distinct metastable atmospheric regimes despite nearly Gaussian statistics: A paradigm model. Proc. Natl. Acad. Sci. USA, 103, 8309–8314

C. Franzke, D. T. Crommelin, A. Fischer, and A. J. Majda, 2008: A hidden Markov model perspective on regimes and metastability in atmospheric flows. J. Climate, 21, 1740–1757.

C. Franzke , I. Horenko, A. J. Majda, and R. Klein, 2009: Systematic metastable atmospheric regime identification in an AGCM. J. Atmos. Sci., 66, 1997–2012.

Typos etc:

Page 4: All ages are according to —> The dating was performed according to ?

We merged the two sentences

*This translates into non-equidistant temporal resolution ranging from sub-annual resolution at the beginning to ~ 5 years at the end of the period 59944.5 – 10276.4 yr b2k. All ages are according to the Greenland Ice Core Chronology 2005 (GICC05), the common age-depth model for both proxies (Vinther et al., 2006; Rasmussen et al., 2006; Andersen et al., 2006; Svensson et al., 2008).*

into one sentence (l.85)

*This translates into non-equidistant temporal resolution ranging from sub-annual resolution at the beginning to ~ 5 years at the end of the period 59944.5 – 10276.4 yr b2k according to the Greenland Ice Core Chronology 2005 (GICC05), the common age-depth model for both proxies (Vinther et al., 2006; Rasmussen et al., 2006; Andersen et al., 2006; Svensson et al., 2008).*

Eqns (6) and (7): I would delete the n! and m! since only the cases n,m=0,1 are considered (and n,m do not appear on the left-hand side anyway).

Agreed, our need to maintain generality is unnecessary, as we only consider n and m either 0 or 1. Thank you, we changed this accordingly.

Page 8: first line: normalisable —> normalised?

Thank you, corrected.

Page 8: s-shape —> S-shape?

Thank you, corrected, and corrected as well in the caption of Fig. 3.

Figure 3: the figure captions for (c) and (d) should be D_{1,0} and D_{0,1} (as in the respective figure labels).?

Thank you for spotting this, the figure's annotations were changed around. We have corrected this.

We thank the referee for the various remarks and corrections, which have helped improve the manuscript. We submit our revised manuscript once more for a renewed appreciation.